# PARAMANU: A FAMILY OF NOVEL EFFICIENT GENERATIVE FOUNDATION LANGUAGE MODELS FOR INDIAN LANGUAGES

## ABSTRACT

We present PARAMANU (which means "atom" in multiple Indian languages), a *family* of novel language models for *Indian languages*. It is a collection of auto-regressive monolingual, bilingual, and multilingual Indian language models pre-trained from scratch, currently covering 10 Indian languages (Assamese, Bangla, Hindi, Konkani, Maithili, Marathi, Odia, Sanskrit, Tamil, Telugu) across 5 scripts (Bangla, Devanagari, Odia, Tamil, Telugu). The models are pretrained with a context size of 1024 on a single GPU, and are of varying sizes ranging from 13.29 M to 367.5 M parameters. We proposed a RoPE embedding scaling method that enables us to pretrain language models from scratch at larger sequence length context size on single GPU without increased GPU memory. ~~than the equivalent GPU memory.~~ We have also developed an efficient and advanced novel tokenizer with least fertility score among existing LLMs for Indian languages using a combination of BPE and Unigram that can also tokenize unseen languages written in the same script or the Roman script. We also proposed language specific tokenization for multilingual models and domain specific tokenization for monolingual language models. In order to avoid the "curse of multi-linguality" in our multilingual MPARAMANU model, we pretrained on comparable corpora by typological grouping using the same script. We proposed and performed pretraining for more than 1 epoch of training for most of our language models. From our results, we observed the language transfer phenomenon from low resource to high resource within languages of the same script and typology. We performed human evaluation of our pretrained models for open end text generation on grammar, coherence, creativity, and factuality metrics for several languages. Our Paramanu models outperformed standard and multilingual large language models (LLMs) by a large margin in performance despite being smaller in size by 64 to 20 times. We studied the impact of language specific tokenization versus language agnostic tokenization for bilingual language modeling. We also studied the impact of BPE versus Unigram tokenization for Devanagari script languages. We further created instruction-tuning datasets and instruction-tuned our pretrained models on 23,000 instructions in respective languages except Hindi, for which we used 75,000 instructions. Comparison with multilingual LLMs on various commonsense reasoning benchmarks for natural language understanding, natural language inference, and machine reading comprehension shows the advantage of our models. The performance of our Paramanu models leads to the conclusion that high quality generative language models are possible without high amount of compute power (FLOPS) and enormous number of parameters.

## 1 INTRODUCTION

World's new age growth arenas are non-English speakers from countries such as India, China, Indonesia, Mexico, South Africa, etc., where more people will connect to the Internet for information need without language acting as boundaries. Multi-lingualism represents freedom of expression and diversity in a country like India. According to a Guardian article[1] by Andras Kornai, "95% of

---

[1] http://labs.theguardian.com/digital-language-divide/

all languages in use today will never gain traction online"; this highlighting the digital language divide. Despite around 7,000 languages, current NLP technologies cover only 12%, leaving many non-English and non-European languages underserved. This, in our personal opinion, is an extremely incapacitating bias and language divide for the global digital world where the non-English, non-European language world has been left out. According to the Indian Census 2011, there are 22 official languages and more than hundred others with a sizeable number of speakers in India. Both Hindi and Bangla (Bengali), despite being the world's 5[th] and 6[th] most spoken languages respectively according to Babel[2] are still underrepresented in today's NLP technology. Multilingual NLP faces the challenges of having lack of quality benchmark datasets covering diverse languages from different language families and especially under-represented languages, and is typically referred as "low-resource" in the NLP community. Large language models (LLMs) such as GPTNeoX (Black et al., 2022), OPT (Zhang et al., 2022), LLaMa (Touvron et al., 2023), PaLM (Chowdhery et al., 2023), GPT-2-XL (Radford et al., 2019), GPT-J (Wang & Komatsuzaki, 2021), etc. have primarily focused on English and mostly European languages whereas other languages have not been given priority. Bloom (Workshop et al., 2023) is considered to be the biggest multilingual auto-regressive model that has been built till now; it has been pretrained on 45 languages including Indian languages. However, Indian languages are morphologically richer and typologically distinct than languages written in the Latin script and, hence, grouping them together without considering linguistics nuances often leads to poor performance of LLMs for low resource languages.

This work is an attempt to make language technology more accessible for Indian languages. In this work, we focus on 10 Indian languages (Assamese, Bangla/Bengali, Hindi, Konkani, Maithili, Marathi, Odia, Sanskrit, Tamil, Telugu) written in 5 distinct scripts (Assamese-Bengali, Devanagari, Odia, Tamil, Telugu) comprising of more than billion speakers in the global world. Our goal is to show that generative language models for low-resource Indian languages can be trained from scratch with limited compute and token budget, without using English corpora. We excluded English due to its linguistic differences with Indian languages in terms of typology, script, morphology and grammar, aiming to maintain language purity and typology grouping. Additionally, existing multilingual models often have an English-centric bias due to the large imbalance in data. To the best of our knowledge, the existing LLMs and multilingual LLMs struggle to generate grammatically correct and coherent sentences in Indian languages (and reasoning ability is too far) as shown in Appendix Tables 20, 26, 27, 28, 29, 30, 30, 32, 33, 35, 36, 37, 38, 39, 40. Many LLMs even generate text in even Arabic (Table 29) or Japanese (Table 39) scripts when prompted with prompts in Bengali, Hindi, and Sanskrit. We also found that LLMs including ChatGPT (Oct'23) were not able to distinguish languages of the same script such as Bengali versus Assamese. Human evaluation by annotators confirm this (Section A.5 in Appendix). Our models, based on Transformer decoders (Vaswani et al., 2017), are enhanced with improvements in the architecture. These enhancements, improvements and novelties make our models efficient, small but strong. They have been pretrained from scratch to support context size of 1024 without requiring higher physical memory on a single NVIDIA A100-PCIE-40GB GPU. Our model architecture has the ability to capture a much higher context size without requiring equivalent physical memory; in contrast, the LLaMa-1 model was pretrained from scratch using 2048 GPUs for a context size of 2048. Our models are of varying sizes and of three types: (1) monolingual language models, (2) bilingual language models, and (3) multilingual language models. Both bilingual and multilingual language models are pretrained from scratch on comparable corpora and with typological grouping of languages to avoid curse of multilinguality.

We summarize our contributions as follows.

1. We propose a RoPE (Su et al., 2022) embedding scaling method that enables us to pretrain language models from scratch at larger sequence length context size than the equivalent GPU memory. We scaled the RoPE embedding through a shrinking factor by dividing the target context length $y$ by max_permissible_context_size_length on single GPU keeping all other hyperparameters fixed such as batch size, vocabulary size.

2. We propose a novel tokenization method using a combination of both Byte-Pair Encoding (BPE) (Sennrich et al., 2016) and Unigram (Kudo, 2018) algorithms.

3. We perform language specific tokenization for multilingual model and domain specific tokenization for monolingual language modeling.

---

[2]https://www.babbel.com/en/magazine/the-10-most-spoken-languages-in-the-world

4. We perform pretraining on comparable corpora for multilingual/bilingual generative language model to handle data imbalance and curse-of-multi-linguality in multilingual language model.

5. We perform more than 1 epoch of pretraining for most our language models. From our results, we observed the language transfer from low resource to high resource within language of the same script and typology. We found that small models trained for more than 1 epoch on high quality data is better than bigger model trained for 1 epoch on not so good quality bigger corpora. We studied the impact of language specific tokenization versus language agnostic tokenization for bilingual language modeling. We also studied the impact of BPE vs Unigram tokenization for Devanagari script languages as a proof-of-concept.

6. We curated books for pretraining dataset and created an instruction-tuning dataset of 23000 instructions and instruction-tune our pretrained Bangla, ~~Hindi,~~ Marathi, Tamil, and Telugu models using 23000 instructions each in respective languages but for Hindi, we used additional 52,000 Alpaca translated Hindi (Taori et al., 2023), total of 75,000 instructions.

## 2 RELATED WORK

The rise of large language models (LLMs) has led to numerous new models competing across various benchmarks, but most remain English-centric. Bloom, the largest multilingual LLM with 170 billion parameters, utilizes 40% English data and 60% from 44 other languages. Despite extensive multilingual training, many LLMs exhibit a significant English bias. Research has shown that a substantial portion of neurons in these models remains inactive; for instance, in the 66 billion parameter OPT model, over 70% of the feed-forward network (FFN) neurons in certain layers are "dead" meaning they do not activate even on diverse datasets. This sparsity in neuron activation limits the model's ability to learn and generalize across languages, particularly those that differ significantly from English (Voita et al., 2024). Consequently, even when trained on multilingual data, these models struggle to effectively process and generate text in languages beyond English, revealing a critical limitation in their design. Recent efforts to adapt English-centric models like Llama for Indian languages (Airavata (Gala et al., 2024), OpenHathi for Hindi, and Tamil) have involved extending vocabulary and fine-tuning techniques such as LoRA (Hu et al., 2022) and QLoRA (Dettmers et al., 2023). However, these models still exhibit a strong English bias and struggle to generate high-quality text in Indian languages. Massively multilingual models (MMTs) (Devlin et al., 2019), (Conneau et al., 2020), (Xue et al., 2021) are pretrained on large corpora but often lack alignment between languages, leading to poor transfer performance for distant languages (Lauscher et al., 2020). The "curse of multilinguality" (CoM) indicates that adding more languages can degrade per-language performance, necessitating larger model capacities and corpora. While language-specific adapters like MAD-X (Pfeiffer et al., 2020) improve performance, they do not generalize well to unseen languages. Recent advancements, such as MAD-G (Ansell et al., 2021) and BAD-X (Parović et al., 2022), focus on bilingual adapters to enhance language transfer. Our work aims to create dedicated models for low-resource Indian languages by developing language-specific generative models from scratch, emphasizing linguistic features, typology, and tailored tokenization.

## 3 METHODOLOGY

### 3.1 DATASET FOR PRETRAINING

Pretrained data was split into 95%-5% training and validation sets so that we do not lose much data for pretraining as the purpose of this work is a step towards developing pretrained generative effective language models from scratch using our novel architecture for Indian languages. Pretraining data covers web scrapped news, blogs from IndicCorp v2 (Doddapaneni et al., 2023), Bangla literature from Vacaspati (Bhattacharyya et al., 2023), Wikipedia articles, curated books of various genres, subjects, education books, magazines in respective Indian languages representing each distinct language community to cover Indian culture, rich history, and knowledge. Our pretraining corpora have no source code, scientific journals/articles, medical and engineering education books, research papers as these are generally written in English in India. Dataset details can be found in Table 19 in Appendix and data distribution can be found in the Figure 3 in Appendix. We apply the following data cleaning and preprocessing techniques as mentioned in Appendix A.4.

| Models | Size | #Tokens pretrained | # Training A100 hours | Script & Family | # Speakers |
|---|---|---|---|---|---|
| Bloom | 7.1B | 340 B | 1.08M | Multilingual | 1B+ |
| OpenHathi (Llama) | 7B | 1 T | N/A | Hindi (Indo-European) | 692M |
| Sarvam | 2B | 4 T | N/A | Multilingual (Devanagri, Dravidian) | 1B+ |
| Paramanu-Bangla (ours) | 108M | 26.21 B | 19.45 | Bengali (Indo-European) | 300M |
| Paramanu-Hindi (ours) | 367M | 66 B | 239 | Devanagari (Indo-European) | 692M |
| Paramanu-Marathi (ours) | 208M | 28.83 B | 88 | Devanagari (Indo-European) | 99M |
| Paramanu-Odia (ours) | 87M | 52.42 B | 84.5 | Odia (Indo-European) | 43M |
| Paramanu-Sanskrit (ours) | 139M | 45 B | 110 | Devanagari (Indo-European) | 0.025M |
| Paramanu-Tamil (ours) | 208M | 26.2 B | 208 | Tamil (Indo-Dravidian) | 77M |
| Paramanu-Telugu (ours) | 208M | 39.32 B | 112.5 | Telugu (Indo-Dravidian) | 95M |
| mParamanu (ours) | 162M | 26.2 B | 118 | Multilingual (Devanagari) | 1B+ |

Table 1: Pretrained LLM tokens, training hours, script, language family and #speakers.

## 3.2 DATASET FOR INSTRUCTION TUNING

We created a dataset of 5,000 human-annotated instructions covering tasks like poem and novel writing, article summarization, grammar correction, and Q&A on topics such as climate change and healthcare in Bangla, reflecting Bengali culture and linguistics. We then used the Google Translate API (goo, 2023) to translate these instructions into Hindi, Marathi, Tamil, and Telugu. Additionally, we translated 15,000 instructions from Dolly (Conover et al., 2023) to these languages and generated 3,000 instructions using the self-instruct technique (Wang et al., 2023) in five Indian languages. Finally, we fine-tuned our pretrained models except Hindi on a total of 23,000 instructions, including human-annotated, machine-generated, and translated instructions. Since creating multilingual datasets using automatic machine translation from Bengali to other Indian languages introduce translation errors but we also performed human checks and corrected them. We found around 8% word errors on average for Hindi and Marathi using Google Translate.

## 3.3 TOKENIZATION

Figure 1 shows the flowchart of our novel tokenization technique both for monolingual and multilingual settings. We performed domain adaptive tokenization for monolingual models using a combination of Byte-Pair encoding (BPE) (Sennrich et al., 2016) and Unigram (Kudo, 2018). We trained separate Byte-Pair encoding (BPE) (Sennrich et al., 2016) and Unigram tokenizers using Sentencepiece (Kudo & Richardson, 2018) module on the high quality part of the pretraining data from scratch. Then, we merge both the independent tokenizers by intersection respectively of size V'. This is similar to merging two different data structures of same size (considering each tokenizer as a list of tokens) by intersection to remove overlapping elements. We used the merged tokenizer to tokenize the pretraning data. During pre-tokenization, NFC normalization was performed on the processed data; digits are split into individual tokens and unknown UTF-8 characters were reduced to byte granularity. For monolingual tokenization, we trained individual BPE and Unigram tokenizers on domain specific data for respective language with same vocabulary size and merged the tokenizers via merge by intersection to remove overlapping tokens and make our specialised tokenizer compact, optimized, effective and highly effective for monolingual data and also performed the same approach to tokenize the multilingual pretraining corpora by performing language specific tokenization in multi-task way where every task is a language and merged the distinct language specific tokenizers by union via intersection, i.e, removing the overlapping tokens. In this way, our mBharat tokenizer was able to learn language specific tokens based on typology and can generalize to unseen languages of the same script. mBharat tokenizer was also exposed to little amount of English high quality corpus to learn its ability to tokenize languages of the Roman script such as English. From 2, we observe that mBharat tokenizer has the least fertility score of 1.66 for languages in Assamese-Bengali, 1.25 for Devanagari (Hindi) and 1.75 for Odia script among tokenizers of LLMs like Sarvam 2B (Sarvam2B, 2024), LLama-3.1 (Dubey et al., 2024), Gemma-2 (Team et al., 2024), and GPT-4o.

## 3.4 MODEL ARCHITECTURE

Multilingual mParamanu, monolingual Paramanu models (Assamese, Bangla, Hindi, Konkani, Marathi, Odia, Tamil, Telugu, and Paramanu-Sanskrit) and bilingual Konkani-Maithili models are based on transformer (Vaswani et al., 2017) based causal decoder architecture (Radford et al.,

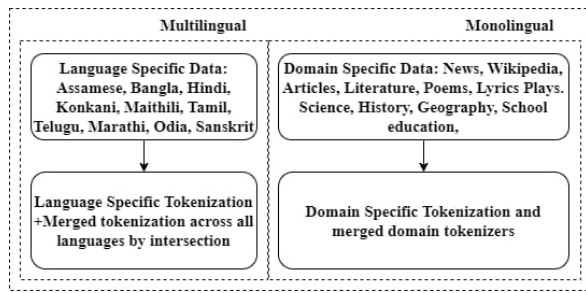

Figure 1: Tokenization technique for monolingual and multilingual setting.

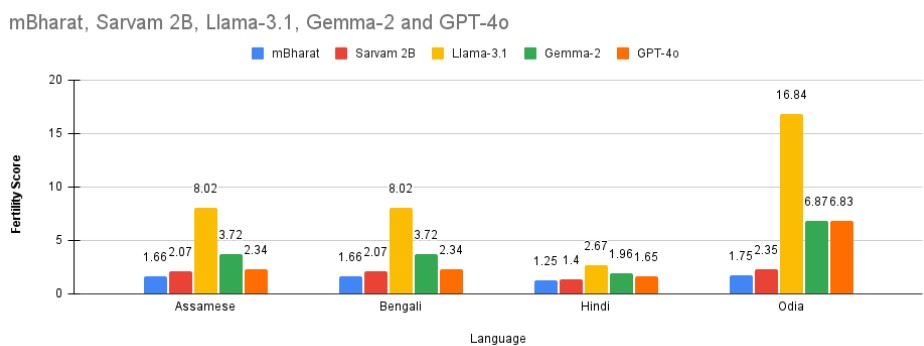

Figure 2: Fertility score of mBharat tokenizer v/s LLMs across languages of 3 scripts (Bengali, Devanagari, and Odia). LLMs score are reported from (Sarvam2B, 2024)

2019) with modifications. The model architecture uses RMSNorm as pre-normalizaion layer with norm_epsilon = 1e-5, approximate GeGLU Activations (Shazeer, 2020) as non-linearity by replacing the standard ReLU non-linearity activation function. The model architecture uses a scaled version of RoPE embedding (Su et al., 2022) with $\theta$=10,000. We scaled the RoPE embedding through a shrinking factor by dividing the target context length $y$ by $permissible\_context\_size\_length$ on single GPU, keeping all other hyperparameters fixed such as batch size, vocabulary size, etc. This allows every $position\_ids$ to be divided by the shrinking ratio in the RoPE embedding methodology. For instance, if the $permissible\_context\_size\_length$ for a given physical memory hardware is 256, then we apply shrinking factor of 16 for target context size of 4096 on Single A100 40G chip during pretraining. Then, a token with $position\_ids$ = 4000 becomes 4000/16 = 250, and the neighbouring token 4001 becomes 4001/16 = 250.06, to be within 0 to 256. This is how we can capture higher context size during pretraining on limited physical memory required to pretrain model at higher context size outside the $permissible\_context\_size\_length$. This modification allows us to pretrain models from scratch at much higher context size than the physical memory required for pretraining. Hence, with limited physical memory and limited GPUs, we can pretrain generative language models from scratch at much higher desired context size. Following (Chowdhery et al., 2023), we remove all biases from dense layers to improve the training stability. Table 17 lists down the various configuration to develop our 13.29M, 26.59M, 87M, 87.25M, 92.63M, 108.5M, 139.3M, 162M, 208M, 350M models. Table 17 lists the different model configuration but we used weight tying (Press & Wolf, 2017) to improve the performance of language models by sharing the weights of the embedding and softmax layers. Therefore, the total number of parameters in our models are typically smaller due to weight tying and different tokenizer sizes of each language specific model than the sizes mentioned in Table 17.

### 3.5 TRAINING

We performed hyperparameter tuning on 15M models to find the optimal tokenizer size for each language specific monolingual corpus, learning rate, learning rate scheduler, and weight decay. We used the concept of $\mu P$ transfer (Yang et al., 2021), and transferred the learned hyperparameters to our bigger models. For further training and hyperparameter tuning details, we mentioned in

Appendix A.3. We find the optimal size of tokenizers by training range of tokenizers of various sizes and optimizing the validation perplexity of the language models and fertility score of the tokenizers.

### 3.6 IMPACT OF LANGUAGE SPECIFIC MERGED-TOKENIZER V/S LANGUAGE AGNOSTIC TOKENIZER FOR BILINGUAL MODEL (KONKANI-MAITHILI 27M GPT) OF TYPOLOGICALLY SAME SCRIPT (DEVANAGARI)

For the language agnostic approach, we merged both the Konkani and Maithili corpus which are of comparable size after our preprocessing in terms of number of sentences (lines) then we trained SentencePiece's BPE algorithm to a vocabulary size of 1750 on the whole corpus and trained our 27M Konkani-Maithili bilingual GPT model and we report the validation perplexity score to be 12.43393. For the language specific approach, we trained individual SentencePiece (Kudo & Richardson, 2018) BPE tokenizers independently on Konkani (vocabulary size of 1k) and Maithili (vocabulary size of 750) corpus respectively. After that, we merged both the language specific tokenizers and also removing the overlapped tokens as both Konkani and Maithili are written in Devanagari script. We used this merged tokenizer to train another 27M bilingual Maithili-Konkani GPT and found the validation perplexity of the bilingual modeling drops to 8.53827 from 12.43393. From our experiments, we infer that language specific tokenization is very important to preserve the language specific linguistics even for thre typologically similar languages (Devanagari script) and the merging operation of the tokenization also helps to omit the overlapping tokens which results in optimized vocabulary size for language modeling resulting in better open-end text generation, lower perplexity score due to optimal size of the tokenizer for multilingual language modeling.

### 3.7 LANGUAGE MODELING FOR PARAMANU-KONKANI (DEVANAGARI SCRIPT) 15M WITH BPE TOKENIZER V/S UNIGRAM TOKENIZER

From our experiments where we tokenized the Konkani corpus with Sentencepiece's Unigram model with a vocabulary size of 1000 and trained a 15 Model and similarly we also trained another 15M model with SentencePiece BPE tokenizer on the same setting to see whether there is an impact of these two different tokenizers on the perplexity score of monolingual language modeling. We found the validation perplexity for Unigram 15M model to be 11.88412 whereas for BPE 15M model is 11.74314 which is slightly lower but not significantly different. However, Unigram tokenizer takes longer time than BPE to get trained on the corpus.

### 3.8 QUANTITATIVE RESULTS BASED ON PERPLEXITY METRIC

Table 2 lists the validation perplexity and MFU metrics of our various pretrained models. In terms of quantitative evaluation of language modeling, the lower the perplexity, the better is the language model. From Table 2, Paramanu-Sanskrit exhibited the lowest validation perplexity among our models, followed by Odia, Bangla, Telugu, mParamanu (162M), Tamil, Marathi, and Hindi, which had the highest perplexity. It is important to note that most models were trained for the same number of steps, regardless of their pretraining dataset size, meaning some may be under-trained. Continued pretraining, such as for Paramanu-Hindi, could further improve perplexity scores.

### 3.9 BASED ON QUANTITATIVE BENCHMARK

We evaluated our models on key Natural Language Understanding (NLU), Natural Language Inference, and Commonsense Reasoning tasks, including their multilingual variants, while benchmarking against multilingual LLMs like Sarvam-2B, Bloom, and Bloomz in two groups, one with LLMs of size <=2B and another group of LLMs of size > 2B. We utilized the translated ARC, HellaSwag, and MMLU datasets from (Lai et al., 2023), employing Eleuther AI's LM Evaluation Harness (Sutawika et al., 2024) for evaluation. Key assessments included HellaSwag (Zellers et al., 2019), which tests common sense reasoning by predicting scenario endings; MMLU (Hendrycks et al., 2021), measuring broad knowledge across diverse subjects; and ARC-Challenge (Clark et al., 2018), which examines complex reasoning with scientific questions. We encountered misalignment issues with the LM Evaluation Harness datasets and our models, preventing evaluations on HellaSwag except for Hindi. Other evaluations included XCOPA (Ponti et al., 2020), assessing cross-lingual commonsense reasoning; XNLI (Conneau et al., 2018), which assesses cross-lingual sentence classification

| Model | Perplexity |
|---|---|
| Paramanu-Assamese 26.59M | 6.620 |
| Paramanu-Bangla 87.25M | 5.069 |
| Paramanu-Bangla 108.5M | 4.102 |
| Paramanu-Hindi 162M | 16.992 |
| Paramanu-Hindi 367.5M | 11.052 |
| Paramanu-Konkani-Maithili 13.29M (merged language specific tokenizer) | 8.538 |
| Paramanu-Konkani-Maithili 13.29M (language agnostic tokenizer) | 12.433 |
| Paramanu-Odia 87M | 3.068 |
| Paramanu-Sanskrit 139.33M | 1.748 |
| mParamanu 92.63M | 8.443 |
| mParamanu 162M | 6.924 |
| Paramanu-Marathi 207.73M | 8.943 |
| Paramanu-Telugu 208.25M | 5.400 |
| Paramanu-Tamil 207.84M | 7.618 |

Table 2: Perplexity of models

| N-shot | XNLI-Hindi | XStoryCloze-Hindi | XStoryCloze-Telugu | XCOPA-Tamil |
|---|---|---|---|---|
| 0 | 33.49 | **52.42** | **56.06** | **54.00** |
| 5 | **34.04** | 51.49 | 54.67 | 52.40 |
| 25 | 33.23 | 52.02 | 55.92 | 49.80 |

Table 3: N-shot evaluation of pretrained Paramanu models across various benchmarks.

across 15 languages; and XStoryCloze (Lin et al., 2022), evaluating story understanding by selecting the correct ending to a four-sentence story. Together, these benchmarks comprehensively assess model performance and reasoning capabilities.

Table 4, Table 5, Table 6, Table 7 and Table 8 evaluate model performance in a zero-shot setting using accuracy metrics across translated benchmarks (ARC, MMLU, HellaSwag) for Bangla, Hindi, Tamil, and Telugu. Table 3 presents n-shot evaluations for XNLI in Hindi, XStoryCloze in Hindi and Telugu, and XCOPA in Tamil. Additionally, Table 5 and Table 6 assess various pretrained monolingual and multilingual models for Devanagari script across MMLU, HellaSwag, ARC, XStoryCloze, XNLI, and Belebele, highlighting cross-lingual language transfer among Devanagari languages (Hindi, Marathi). Notably, mParamanu, pretrained on low-resource Devanagari languages (Sanskrit, Konkani, Maithili), achieved scores of 25.86 for MMLU-Marathi, 24.84 for Hindi, 28 for Belebele-Marathi, and 25.44 for Belebele-Hindi, indicating effective language transfer to medium (Marathi) and high-resource (Hindi) languages using the mBharat tokenizer. Interestingly, neither Paramanu-Sanskrit nor mParamanu were pretrained on Hindi or Marathi but still performed well on their benchmarks. This is possibly due to the same script.

Table 5, Table 13 in Appendix and Table 6, Table 14 in Appendix show that Paramanu models exhibit superior performance across various benchmarks in Devanagari languages, despite their smaller sizes and being pretrained on fewer tokens than larger multilingual LLMs. Specifically, Paramanu-Marathi (208M) outperformed Sarvam (2B), OpenHathi (7B), and Bloom (560M) on the Marathi benchmark. Similarly, mParamanu (162M) outperformed Paramanu-Hindi (367M) and demonstrated competitiveness against larger models. Notably, Paramanu-Hindi-instruct (356M) surpassed all larger multilingual LLMs, except Bloomz (7B), by a significant margin, benefiting from instruction tuning on a dataset of 27,000 Hindi instructions and additional 52,000 Alpaca machine-translated instructions. In contrast, Bloomz (7B) was trained on hundreds of thousands of instructions. If the 367M Hindi model underwent more training steps, it could potentially achieve even better performance, as many models were pretrained for the same duration regardless of dataset size. The stronger performance of mParamanu in Hindi illustrates effective language transfer within the same script and typology. Notably, Paramanu-Sanskrit (139M), pretrained on 45 billion tokens, achieved an average score of 31.05, surpassing both Hindi models and closely approaching Bloom (560M) and Bloomz (560M). Its lower perplexity (1.75) compared to Paramanu-Hindi (11.05) further supports the notion that additional pretraining for the Hindi model could significantly enhance its downstream performance.

From Table 3, we observe that the performance of our models drop from zero-shot setting to 25 shot setting on XNLI-Hindi, XStoryCloze for Hindi and Telugu, and XCOPA for Tamil. This type of phenomenon has also been observed in PlanningBench (Valmeekam et al., 2024) where GPT-3.5-Turbo, GPT-4, GPT4-o performance on Blocksworld dropped from 0 shot to 1-shot significantly.

| Models | MMLU-Bangla | ARC-Bangla | Belebele-Bangla | Average (Bangla) | Belebele-Assamese |
|---|---|---|---|---|---|
| Paramanu-Bangla 108M (ours) | 23.82 | 25.75 | 25.11 | 24.89 | 25.33 |
| Paramanu-Bangla-instruct 108M (ours) | **27.60** | **28.50** | **32.45** | **29.52** | **30.54** |
| mParamanu 162M (ours) | 25.29 | 20.19 | 27.44 | 24.31 | 29.00 |
| Bloom 560M | 22.61 | 26.00 | 22.89 | 23.83 | 22.78 |
| Bloomz 560M (instruction-tuned) | 25.82 | 23.43 | 22.77 | 24.01 | 25.11 |
| Bloom 1.1B | 23.90 | 24.37 | 26.00 | 24.75 | 26.89 |
| Sarvam 2B | 24.05 | 28.40 | 23.22 | 25.22 | 27.78 |

Table 4: Zero-shot evaluation of LLMs (<=2B) across translated benchmarks of MMLU, HellaSwag, ARC datasets, and Belebele in Bengali script. All benchmarks report Accuracy except for ARC which reports Normalized Accuracy. Max scores are in bold.

| Models | MMLU-Marathi | ARC-Marathi | Belebele-Marathi | Average (Marathi) |
|---|---|---|---|---|
| mParamanu 162M (ours) | 25.68 | 22.16 | 28.00 | 25.28 |
| Paramanu-Hindi 367M (ours) | 23.78 | 24.16 | 24.66 | 24.20 |
| Paramanu-Hindi-instruct 367M (ours) | **28.72** | **27.85** | **32.00** | **29.52** |
| Paramanu-Marathi 208M (ours) | 25.39 | 26.49 | 27.33 | 26.40 |
| Paramanu-Sanskrit 139M (ours) | 24.96 | 26.49 | 24.33 | 25.26 |
| Bloom 560M | 22.78 | 24.50 | 27.00 | 24.76 |
| Bloomz 560M (instruction-tuned) | 26.20 | 24.24 | 25.44 | 25.29 |
| Bloom 1B | 23.93 | 25.10 | 28.33 | 25.78 |
| Sarvam 2B | 23.96 | 27.53 | 26.77 | 26.08 |

Table 5: Zero-shot evaluation of LLMs (<=2B) for cross-lingual language transfer in Marathi. All benchmarks report Accuracy except for ARC (Normalized Accuracy). Max scores are in bold.

Perhaps n-shot examples become additional soft constraints on the generation which might be the reason of degradation of performance from the original training dataset. From Table 9, we see that Paramanu-Bangla 108M outperformed Bloom 560M by 1.06% points, Bloomz 560M by 1.05% points, Bloom 1.1B on average score across MMLU, ARC, and Belebele benchmarks, and by 1.21% points on MMLU over Bloom 560M depsite being smaller by 10.2 times compared to Bloom 1.1B and being pretrained on 26.21 billon of tokens. However, Paramanu-Bangla 108M is extensively trained only on Bangla literature corpus. With further instruction-tuning on 27k Bangla instructions, Paramanu-Bangla-instruct 108M outperformed Bloom 560M, Bloomz 560M, Bloom 1.1B, Sarvam 2B, Bloom 7B on average score of MMLU, ARC, and Belebele benchmarks for Bangla respectively.

Table 7 and Table 8 compare pretrained multilingual LLMs (<2B) and instruction-tuned models on Tamil and Telugu benchmarks. Table 15 and Table 16 in Appendix compares our model with LLMs (> 2B) Our model, Paramanu-Tamil (208M), outperformed larger multilingual LLMs like Bloom (560M), Bloomz (560M), and Bloom (1.1B) across four benchmarks (Belebele, XCOPA, MMLU, and ARC) in Tamil, coming close to Sarvam (2B) despite being much smaller and trained on fewer tokens. On MMLU-Tamil, both Paramanu-Tamil and Paramanu-Tamil-instruct outperformed Sarvam (2B) by 2.89 percentage points, with Paramanu-Tamil pretrained on 26.2 billion tokens. Paramanu-Tamil-instruct surpassed Bloom (7B), despite being 34 times smaller, and outperformed Bloom (1.1B) by 2.28 points, Bloom (560M) by 3 points, and Bloomz (560M) by 1.17 points. For Telugu, Paramanu-Telugu-instruct (208M) outperformed Bloom (560M) by 2.72 points, Bloomz by 2.25 points, Bloom (1.1B) by 1.46 points, and Bloom (7B) by 1.22 points, with Paramanu-Telugu pretrained on 39.32 billion tokens. The improvements in metric scores for Tamil and Telugu instruction-tuned models were modest, likely due to lower-quality machine translations from Bangla compared to Hindi. Nonetheless, these results show strong performance of our models on various NLP tasks despite their smaller size and fewer tokens, challenging the notion that larger models are always better. Our findings suggest that smaller pretrained models can excel when trained on high-quality, preprocessed data over multiple epochs, outperforming larger models trained on lower-quality data for an epoch.

### 3.9.1 BASED ON HUMAN EVALUATION

We hard-prompted various LLMs (GPT-2 XL, GPT Neo 1.3B, LLaMa 2 7B, OPT 6.7B, and the multilingual Bloom series) alongside our pretrained models (Paramanu-Bangla, Paramanu-Hindi, and mParamanu for Sanskrit) without fine-tuning. The prompts reflected the local, cultural, and literary contexts of Assamese, Bangla, Hindi, Konkani, Maithili, Odia, and Sanskrit. Due to resource constraints, extensive evaluations focused on Paramanu-Bangla, Paramanu-Hindi, and mParamanu. The top three predictions from each model were generated with temperature = 1.0 and top$_p$ = 0.9.

| Models | MMLU-Hindi | HellaSwag-Hindi | ARC-Hindi | XStoryCloze-Hindi | XNLI-Hindi | Belebele-Hindi | Average (Hindi) |
|---|---|---|---|---|---|---|---|
| mParamanu 162M (ours) | 24.84 | 24.87 | 22.35 | 49.24 | 33.70 | 25.44 | 30.07 |
| Paramanu-Hindi 367M (ours) | 24.38 | 24.83 | 27.05 | 47.92 | 32.00 | 23.33 | 29.92 |
| Paramanu-Hindi-instruct 367M (ours) | **30.25** | 29.42 | **30.23** | 58.00 | 40.25 | **42.78** | **40.14** |
| Paramanu-Marathi 208M (ours) | 25.49 | 26.59 | 23.97 | 48.71 | 33.73 | 27.33 | 30.97 |
| Paramanu-Sanskrit 139M (ours) | 25.16 | 25.64 | 25.17 | 50.23 | 34.46 | 25.66 | 31.05 |
| Bloom 560M | 23.67 | 27.50 | 23.88 | 54.79 | 40.80 | 26.44 | 32.84 |
| Bloomz 560M (instruction-tuned) | 25.87 | 26.48 | 24.40 | 55.53 | 35.58 | 26.00 | 32.31 |
| Bloom 1B | 23.86 | 28.28 | 24.74 | 55.59 | 42.77 | 28.00 | 33.87 |
| Sarvam 2B | 24.54 | **33.66** | 28.00 | **60.29** | **46.74** | 24.44 | 36.27 |

Table 6: Zero-shot evaluation of LLMs(<=2B) for cross-lingual language transfer in Hindi. All benchmarks report Accuracy except for ARC (Normalized Accuracy). Max scores are in bold.

| Models | Belebele-Tamil | XCOPA-Tamil | MMLU-Tamil | ARC-Tamil | Average (Tamil) |
|---|---|---|---|---|---|
| Paramanu-Tamil 208M (ours) | 26.88 | 57.60 | 24.37 | 24.51 | 33.34 |
| Paramanu-Tamil-instruct 208M (ours) | **30.22** | 56.00 | **26.95** | 26.04 | 34.80 |
| Bloom 560M | 27.22 | 55.80 | 23.95 | 25.57 | 33.13 |
| Bloomz 560M (instruction-tuned) | 23.55 | 58.60 | 25.78 | 25.30 | 33.30 |
| Bloom 1.1B | 25.77 | 57.00 | 24.67 | 24.34 | 32.94 |
| Sarvam 2B | 27.44 | **63.00** | 24.06 | **26.53** | **35.25** |

Table 7: Zero-shot evaluation of LLMs (<=2B) in Tamil script models. All benchmarks report Accuracy except for ARC which reports Normalized Accuracy. Max scores are in bold.

~~Native speakers assessed the outputs on Grammar, Coherence, Creativity, and Factuality, each scored from 0 (worst) to 5 (best).~~ For human evaluation, we asked 10 annotators to evaluate top-3 responses from models for each prompt on a scale of 0 (worst) to 5 (best). We report the average score of all ratings. We also have reported normalised scores of ratings in Table 19 in appendix to handle inconsistencies among annotators. We reached inter-annotator kappa score of 0.85 for Bengali, 0.79 for Hindi, and 0.72 for Sanskrit. Figure 4 in the Appendix shows the bar chart for inter-annotator agreement's Fleiss Kappa score. For Factuality, higher scores indicated better alignment with real events, with some evaluators assigning a score of 0 when premises could not be verified.

Table 24 displays the human evaluation of Paramanu-Bangla 87.25M model for the mentioned Bangla prompts. Paramanu-Bangla 87.25M model scored an average score of 3.5/5 on grammar, 3.325/5 on coherence, 3.225/5 on creativity, and 3.2/5 on factuality metrics across top 3 generations for each Bangla prompt. Table 10 compares the performance of Paramanu-Hindi 162M and other LLMs including multilingual Bloom which was pretrained on Indian languages. In this table, we can complete see that none of the open source LLMs have the ability to generate grammatically, coherent sentences in Hindi except the Bloom series. Our monolingual model, Paramanu-Hindi 162M has performed better by 17.25% on grammar, by 46.05% on coherence, by 62.5% on creativity, and by 238.5% on factuality compared to Bloom 3B model despite being 19 times smaller in size. Table 11 in Appendix compares the performance of mParamanu-162M and other LLMs including multilingual Bloom which was pretrained on Indian languages. We can see that none of the LLMs have the ability to generate grammatically, coherent sentences in Sanskrit keeping aside the factuality. Our multilingual model, mParamanu-162M has scored the highest among all on grammar (3.75/5), coherence (3.166/5), creativity (2.166/5), and factuality (1.75/5) whereas Bloom 3B scored 0.166/5 on grammar, 0.0833/5 on coherence, and 0/5 for both creativity and factuality metrics respectively. GPT-3.5-Turbo (ChatGPT) has scored very poorly 0.25/5 on grammar & coherence metrics, 0.1818/5 in creativity and 0.33/5 on factuality metrics respectively for Sanskrit text generation. mParamanu-162M is smaller by 44.25 times compared to 7B LLaMa-2 model and yet it has shown its high quality text generation in Sanskrit as compared to ChatGPT, LLaMa, and Bloom series of models.

## 4    CONCLUSIONS AND FUTURE WORK

In this paper, we present a series of novel efficient small monolingual, bilingual, and multilingual pretrained auto regressive models the range of 13.5M to 367.5M for 10 Indian languages across 5 scripts excluding English, thereby enabling GenAI NLP technology accessible to billion speakers in the world and working towards including underrepresented languages to reduce the language bias and divide in today's NLP technology. We proposed a RoPE embedding scaling method that enables us to pretrain language models from scratch at larger sequence length context without increased GPU memory. We proposed novel tokenization technique of combining both BPE and Unigram tokenizers into a single tokenizer. We also proposed and performed language specific tokenization for

| Models | Belebele-Telugu | XStoryCloze-Telugu | MMLU-Telugu | ARC-Telugu | Average (Telugu) |
|---|---|---|---|---|---|
| Paramanu-Telugu 208M (ours) | 26.00 | 51.42 | 25.12 | **26.32** | 32.22 |
| Paramanu-Telugu-instruct 208M (ours) | **27.50** | 58.00 | **26.75** | 25.75 | **34.50** |
| Bloom 560M | 23.55 | 55.65 | 24.10 | 23.85 | 31.78 |
| Bloomz 560M (instruction-tuned) | 22.44 | 54.86 | 26.82 | 24.91 | 32.25 |
| Bloom 1.1B | 26.88 | 56.38 | 24.53 | 24.38 | 33.04 |
| Sarvam 2B | **27.66** | **60.09** | 24.67 | 25.78 | **34.55** |

Table 8: Zero-shot evaluation of LLMs (<=2B) in Telugu script models. All benchmarks report Accuracy except for ARC which reports Normalized Accuracy. Max scores are in bold.

| Model | Grammar | Coherence | Creativity | Factuality |
|---|---|---|---|---|
| GPT2-XL | 0.45833 | 0.37500 | 0.37500 | 0.37500 |
| GPT-Neo 1.3B | 0.91666 | 0.91666 | 0.91666 | 0.91666 |
| OPT 6.7B | 0.70833 | 0.70833 | 0.70833 | 0.70833 |
| GPT-J 6B | 1.12500 | 0.95833 | 0.95833 | 0.95833 |
| LLaMa 2 7B | 0.70833 | 0.70833 | 0.70833 | 0.70833 |
| Bloom 560M | 1.70833 | 1.41666 | 1.41666 | 1.37500 |
| Bloom 1.1B | 1.33333 | 1.29166 | 1.29166 | 1.29166 |
| Bloom 3B | 1.54166 | 1.29166 | 1.33333 | 1.33333 |
| Bloom 7.1B | 1.75000 | 1.16666 | 1.16666 | 1.08333 |
| GPT-3.5-Turbo | 0.58330 | 0.75000 | 0.58330 | 0.58330 |
| Paramanu-Bangla 108.5M (ours) | **4.66666** | **4.58333** | **3.76280** | **3.45833** |

Table 9: Human evaluation results (average scores of top-3 generations per prompt) of open-end text generation of Paramanu-Bangla v/s LLMs for 4 Bangla prompts on Grammar, Coherency, Creativity, and Factuality metrics. Scale is 0 (worst) to 5 (best). GPT-3.5-Turbo was accessed in October 2023.

multilingual language modeling and domain specific tokenization for monolingual language modeling for strong performance. Our multilingual tokenizer, mBharat shows the best fertility scores among Indian language tokenizers. We considered typological grouping and pretraining on comparable size of each monolingual language corpus for our multilingual mParamanu 162M to handle data imbalance and curse of multilinguality. We evaluated our models for open-end text generation with human evaluators on grammar, coherency, creativity, and factual metrics. We reached inter-annotator kappa score of 0.85 for Bengali, 0.79 for Hindi, and 0.72 for Sanskrit. In our evaluation, we found that none of the popular existing LLMs can generate grammatically correct and coherent sentences in 10 Indian languages despite being pretrained on Indian language corpora. Our efficient generative language models have performed better than Bloom 7B, LLaMa-2 7B, OPT 6.7B, GPT-J 6B, GPTNeo 1.3B, GPT2-XL models for open-end text generation in Assamese, Bangla, Hindi, Odia, and Sanskrit despite being 66 times to 20 times smaller in size. We also evaluated our models and compared with several multilingual LLMs across various NLU, NLI, and commonsense reasoning benchmarks. Our models outperformed most multilingual LLMs of size 2B and performed very competitive or even better than LLMs of size 7B on various LLM benchmarks despite being smaller in size by by multiple order of magnitude compared with LLMs whose size is bigger by multiple order of magnitude. We observed language transfer phenomena from low-resource to high resource languages of same script and typology. We also instruction-tuned our pretrained models for Bangla, Hindi, Marathi, Tamil, and Telugu and show their task handling capabilities.

In future, we would like to extend our multilingual model to 22 official Indian languages and align our generative language models with multimodal encoders to develop multimodal generative language models for Indian languages.

| Model | Grammar | Coherence | Creativity | Factuality |
|---|---|---|---|---|
| GPT2-XL | 0.0000 | 0.0000 | 0.0000 | 0.0000 |
| GPT-Neo 1.3B | 0.0000 | 0.0000 | 0.0000 | 0.0000 |
| OPT 6.7B | 0.5833 | 0.1667 | 0.1667 | 0.0000 |
| GPT-J 6B | 0.5833 | 0.2500 | 0.0000 | 0.0000 |
| LLaMa 2 7B | 1.3333 | 0.3333 | 0.5000 | 0.2083 |
| Bloom 560M | 2.7917 | 2.4583 | 1.0000 | 1.1667 |
| Bloom 1.1B | 3.2917 | 2.7917 | 1.6250 | 1.3333 |
| Bloom 3B | 4.0833 | 3.1666 | 2.0000 | 1.0833 |
| Bloom 7.1B | 3.2917 | 2.7917 | 1.6250 | 1.3333 |
| Paramanu-Hindi 162M (ours) | **4.7917** | **4.6250** | **3.2500** | **3.6666** |

Table 10: Human Evaluation results (avg scores of top 3 generations per prompt) of open-end text generation of Paramanu-Hindi 162M v/s LLMs for 4 Hindi prompts on Grammar, Coherency, Creativity, and Factuality metrics. Scale is 0 (worst) to 5 (best)

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

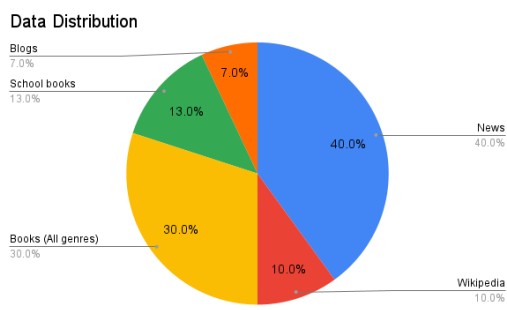

Figure 3: Pretraining Data distribution.

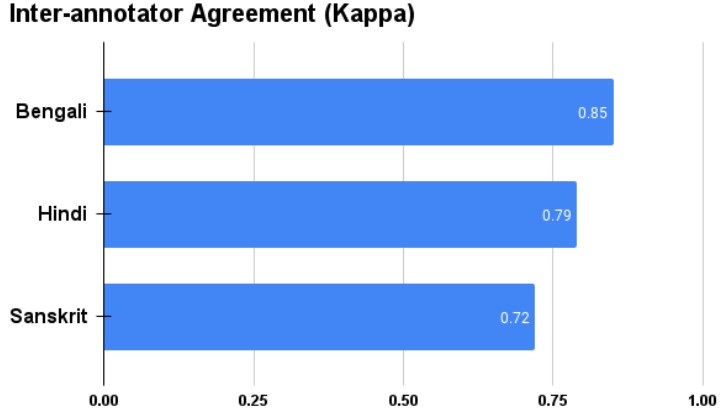

Figure 4: Inter-Annotator Agreement (Kappa)

## A   APPENDIX

| Model | Grammar | Coherence | Creativity | Factuality |
|---|---|---|---|---|
| GPT2-XL | 0.17 | 0.08 | 0.00 | 0.00 |
| GPT-Neo 1.3B | 0.25 | 0.00 | 0.00 | 0.00 |
| OPT 6.7B | 0.00 | 0.00 | 0.00 | 0.00 |
| GPT-J 6B | 0.33 | 0.33 | 0.00 | 0.00 |
| LLaMa 2 7B | 0.42 | 0.33 | 0.42 | 0.00 |
| Bloom 560M | 0.08 | 0.00 | 0.08 | 0.00 |
| Bloom 1.1B | 0.08 | 0.00 | 0.00 | 0.00 |
| Bloom 3B | 0.17 | 0.08 | 0.00 | 0.00 |
| GPT-3.5-Turbo | 0.25 | 0.25 | 0.18 | 0.33 |
| mParamanu 162M (ours) | **3.75** | **3.17** | **2.17** | **1.75** |

Table 11: Human evaluation results (average scores for top-3 generations per prompt) of open-end text generation of mParamanu v/s LLMs for 4 Sanskrit prompts on various metrics. Scale is from 0 (worst) to 5 (best). GPT-3.5-Turbo was accessed in October 2023.

### A.1   MODEL PARAMETERS

### A.2   LANGUAGE DEMOGRAPHICS

### A.3   TRAINING

Following (Hoffmann et al., 2022b), we set $lr$ decay steps to $max\_steps$ and the minimum $lr$ is set nearly to $0.1 \cdot lr$. The $lr$ schedule starts with a linear warm-up from 0 to the maximum $lr$ at 1000

| Models | MMLU-Bangla | ARC-Bangla | Belebele-Bangla | Average (Bangla) | Belebele-Assamese |
|---|---|---|---|---|---|
| Paramanu-Bangla 108M (ours) | 23.82 | 25.75 | 25.11 | 24.89 | 25.33 |
| Paramanu-Bangla-instruct 108M (ours) | 27.60 | 28.50 | 32.45 | 29.52 | 30.54 |
| mParamanu 162M (ours) | 25.29 | 20.19 | 27.44 | 24.31 | 29.00 |
| Bloom 7B | 27.10 | 26.09 | 23.22 | 25.47 | 23.11 |
| Bloomz 7B (instruction-tuned) | 32.46 | 27.20 | 53.67 | 37.77 | 48.00 |

Table 12: Zero-shot evaluation of LLMs (> 2B) across translated benchmarks of MMLU, HellaSwag, ARC datasets, and Belebele in Bengali script. All benchmarks report Accuracy except for ARC which reports Normalized Accuracy.

| Models | MMLU-Marathi | ARC-Marathi | Belebele-Marathi | Average (Marathi) |
|---|---|---|---|---|
| mParamanu 162M (ours) | 25.68 | 22.16 | 28.00 | 25.28 |
| Paramanu-Hindi 367M (ours) | 23.78 | 24.16 | 24.66 | 24.20 |
| Paramanu-Hindi-instruct 367M (ours) | **28.72** | **27.85** | 32.00 | 29.52 |
| Paramanu-Marathi 208M (ours) | 25.39 | 26.49 | 27.33 | 26.40 |
| Paramanu-Sanskrit 139M (ours) | 24.96 | 26.49 | 24.33 | 25.26 |
| Bloom 7B | 27.30 | 25.54 | 24.00 | 25.61 |
| Bloomz 7B (instruction-tuned) | 32.62 | 27.44 | 53.00 | 37.68 |
| OpenHathi 7B | 26.09 | 24.24 | 25.88 | 25.40 |
| Airavata 7B (instruction-tuned) | 26.15 | 23.90 | 29.89 | 26.64 |

Table 13: Zero-shot evaluation of LLMs (>2B) for cross-lingual language transfer in Marathi. All benchmarks report Accuracy except for ARC which reports Normalized Accuracy.

steps, followed by a cosine decay to the minimum $lr$ until the end of an epoch of training. We used the following equation for $lr$ decay ratio.

$$lr_{decay\_ratio} = \frac{t - warmup_{steps}}{lr_{decay\_steps} - warmup_{steps}}$$

where $t$ is the current training step.

### A.3.1    26.58M PARAMANU-ASSAMESE MODEL

We used the same training procedure as mentioned in 3.5 but with a batch size of 64, gradient accumulation steps of 4, and the maximum sequence length set to 1024, i.e., 262,144 tokens per iteration and transferred the learned hyperparameters from 15M model to 42M model using ($\mu$P) transfer. We set maximum learning rate ($lr$) to 3e-3 (max), weight decay to 1e-1. We trained our bigger models with fused $AdamW$ optimizer for an epoch of training with $\beta_1$=0.9, $\beta_2$=0.95, dropout of 0.0, and scaled the gradient norms using a maximum norm clipping value of 1.0, and weight decay of 0.1. For our experiments and modeling, we implemented our code using Pytorch 2.0, in-house optmized CUDA kernels and used `torch.compile` feature for every model. To further speedup training, we used BF16 mixed precision training.

### A.3.2    87.25M PARAMANU-BANGLA MODEL

We used the same training procedure as mentioned in 3.5 but with a batch size of 32, gradient accumulation steps of 8, and the maximum sequence length set to 1024, i.e., 262,144 tokens per iteration and transferred the learned hyperparameters from 15M model to 110M model using ($\mu P$) transfer.

### A.3.3    108.5M PARAMANU-BANGLA MODEL

We used the same training procedure as mentioned in 3.5 but with a batch size of 32, gradient accumulation steps of 8, and the maximum sequence length set to 1024, i.e., 262,144 tokens per iteration and transferred the learned hyperparameters from 15M model to 140M model using ($\mu P$) transfer.

### A.3.4    162M PARAMANU-HINDI MODEL

We used the same training procedure as mentioned in 3.5 but with a batch size of 32, gradient accumulation steps of 8, the maximum sequence length set to 1024, i.e., 262,144 tokens per iteration and transferred the learned hyperparameters from 15M model to 162M model using ($\mu P$) transfer.

| Models | MMLU-Hindi | HellaSwag-Hindi | ARC-Hindi | XStoryCloze-Hindi | XNLI-Hindi | Belebele-Hindi | Average (Hindi) |
|---|---|---|---|---|---|---|---|
| mParamanu 162M (ours) | 24.84 | 24.87 | 22.35 | 49.24 | 33.70 | 25.44 | 30.07 |
| Paramanu-Hindi 367M (ours) | 24.38 | 24.83 | 27.05 | 47.92 | 32.00 | 23.33 | 29.92 |
| Paramanu-Hindi-instruct 367M (ours) | 30.25 | 29.42 | 30.23 | 58.00 | 40.25 | 42.78 | 40.14 |
| Paramanu-Marathi 208M (ours) | 25.49 | 26.59 | 23.97 | 48.71 | 33.73 | 27.33 | 30.97 |
| Paramanu-Sanskrit 139M (ours) | 25.16 | 25.64 | 25.17 | 50.23 | 34.46 | 25.66 | 31.05 |
| Bloom 7B | 27.04 | 31.39 | 26.36 | 60.55 | 47.18 | 23.00 | 35.92 |
| Bloomz 7B (instruction-tuned) | 35.55 | 28.57 | 29.36 | 57.71 | 40.52 | 53.11 | 40.80 |
| OpenHathi 7B | 27.69 | 30.54 | 25.51 | 57.04 | 39.03 | 32.66 | 35.41 |
| Airavata 7B (instruction-tuned) | 30.43 | 29.53 | 25.60 | 55.59 | 39.04 | 41.44 | 36.93 |

Table 14: Zero-shot evaluation of LLMs (>2B) for cross-lingual language transfer in Hindi. All benchmarks report Accuracy except for ARC which reports Normalized Accuracy.

| Models | Belebele-Tamil | XCOPA-Tamil | MMLU-Tamil | ARC-Tamil | Average (Tamil) |
|---|---|---|---|---|---|
| Paramanu-Tamil 208M (ours) | 26.88 | 57.60 | 24.37 | 24.51 | 33.34 |
| Paramanu-Tamil-instruct 208M (ours) | 30.22 | 56.00 | 26.95 | 26.04 | 34.80 |
| Bloom 7B | 25.55 | 59.20 | 26.39 | 24.69 | 33.95 |
| Bloomz 7B (instruction-tuned) | 50.66 | 57.40 | 29.48 | 28.10 | 41.41 |

Table 15: Zero-shot evaluation of LLMs (>2B) in Tamil script models. All benchmarks report Accuracy except for ARC which reports Normalized Accuracy.

### A.3.5   367.5M PARAMANU-HINDI MODEL

We used the same training procedure as mentioned in 3.5 but with a batch size of 32, gradient accumulation steps of 16, the maximum sequence length set to 1024, i.e., 524,288 tokens per iteration and transferred the learned hyperparameters from 15M model to 367.5M model using ($\mu$P) transfer. After 1 epoch of training, the average validation perplexity is 11.05240 whereas the average training perplexity is 10.99616.

### A.3.6   87M PARAMANU-ODIA MODEL

We used the same training procedure as mentioned in 3.5 but with a batch size of 64, gradient accumulation steps of 8, and the maximum sequence length set to 1024, i.e., 524,288 tokens per iteration and transferred the learned hyperparameters from 15M model to 110M model using ($\mu P$) transfer.

### A.3.7   139.3M PARAMANU-SANSKRIT MODEL

We used the same training procedure as mentioned in 3.5 but with a batch size of 64, gradient accumulation steps of 8, and the maximum sequence length set to 1024, i.e, 524,288 tokens per iteration and transferred the learned hyperparameters from 15M model to 175M model using ($\mu P$) transfer.

### A.3.8   13.5M BILINGUAL KONKANI-MAITHILI GPT MODEL

Both Konkani and Maithili languages are typologically similar (Devanagari script). For bilingual model, we trained two 27M models with language agnostic tokenization and language specific tokenization to study the impact of language specific tokenization against language agnostic tokenization for multilingual language modeling. For training with the language specific tokenization, where we basically trained independent BPE tokenizer on Konkani and Maithili corpora with a tokenizer size of 1000 and 750 respectively and merged them together. For language agnostic tokenization, we trained BPE tokenizer on the merged corpora of Konkani and Maithili with a tokenizer size of 1750.

We used the same training procedure as mentioned in 3.5 but with a batch size of 128, gradient accumulation steps of 2, and the maximum sequence length set to 1024, i.e., 262,144 tokens per iteration . We transferred the learned hyperparameters from 15M model to 27M model using ($\mu$P) transfer.

### A.3.9   92.63M MULTILINGUAL **MPARAMANU MODEL**

For our multilingual mParamanu model, we selected the languages on the basis of typological grouping and having comparable corpora as shown in the Table 19, to avoid pretraining our multilingual model on disproportionate corpora of multiple languages. Thus, we avoid adding the Hindi corpus

| Models | Belebele-Telugu | XStoryCloze-Telugu | MMLU-Telugu | ARC-Telugu | Average (Telugu) |
|---|---|---|---|---|---|
| Paramanu-Telugu 208M (ours) | 26.00 | 51.42 | 25.12 | 26.32 | 32.22 |
| Paramanu-Telugu-instruct 208M (ours) | 27.50 | 58.00 | 26.75 | 25.75 | 34.50 |
| Bloom 7B | 24.66 | 57.37 | 26.62 | 24.47 | 33.28 |
| Bloomz 7B (instruction-tuned) | 43.11 | 58.23 | 29.55 | 27.98 | 39.71 |

Table 16: Zero-shot evaluation of LLMs (> 2B) in Telugu script models. All benchmarks report Accuracy except for ARC which reports Normalized Accuracy.

Test input indic multilingual text in 7 languages: ভাৰত মোৰ মাতৃভূমি ভারত আমার মাতৃভূমি সাডা দেশ भारत है भारत हो म्हजो देश भारत हमर देश अछि ଭାରତ ହେଉଛି ମୋର ମାତୃଭୂମି अस्माकं देशः भारतवर्षः

Tokenized by LLaMA-2 32k tokenizer:['_', '<0xE0>', '<0xA6>', '<0xAD>', '01', '<0xE0>', '<0xA7>', '<0xB0>', 'ত', '_', 'ম', 'ো1', '<0xE0>', '<0xA7>', '<0xB0>', '_', 'ম', '01', 'ত', '<0xE0>', '<0xA7>', '<0x83>', '<0xE0>', '<0xA6>', '<0xAD>', '<0xE0>', '<0xA7>', '<0x82>', 'ম', 'ি', '_', '<0xE0>', '<0xA6>', '<0xAD>', '01', 'ৰ', 'ত', '_', '<0xE0>', '<0xA6>', '<0x86>', 'ম', '01', 'ৰ', '_', 'ম', '01', 'ত', '<0xE0>', '<0xA7>', '<0x83>', '<0xE0>', '<0xA6>', '<0xAD>', '<0xE0>', '<0xA7>', '<0x82>', 'ম', 'ি', '_', 'স', '01', 'ড', '01', '_', 'দ', 'ে', 'শ', '_', 'भ', '01', 'र', 'त', '_', 'ह', 'ै', '_', 'भ', '01', 'र', 'त', '_', 'ह', '01', '_', 'म', '्', 'ह', 'ज', '01', '_', 'द', 'े', 'श', '_', 'भ', '01', 'र', 'त', '_', 'ह', 'म', 'र', '_', 'द', 'े', 'श', '_', 'अ', '<0xE0>', '<0xA4>', '<0x9B>', 'ि', '_', '<0xE0>', '<0xAC>', '<0xAD>', '<0xE0>', '<0xAC>', '<0xBE>', '<0xE0>', '<0xAC>', '<0xB0>', '<0xE0>', '<0xAC>', '<0xA4>', '_', '<0xE0>', '<0xAC>', '<0xB9>', '<0xE0>', '<0xAD>', '<0x87>', '<0xE0>', '<0xAC>', '<0x89>', '<0xE0>', '<0xAC>', '<0x9B>', '<0xE0>', '<0xAC>', '<0xBF>', '_', '<0xE0>', '<0xAC>', '<0xAE>', '<0xE0>', '<0xAD>', '<0x8B>', '<0xE0>', '<0xAC>', '<0xB0>', '_', '<0xE0>', '<0xAC>', '<0xAE>', '<0xE0>', '<0xAC>', '<0xBE>', '<0xE0>', '<0xAC>', '<0xA4>', '<0xE0>', '<0xAD>', '<0x83>', '<0xE0>', '<0xAC>', '<0xAD>', '<0xE0>', '<0xAD>', '<0x82>', '<0xE0>', '<0xAC>', '<0xAE>', '<0xE0>', '<0xAC>', '<0xBF>', '_', 'अ', 'स', '्', 'म', '01', 'क', 'ं', '_', 'द', 'े', 'श', '<0xE0>', '<0xA4>', '<0x83>', '_', 'भ', '01', 'र', 'त', 'व', 'र', '्', 'ष', '<0xE0>', '<0xA4>', '<0x83>']

Tokenized by mBharat tokenizer:['_ভাৰত', '_মোৰ', '_মাতৃ', 'ভূ', 'মি', '_ভারত', '_আমার', '_মাতৃ', 'ভূ', 'মি', '_সা', 'ডা', '_দেশ', '_भारत', '_है', '_भारत', '_हो', '_म्ह', 'जो', '_देश', '_भारत', '_हम', 'र', '_देश', '_अछि', '_ଭାରତ', '_ହେଉଛି', '_ମୋର', '_ମା', 'ତୃ', '_', 'ভূমি', '_अ', 'स्मा', 'कं', '_देश', 'ः', '_भारत', 'वर्ष', 'ः']

Figure 5: mBharat Tokenizer v/s LLaMa-2 tokenizer for input in 7 Indian languages.

in the multilingual pretraining. We grouped Bodo, Dogri, Konkani, Maithili, and Sanskrit language corpora in the same group as they follow Devanagari script, Assamese and Bangla (Bengali) were grouped together as they follow Assamese-Bengali script and Odia was kept as independent as it is typologically different than the Devanagari and Assamese-Bengali script during tokenization and pretraining. The multilingual model mParamanu was trained on the clean preprocessed pretraining data of Assamese, Bangla (Vacaspati), Bodo-Dogri (merged), Konkani, Maithili, Odia, and Sanskrit corpora as shown in Table 19.

We used the same training procedure as mentioned in 3.5 but with a batch size of 32, gradient accumulation steps of 8, maximum sequence length set to 1024, i.e., 524,288 tokens per iteration and a language specific multilingual tokenizer (mBharat).

### A.3.10   162M MULTILINGUAL mPARAMANU MODEL

We used the same training procedure as mentioned in 3.5 but with a batch size of 32, gradient accumulation steps of 8, the maximum sequence length set to 1024, i.e., 262,144 tokens per iteration. We used language specific tokenization for the multilingual mBharat-162M model. It took 4 days and 11 hours to get trained on single A100-40G GPU with the mentioned specifications.

### A.4   DATA CLEANING METHODOLOGY

1. For languages in Devanagari, Assamese-Bengali, and Odia script (Assamese, Bangla, Hindi, Konkani, Maithili, Odia, Sanskrit), we perform splitting of sentences by punctuation "dari"/"danda" (|)

2. Removal of non literal characters for a given language and Unicode normalization

3. White space normalization.

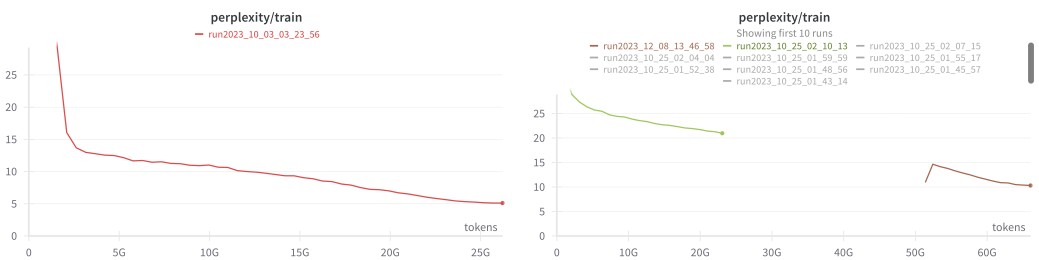

(a) Training Perplexity for Paramanu-Bangla v/s Tokens in Billion (G=billion)
(b) Training Perplexity for Paramanu-Hindi v/s Tokens in Billion (G=billion)

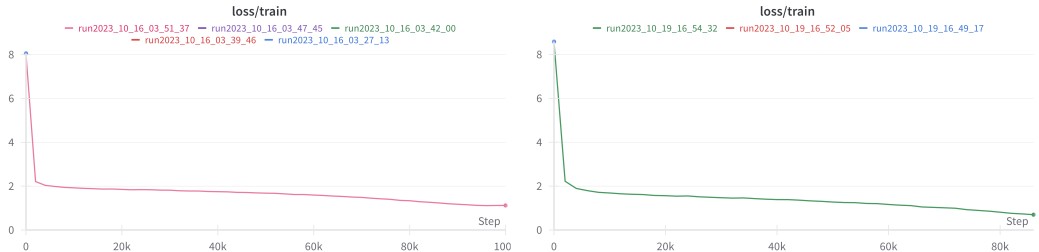

(c) Training Perplexity of Paramanu-Odia v/s training steps.
(d) Training Perplexity of Paramanu-Sanskrit v/s training steps.

Figure 6: Training Perplexity v/s Tokens and Training Perplexity v/s training steps for Paramanu pretrained models.

| n_params | d_model | n_layers | n_heads | dim_head | max_seq_length |
|---|---|---|---|---|---|
| 15M | 288 | 6 | 6 | 48 | 512 |
| 27M | 512 | 4 | 8 | 64 | 1024 |
| 42M | 512 | 8 | 8 | 64 | 1024 |
| 110M | 768 | 12 | 12 | 64 | 1024 |
| 140M | 768 | 15 | 12 | 64 | 1024 |
| 350M | 1024 | 12 | 16 | 64 | 1024 |
| 425M | 1280 | 18 | 10 | 128 | 1024 |

Table 17: Model architectures considering tokenizer size of 32000

4. Removal of English literals, Roman digits, French, German, Italian, Russian, Chinese literals and punctuation following Unicode representation using regular expressions

5. Removal of emoticons, symbols, pictographs, transport & map symbols, and iOS flags following Unicode representation using regular expression

| Language | Family | Script | #Speakers |
|---|---|---|---|
| Assamese | Indo-European | Assamese-Bengali | 24 M |
| Bangla | Indo-European | Bengali | 300 M |
| Hindi | Indo-European | Devanagari | 692 M |
| Konkani | Indo-European | Devanagari | 2 M |
| Maithili | Indo-European | Devanagari | 14 M |
| Marathi | Indo-European | Devanagari | 99 M |
| Odia | Indo-European | Odia | 43 M |
| Sanskrit | Indo-European | Devanagari | 0.025 M |
| Tamil | Indo-Dravidian | Tamil | 77 M |
| Telugu | Indo-Dravidian | Telugu | 95 M |

Table 18: Speaker estimates according to the Indian Census 2011

6. Removal of links, emails, HTML/XML tags, emojis, language specific punctuation, personal information like phone number, address, ID number using regular expression. We also deduplicated web scrapped pretraining corpora in respective languages

| Language | Corpus Source | Corpus Size | #Sentences |
|---|---|---|---|
| Assamese | Indic Corp v2 + Wikipedia + Curated books (ours) | 3.2 GB | 5,734,166 |
| Bangla | Vacasapati + Wikipedia + Curated books (ours) | 3.6 GB | 22,533,608 |
| Hindi | IITB monolingual + Wikipedia + Curated books (ours) | 15.8 GB | 52,124,643 |
| Konkani | Indic Corp v2 | 516.5 MB | 1,337,693 |
| Maithili | Indic Corp v2 | 191.3 MB | 947,386 |
| Marathi | Indic Corp v2 + Wikipedia + Curated books (ours) | 12.5 GB | 34,567,839 |
| Odia | Indic Corp v2 + Wikipedia | 6.2 GB | 14,657,392 |
| Sanskrit | Indic Corp v2 | 6.7 GB | 17,034,631 |
| Tamil | Indic Corp v2 + Wikipedia + Curated books (ours) | 10.7GB | 27,872,768 |
| Telugu | Indic Corp v2 + Wikipedia + Curated books (ours) | 13.5 GB | 40,241,847 |

Table 19: Pretraining Data details after data cleaning.

## A.5 HUMAN EVALUATION

### A.5.1 MPARAMANU-162M VS BLOOM 1.1B EVALUATION

From Table 20 we see that Bloom 1.1B model could not distinguish languages of the same script so when we prompt Bloom with Konkani, Maithili, and Sanskrit (Devanagari script), Bloom only generated incoherent, grammatically incorrect text in Hindi whereas our multilingual model, mParamanu-162M was able to recognise prompt in respective distinguished languages Konkani, Maithili, and Sanskrit (Devanagari) to generate grammatically sound and coherent text in respective languages. Similarly, Bloom 1.1B could not identify Assamese (Assamese-Bengali Script) and generated out of context Bangla words including series of Bangla alphabets for the Assamese prompt as shown in the table 20 whereas mParamanu-162M model generated coherent sentences in Assamese when prompted with Assamese prompt. However, mParamanu-162M model is 7 times smaller in size than Bloom 1.1B and has better text generation capabilities in terms of grammar, coherence metrics for languages under consideration. This makes us to ascertain that despite Bloom being pretrained on 45 languages including Indian languages yet its text generation capabilities is very limited in terms of grammatically correct and coherent text generation for Indian languages. Therefore, its not the right way to pretrain on many languages together varying script at the same time without considering linguistics typology, features, and disproportionate pretraining low quality corpus in multilingual setting.

### A.5.2 BANGLA EVALUATION

We hard prompted the pretrained models with the following Bangla prompts, prompt 1: ফে-লুদা তোপসেকে বলল, যা লালমোহনবাবুকে খবর দে! (phēludā tōpasēkē balala, yā lālamōhanabābukē khabara dē) – Feluda told Tapshe, Go and call Lalmohanbabu (from a popular detective novel series), prompt 2: এমন সময় হঠাৎ বাঘের ডাক! (ēmana samaẏa haṭhāt bāghēra ḍāka!) – suddenly there was a tiger roar (astonishing fearful expression), prompt 3: অপু এসে ডাকল, মা! (apu ēsē ḍākala, mā!) – Apu came and called, Mother (Apu is a character in a popular novel, also adapted into a popular movie), prompt 4: সে খুব খুশী হলো। (sē khuba khuśī halō) – she became very happy (generic statement). All these prompts are related either to common expression in Bangla or covers diverse Bangla literature and cultural context in Bangla speaking community in West Bengal, India.

Table 24 displays the human evaluation of Paramanu-Bangla 87.25M model for the mentioned Bangla prompts. We see Paramanu-Bangla 87.25M model scored an average score of 3.5/5 on grammar, 3.325/5 on coherence, 3.225/5 on creativity, and 3.2/5 on factuality metrics across top 3 generations for each Bangla prompt. Table 9 compares the performance of Paramanu-Bangla 108.5M and other LLMs including multilingual Bloom which was pretrained 45 languages including Indian languages. In this table, we can see that none of the open source LLMs have the ability

| Prompt | mParamanu-162M | Bloom 1.1B |
|---|---|---|
| মোৰ দেশ ভাৰত
(Assamese)
(ōra dēśa bhārata) | মোৰ দেশ ভাৰত এখন সাংস্কৃতিক কেন্দ্ৰ
(mōra dēśa bhārata ēkhana sāṁskṛtika kēndra)
মোৰ দেশ ভাৰত বিশ্বৰ বিভিন্ন ঠাইৰ সংস্কৃতিকলৈয়ে গঢ় লৈ উঠিছে
(mōra dēśa bhārata biśbara bibhinna ṭhāira saṁskṛtikalaiẏē gaṛha lai uṭhichē) | মোৰ দেশ ভাৰত (সোসাইটি বাংলাদেশ)।তথ্যসূত্ৰ বহিঃসংযোগ বিষয়শ্ৰেণী:বাংলাদেশের দৰ্শনীয় স্থান বিষয়শ্ৰেণী:বাংলাদেশের প্ৰাচীন স্থাপত্য বিষয়শ্ৰেণী:বাংলাদেশের স্থাপত্য ঐতিহ্য
(mōra dēśa bhārata (sōsāiṭi bāṁlādēśa)'.tathyasūtra bahiḥsaṁyōga biṣaẏaśrēṇī:bāṁlādēśēra darśanīẏa sthāna biṣaẏaśrēṇī:bāṁlādēśēra prācīna sthāpatya biṣaẏaśrēṇī:bāṁlādēśēra sthāpatya aitihya)
মোৰ দেশ ভাৰতত (ক) খ) গ) ঘ) চ) ম) ক) খ) গ) ঘ) চ) ম) ক) খ) গ) ঘ) চ) ম) ক) খ) গ) ঘ) চ) ম) ক) খ) গ) ঘ) ▨) চ) ম) ক) খ) গ) ঘ) ▨) চ
(mōra dēśa bhāratata (ka) kha) ga) gha) ◆) ca) ma) ka) kha) ga) gha) ◆) ca) ma) ka) kha) ga) gha) ◆) ca) ma) ka) kha) ga) gha) ◆) ca) ma) ka) kha) ga) gha) ◆) ca) ma) ka) kha) ga) gha) ◆) ca) |
| আমার দেশ ভারত
(Bengali)
āmāra dēśa bhārata | আমার দেশ ভারতবর্ষ, আমাদের দেশআমার দেশ ভারতবর্ষ হিন্দুস্থান
(āmāra dēśa bhāratabarṣa, āmādēra dēśaāmāra dēśa bhāratabarṣa hindusthāna)
আমার দেশ ভারতবর্ষের বিরাট অংশ এক সময় বঙ্গদেশের মত ছিল
(āmāra dēśa bhāratabarṣēra birāṭa aṁśa ēka samaẏa baṅgadēśēra mata chila) | আমার দেশ ভারত, বাংলাদেশ, পাকিস্তান, শ্রীলঙ্কা, মায়ানমার, থা-ইল্যান্ড, ইন্দোনেশিয়া, মায়ানমার, শ্রীলঙ্কা, দক্ষিণ কোরিয়া, জাপান, চীন, জার্মানি, থাইল্যান্ড, চীন, ভারত, ইন্দোনেশিয়া, মায়ানমার, শ্রীল-ঙ্কা, মায়ানমার, থাইল্যান্ড, ইন্দোনেশিয়া, মায়ানমার, শ্রীলঙ্কা, দক্ষিণ কোরিয়া, জাপান, চীন, জার্মানি, থাইল্যান্ড, চীন, ভারত, ইন্দোনেশি-য়া, মায়ানমার, শ্রীলঙ্কা, মায়ানমার, থাইল্যান্ড
(āmāra dēśa bhārata, bāṁlādēśa, pākistāna, śrīlaṅkā, māẏānamāra, thāilyāṇḍa, indōnēśiẏā, māẏānamāra, śrīlaṅkā, dakṣiṇa kōriẏā, jāpāna, cīna, jārmāni, thāilyāṇḍa, cīna, bhārata, indōnēśiẏā, māẏānamāra, śrīlaṅkā, māẏānamāra, thāilyāṇḍa, indōnēśiẏā, māẏānamāra, śrīlaṅkā, dakṣiṇa kōriẏā, jāpāna, cīna, jārmāni, thāilyāṇḍa, cīna, bhārata, indōnēśiẏā, māẏānamāra, śrīlaṅkā, māẏānamāra, thāilyāṇḍa)
আমার দেশ ভারত ও মুসলিম জাতির ঐক্য ও কল্যাণ কামনার সঙ্গে একাত্মতা পোষণ করার মাধ্যমে এ দেশ ও মুসলিম জাতির কল্যাণ কামনা করছি।
(āmāra dēśa bhārata ō musalima jātira aikya ō kalyāṇa kāmanāra saṅgē ēkātmatā pōṣaṇa karāra mādhyamē ē dēśa ō musalima jātira kalyāṇa kāmanā karachi.) |
| भारत हमर देश अछि
(Maithili)
(bhārata hamara dēśa achi) | भारत हो म्हजो देश हो मोलांत
(bhārata hō mhajō dēśa hō mōlāṁta)
भारत हमर देश अछि और ओ देशक सर्वोच्च पदसँ ओतप्रोत अछि
(bhārata hamara dēśa achi aura ō dēśaka sarvōcca padasaṁ ōtaprōta achi) | भारत हमर देश अछि, काहे अछि ?''हे राम !''हे राम !''हे राम !''हे राम !''हे राम !
(bhārata hamara dēśa achi, kāhē achi ?``hē rāma !''hē rāma !``hē rāma !``hē rāma !``hē rāma!)
(generating Hindi) भारत हमर देश अछि। मिथुन राशि के जातक के लिए आज का दिन काफी अच्छा है। आज की दिन में नौकरी के लिए कोई परेशानी नहीं रहेगी, साथ ही आज के दिन आपके लिए कुछ अच्छा काम मिल सकता है।
(bhārata hamara dēśa achi. mithuna rāśi kē jātaka kē liē āja kā dina kāphī acchā hai. āja kī dina mēṁ naukarī kē liē kōī parēśānī nahīṁ rahēgī, sātha hī āja kē dina āpakē liē kucha acchā kāma mila sakatā hai.)
(generating Hindi) |
| ଭାରତ ହେଉଛି ମୋର ମାତୃଭୂମି
(Odia) | ଭାରତ ହେଉଛି ମୋର ମାତୃଭୂମି ଜାତିର ଏକତ୍ରୀକରଣ ହୋଇଛି
(bhārata hēuchi mōra mātṛbhūmi jātira ēkatrīkaraṇa hōichi)
ଭାରତ ହେଉଛି ମୋର ମାତୃଭୂମି ଓ ମୋ ନିଜସ୍ୱତା (bhārata hēuchi mōra mātṛbhūmi ō mō nijasvatā) | ଭାରତ ହେଉଛି ମୋର ମାତୃଭୂମି। (bhārata hēuchi mōra mātṛbhūmi.)
ଭାରତ ହେଉଛି ମୋର ମାତୃଭୂମି ଓ ଦେଶର ଛଟମଟ ଶୌଚାଳୟ।
(bhārata hēuchi mōra mātṛbhūmi ō dēśara chaṭamaṭa śaucālaẏa) |
| अस्माकं देश: भारतवर्ष:
(Sanskrit)
(asmākaṁ dēśaḥ bhāratavarṣaḥ) | अस्माकं देश: भारतवर्ष: सर्वातिशायी (asmākaṁ dēśaḥ bhāratavarṣaḥ sarvātiśāyī)
अस्माकं देश: भारतवर्ष: वर्तते (asmākaṁ dēśaḥ bhāratavarṣaḥ vartatē) | अस्माकं देश: भारतवर्ष: ॥ ॥॥॥ (asmākaṁ dēśaḥ bhāratavarṣaḥ) |

Table 20: mParamanu-162M v/s Bloom 1.1B open end text generation for Indic prompts.

| Input Prompt | Text Generation |
|---|---|
| কামাখ্যা মন্দিৰ (Kamakhya Temple) | কামাখ্যা মন্দিৰত উপস্থিত হ'ব গৃহমন্ত্ৰী অমিত শ্বাহ । (kāmākhyā mandirata upasthita ha'ba gṛhamantrī amita śbāha.) |
| | কামাখ্যা মন্দিৰ দৰ্শনৰ পাছতে মুখ্যমন্ত্ৰী হিমন্ত বিশ্ব শৰ্মাই ফ্লেগ স্ব' মুকলি কৰিব । (kāmākhyā mandira darśanara pāchatē mukhyamantrī himanta biśba śarmāi phlēga śba' mukali kariba.) |
| | কামাখ্যা মন্দিৰলৈ যোৱা পথত যোৱা কোনো এজন লোককে ক'ভিড টেষ্ট কৰিবলৈ অনুৰোধ আৰু আহ্বান । (kāmākhyā mandiralai yōvā pathata yōvā kōnō ējana lōkakē ka'bhiḍa ṭēṣṭa karibalai anurōdha āru āhbāna.) |
| অসম চাহ (Assam Tea) | অসম চাহ মজদুৰ সংঘৰ মৰিগাঁও শাখাৰ ধৰ্মঘট । (asama cāha majadura saṁghara marigām̐ō śākhāra dharmaghaṭa.) |
| | অসম চাহ মজদুৰ সংঘৰ নিৰ্দেশ মৰ্মে শ্ৰমিকৰ মজুৰি ৩৫০টকালৈ বৃদ্ধি কৰাৰ দাবীত ১ নৱেম্বৰ পৰা ৩ নৱেম্বৰলৈ তিনিদিনীয়া কাৰ্যসূচীৰে বাগিছা সমূহৰ কাৰ্যবাহী পৰিচালক মজলিলাত চাহ শ্ৰমিকৰ বিভিন্ন দাবীত চতুৰ্থ বৰ্গৰ কৰ্মচাৰীৰ এক বিশাল গণ ধৰ্মঘট কাৰ্যসূচী ৰূপায়ণ কৰা হয় । (asama cāha majadura saṁghara nirdēśa marmē śramikara majuri 350ṭakālai bṛddhi karāra dābīta 1 navēmbara parā 3 navēmbaralai tinidinīyā kāryasūcīrē bāgichā samūhara kāryabāhī paricālaka majalilāta cāha śramikara bibhinna dābīta caturtha bargara karmacārīra ēka biśāla gaṇa dharmaghaṭa kāryasūcī rūpāyaṇa karā haya.) |
| | অসম চাহ মজদুৰ সংঘৰ যোৰহাট শাখাৰ সভাপতি ৰাজেন গোঁহাই আৰু সম্পাদক সৌৰভ গায়নে কয় যে পৰৱৰ্তী সময়ত উপযুক্ত ন্যায় ব্যৱস্থাৰ ক্ষেত্ৰত অসম চৰকাৰৰ মিনিষ্ট্ৰী অব বেংক লট অৱ অসম আৰু অসম আৰাম গৃহৰ ব্যৱস্থাৰ বাবে বহু লোকক অসম প্ৰেমী দৰিদ্ৰ ৰজাঘৰীয়া আৰ্থিক সাহাৰ্য্য প্ৰদান কৰি আহিছে । (asama cāha majadura saṁghara yōrahāṭa śākhāra sabhāpati rājēna gōm̐hāi āru sampādaka saurabha gāyanē kaya yē paravartī samayata upayukta nyāya byavasthāra kṣētrata asama carakārara minisṭrī aba bēṁka laṭa ava asama āru asama ārāma gṛhara byavasthāra bābē bahu lōkaka asama prēmī daridra rajāgharīyā ārthika sāhāryya pradāna kari āhichē.) |
| ভূপেন হাজৰিকা (Bhupen Hazarika) | ভূপেন হাজৰিকা ন্যাস গুৱাহাটীয়ে এই ন্যাসৰ বাবে প্ৰদান কৰিছে জনজাতীয় পুৰস্কাৰ । (bhūpēna hājarikā nyāsa guvāhāṭīyē ēi nyāsara bābē pradāna karichē janajātīya puraskāra.) |
| | ভূপেন হাজৰিকা হেমচন্দ্ৰ বৰুৱা অসমীয়া বিনোদন শাস্ত্ৰৰ পিতৃ স্বৰ্গীয় পিতাপত্ৰৰ স্বত্বাধিকাৰী হেমচন্দ্ৰ গোস্বামী আছিল অসমীয়া ভাষা সাহিত্যৰ এগৰাকী বিশিষ্ট প্ৰতিভাশালী ব্যক্তি । (bhūpēna hājarikā hēmacandra baruvā asamīyā binōdana śāstrara pitṛ sbargīya pitāpatrara sbatbādhikārī hēmacandra gōsbāmī āchila asamīyā bhāṣā sāhityara ēgarākī biśiṣṭa pratibhāśālī byakti.) |
| | ভূপেন হাজৰিকা সেতুঘাট ৰ ওপৰত উজনি অসমত চলোৱা এক বহৎ অভিযানত বৃহৎ পৰিমাণৰ ড্ৰাগছ জব্দ কৰা হৈছে । (bhūpēna hājarikā sētughāṭa ra ōparata ujani asamata calōvā ēka bahat abhiyānata bṛhat parimāṇara ḍrāgacha jabda karā haichē.) |
| বিহু (Bihu) | বিহু অসমৰ এবিধ লোকানাট্যানুষ্ঠানৰ অন্যতম এক লোকনাট্যানুষ্ঠান । (bihu asamara ēbidha lōkānāṭyānuṣṭhānara anyatama ēka lōkānāṭyānuṣṭhāna.) |
| | বিহু উদযাপনৰ জৰিয়তেও মাক দেউতাকৰ উপস্থিতি শিশুৰ বেলিকাও স্বচ্ছলতা লাভ কৰিব পৰা যায় । (bihu udayāpanara jariyatēō māka dēutākara upasthitita śiśura bēlikāō sbacchalatā lābha kariba parā yāya.) |
| | বিহু চেৰে মাজুলীৰ আকাশ বতাহ মুখৰিত । (bihu cērē mājulīra ākāśa batāha mukharita.) |

Table 21: Paramanu-Assamese open end text generation for Assamese prompts capturing local context and culture.

| Input Prompt | Text Generation |
|---|---|
| ଜଗନ୍ନାଥ ମନ୍ଦିର (Jagannath Mandir) | ଜଗନ୍ନାଥ ମନ୍ଦିରକୁ ଦର୍ଶନ ପାଇଁ ଆସୁଥିବା ପିଲାମାନେ ମହାପ୍ରଭୁଙ୍କ ଆଦ୍ୟ ସେବକ ଶାସନ ଗାଦିରେ ବସିଛନ୍ତି । (jagannātha mandiraku darśana pāiṁ āsuthibā pilāmānē mahāprabhuṅka ādya sēbaka śāsana gādirē basichanti.) |
| | ଜଗନ୍ନାଥ ମନ୍ଦିରକୁ ସଂଯୋଗ କରୁଥିବା ଜଗନ୍ନାଥ ମନ୍ଦିରର ତିହ୍ୟ ସ୍ଥାନର ବିକାଶ ପାଇଁ ୫କୋଟି ଟଙ୍କା ଅନୁଦାନ ଦିଆଯାଇଥିବାବେଳେ ବସ୍ ଷ୍ଟାଣ୍ଡର ନବୀକରଣ ନିର୍ମାଣ କରାଯାଇ । (jagannātha mandiraku saṁyōga karuthibā jagannātha mandirara tihya sthānara bikāśa pāiṁ 5kōṭi ṭaṅkā anudāna diāyāithibābēḷē bas ṣṭāṇḍara nabīkaraṇa karāyāi.) |
| | ଜଗନ୍ନାଥ ମନ୍ଦିର ପରିଚାଳନା କମିଟିର ପୂର୍ବତନ ସଦସ୍ୟ ହେଉଛନ୍ତି ଅନନ୍ତ କୁମାର ଜେନା । (jagannātha mandira paricāḷanā kamiṭira pūrbatana sadasya hēuchanti ananta kumāra jēnā.) |
| ପୁରୀ ସାଗର ବେଳାଭୂମି (Puri Sea Beach) | ପୁରୀ ସାଗର ବେଳାଭୂମିରେ ଶୁଭାରମ୍ଭ ହୋଇଛି ଆନ୍ତର୍ଜାତୀୟ ଖ୍ୟାତି ସମ୍ପର୍ଣ୍ଣ ବାଲୁକା ଶିଳ୍ପୀ ସୁଦର୍ଶନ ପଟ୍ଟନାୟକଙ୍କ ଦ୍ୱାରା ବିଶ୍ୱ ବାଲୁକା କୃତ । (purī sāgara bēḷābhūmirē śubhārambha hōichi āntarjātīya khyāti samparṇṇa bālukā śilpī sudarśana paṭṭanāyakaṅka dvārā biśva bālukā kṛt.) |
| | ପୁରୀ ସାଗର ବେଳାଭୂମିର ମହୋତ୍ସବର ପ୍ରଥମ ଦିନରେ ପୁରୀ ଶ୍ରୀଜଗନ୍ନାଥ ମନ୍ଦିରରେ ଆକର୍ଷଣୀୟ ନୃତ୍ୟକଳା ଦୃଶ୍ୟ । (purī sāgara bēḷābhūmi mahōtsabara prathama dinarē purī śrījagannātha mandirarē ākarṣaṇīya nṛtyakaḷā dṛśya.) |
| | ପୁରୀ ସାଗର ବେଳାଭୂମିରେ ୩୨ ଫୁଟର ପାଣ୍ଠେରା । (purī sāgara bēḷābhūmirē 32 phuṭara pānṭhērā.) |
| ଚିଲିକା ହ୍ରଦ (Chilka Lake) | ଚିଲିକା ହ୍ରଦ ଭିତରକନିକା ପ୍ରଭୃତିରେ ବିଭିନ୍ନ ପ୍ରଜାତିର ଡଲଫିନ୍ ମିଳୁଥିଲା । (cilikā hrada bhitarakanikā prabhṛtirē bibhinna prajātira ḍalaphina miḷuthilā.) |
| | ଚିଲିକା ହ୍ରଦ ଭୂଇଁରେ ଛୁଇଁବ ସମୁଦ୍ର ଆସିଲା ସାମାନ୍ୟ ବହଳିଆ କୁଆର । (cilikā hrada bhūim̐rē chuim̐ba samudra āsilā sāmānya bahaḷiā juāra.) |
| | ଚିଲିକା ହ୍ରଦ ବ୍ରାହ୍ମଣୀ ବୈତରଣୀ ଓ ଅନ୍ୟ ଜଳଦ୍ୱୀପ ଦେଇ ଏହି ନଦୀ ପୁରୀକୁ ପ୍ରଜନନ କରେ ଓ ବ୍ରାହ୍ମଣୀ ନଦୀର ଜଳ ଯାଜପୁର ଟିକାୟତନନ୍ଦପୁରରେ ପ୍ରସ୍ତାବିତ ରାସପଥର ପୁନରୁଦ୍ଧାର କରି ଏ ଜିଲ୍ଲା ପ୍ରଗତିର ସୂତ୍ରପାତ ହୁଏ । (cilikā hrada brāhmaṇī baitaraṇī ō anya jaladvīpa dēi ēhi nadī purīku prajanana karē ō brāhmaṇī nadīra jala yājapura ṭikāyatanandapurarē prastābita rāsapathara punaruddhāra kari ē jillā pragatira sutrapāta huē.) |
| କୋଣାର୍କ ସୂର୍ଯ୍ୟ ମନ୍ଦିର (Konark Sun Temple) | କୋଣାର୍କ ସୂର୍ଯ୍ୟ ମନ୍ଦିରକୁ ବିଶ୍ୱ ତିହ୍ୟର ମାନ୍ୟତା ଦାବିରେ ମୁଖ୍ୟମନ୍ତ୍ରୀଙ୍କୁ ଚିଠି ଲେଖିଲେ କୋଣାର୍କ ବିଧାୟକ । (kōṇārka sūryya mandiraku biśva tihyara mānyatā dābirē mukhyamantrīnku ciṭhi lēkhilē kōṇārka bidhāyaka.) |
| | କୋଣାର୍କ ସୂର୍ଯ୍ୟ ମନ୍ଦିର ପାଦଦେଶରେ କୋଣାର୍କ ମହୋତ୍ସବ ଉଦ୍ ଘାଟିତ । (kōṇārka sūryya mandira pādadēśarē kōṇārka mahōtsaba udghāṭita.) |
| | କୋଣାର୍କ ସୂର୍ଯ୍ୟ ମନ୍ଦିରର ବିଭିନ୍ନ ଅବ୍ୟବସ୍ଥା ନେଇ କୋଣାର୍କ ଯାତ୍ରୀ ପ୍ରତିବାଦ କରିବା ପରେ ବିଦେଶୀ ଯାତ୍ରୀଙ୍କ ଆଗମନ ଉପରେ ପ୍ରଶାସନ ୧୪୪ ଧାରା ଲାଗୁ କରିଥିଲା । (kōṇārka sūryya mandirara bibhinna abyabasthā nēi kōṇārka yātrī pratibāda karibā parē bidēśī yātrīṅka āgamana uparē praśāsana 144 dhārā lāgu karithilā.) |

Table 22: Paramanu-Odia's text generation for 4 Odia prompts capturing local context and culture.

| Language | Input Prompt | Text Generation |
|---|---|---|
| Assamese | ভাৰত মোৰ মাতৃভূমি | মোৰ দেশ ভাৰত বিশ্বৰ বিভিন্ন ঠাইৰ সংস্কৃতিকলৈয়ে গঢ় লৈ উঠিছে।
মোৰ দেশ ভাৰত এখন সাংস্কৃতিক কেন্দ্র। |
| Bangla | ভারত আমার মাতৃভূমি | আমার দেশ ভারতবর্ষ আমাদের দেশ ।
আমার দেশ ভারতবর্ষের বিরাট অংশ এক সময় বঙ্গদেশের মত ছিল। |
| Dogri | साडा देश भारत है | साडा देश भारत है हाँ आज के समय से आने वाला उत्तर भारतीय है जहां धीरूभाई अम्बानी फॉर युवा का मुजफ्फरपुर का लोग है वहां के बावजूद बिग बॉस नाम की तरफ से भी कुछ लकी। |
| Konkani | भारत हो म्हजो देश | भारत हो म्हजो देश राष्ट्राची वाट लायल्या
भारत हो म्हजो देश भारत म्हज्या इतिहासांत देव म्हज्या इतिहासांत आस्पावता आदलें पैकूच आसा त्या इतिहासाचो आदलो काळ आशिल्लो हाका इतिहासाचो चवथो भाग म्हण्टा। |
| Maithili | भारत हमर देश अछि | भारत हमर देश अछि और देशक सर्वोच्च पदसँ ओतप्रोत अछि
भारत हो म्हजो देश हो मोलांत। |
| Odia | ଭାରତ ହେଉଛି ମୋର ମାତୃଭୂମି | ଭାରତ ହେଉଛି ମୋର ମାତୃଭୂମି ଜାତିର ଏକତ୍ରୀକରଣ ହୋଇଛି
ଭାରତ ହେଉଛି ମୋର ମାତୃଭୂମି ଓ ମୋ। ନିଜସ୍ୱର। |
| Sanskrit | अस्माकं देशः भारतवर्षः | अस्माकं देशः भारतवर्षः सर्वातिशायी।
अस्माकं देशः भारतवर्षः इति ख्यातः। |

Table 23: mParamanu 162 M pretrained model text generation

| Prompt | Text Generation (top-3) | Grammar | Consistency | Coherency | Factuality |
|---|---|---|---|---|---|
| ফেলুদা তোপসেকে বলল, যা লালমোহনবাবুকে খবর দে! | ফেলুদা তোপসেকে বলল, যা লালমোহনবাবুকে খবর দে! আমরা আলাম ঘর থেকে বেরোতেই ধূর্জটিবাবুর আর্তনাদ শুনলাম। তারপর ফেলুদা গলা পলান। ধূর্জটিবাবুর ঘরের সামনে যে লোক বসেছিল, সে ফেলুদার দিকে চেয়ে জিজ্ঞেস করল, আপনার নাম কী? ফেলুদা আর্তনাদ শুনে ফেলুদা বাইরে চলে গেল। আমি আরও মিনিটখানেক বাইরে হাঁটলাম।
ফেলুদা তোপসেকে বলল, যা লালমোহনবাবুকে খবর দে! উনি আসছেন।
ফেলুদা তোপসেকে বলল, যা লালমোহনবাবুকে খবর দে! ভদ্রলোক জিগগেস করবেন, আর সত্যি বলতে কী, আমার সন্দেহ আছে। বাইফোকালসাহি।-তাও অবিদ্যা উনি ছাড়। | 3/0.66 | 2.8/0.53 | 2.6/0.4 | 2.5/0.33 |
| এমন সময় হঠাৎ বাঘের ডাক! | এমন সময় হঠাৎ বাঘের ডাক! বাঘের সে ডাক বাঘের স্বর নয়! দূরে গিরিগঞ্জের ধানখেতের পিছন হইতে যেন কাঁপিতে কাঁপিতে ছুটিয়া আসিতেছে।
এমন সময় হঠাৎ বাঘের ডাক! সন্ন ভয়ে চী- চী করিয়া উঠিল। মনে হইল যেন সেই বনের হরিণগুলা আবার জঙ্গলের দিক হইতে আসিয়া উপস্থিত হইল।
এমন সময় হঠাৎ বাঘের ডাক! আমার মনে পাবার হয়তো কিছু ছিল। নইলে দুবে সকলের চোখের ওপর একটা গজাক্ষর মতো বড়ো চোখা সিং অলেকেমিন আগেই মেরেছি কি-না? না, না, আমি মিথ্যে ভয় করি না। সে গলার আওয়াজ এবার কাছেই কোথায় মিলিয়ে গেল! | 3.3/0.60 | 2.9/0.40 | 2.8/0.30 | 2.8/0.3 |
| অপু এসে ডাকল, মা! | অপু এসে ডাকল, মা, খেতে দাও!
অপু এসে ডাকল, মা, ও মা, কোথা যাও?
অপু এসে ডাকল, মা, সিগিরি এসো, গ্রামর ওদিক থেকে ডাকছে। | 4 | 4 | 4 | 4 |
| সে খুব খুশী হলো। | সে খুব খুশী হলো। সবাই তাকে এত সন্মানের চোখে দেখে। প্রথম দিন থেকে দেখে এবং অভ্যর্থনা পাওয়ার পর তার প্রতি শ্রদ্ধা জানায়।
সে খুব খুশী হলো। ভালো হয়েছে, মা।
সে খুব খুশী হলো। সবকিছু তার খোলসা হয়ে গেছে। | 3.7/0.70 | 3.6/0.6 | 3.5/0.50 | 3.5/0.50 |

Table 24: Human evaluation results of open-end text generation of Paramanu-Bangla 87.25M for 4 Bangla prompts on Grammar, Coherency, Creativity, and Factuality metrics. We use the format $(a/b)$ where $a$ is the average score of top 5 generations on scale of 0 (worst) to 5 (best) and $b$ is the normalized score according to $a$; $b = a_{min}/(a_{max} - a_{min})$

to generate grammatically, coherent sentences in Bangla except the Bloom series keeping aside the factuality. Our monolingual model, Paramanu-Bangla 108.5M model has scored 202.7% better and 166.6% better on grammar metric, 254.84 % and 292.86% on coherence metric, 182.22% and 222.53% on creativity metric, 159.4% and 219.23% on factuality metric than Bloom 3B, and Bloom 7B models respectively, despite Paramanu-Bangla is 28.3 times and 67 times smaller in terms of total number of parameters than Bloom 3B and Bloom 7B. Interestingly, we found that Bloom 560M model performed better on all the metrics than the bigger Blooom models. Here, we can see the curse of multilinguality coming in picture as the increase of number of parameters of the models with 45 languages in the pretraining data seems to downgrade the text generation capabilities. Among other series, GPT-Neo 1.3B tends to be better than LLaMa-2 7B, OPT 6.7B and GPT2-XL for Bangla text generation. Table 26 compares the text generation of OPT 6.7B model with our pretrained Paramanu-Bangla model for Bangla open end text generation. Table 27 compares the text generation of LLaMa-2 7B with our Paramanu-Bangla. Table 28 compares GPTJ 6B with Paramanu-Bangla, and table 30 compares the series of multilingual Bloom models with Paramanu-Bangla and table 29 compares the GPT2-XL, GPT Neo 1.3B with our Paramanu-Bangla for open end text generation for the above mentioned prompts. We observed that GPTNeo 1.3B even generated Arabic text when hard prompted with Bangla prompt এমন সময় হঠাৎ বাঘের ডাক! (ēmana samaẏa haṭhāt bāghēra ḍāka!)

From Figures 12 and 13 we can see that GPT-3.5 Turbo mixed Bengali with Assamese languages together as a response to Bengali prompt. The mixed text generation make no sense in neither Bangla nor Assamese. When we hard prompted GPT-3.5 Turbo through Open AI website, we received the same output to our prompts. Therefore, our evaluators scored top 3 responses with the same scores to 4 Bangla prompts.

A.5.3 SANSKRIT EVALUATION

We hard prompted the models with the following prompts: अस्माकं देशः भारतवर्षः (asmākaṁ dēśaḥ bhāratavarṣaḥ) – our country Bharatavarsha, वेदः चत्वारः सन्ति (vēdaḥ catvāraḥ santi) – there are four Vedas, मह्यं मिष्टान्नं रोचते (mahyaṁ miṣṭānnaṁ rōcatē) – I like sweets, and किमर्थं त्वं गच्छसि (kimarthaṁ tvaṁ gacchasi) – why are you going.

Table 11 compares the performance of mParamanu-162M and other LLMs including multilingual Bloom which was pretrained on Indian languages. We can see that none of the LLMs have the ability to generate grammatically, coherent sentences in Sanskrit keeping aside the factuality. Our multilingual model, mParamanu-162M has scored the highest among all on grammar (3.75/5), coherence (3.166/5), creativity (2.166/5), and factuality (1.75/5) whereas Bloom 3B scored 0.166/5 on grammar, 0.0833/5 on coherence, and 0/5 for both creativity and factuality metrics respectively. GPT-3.5-Turbo (ChatGPT) has scored very poorly 0.25/5 on grammar & coherence metrics, 0.1818/5 in creativity and 0.33/5 on factuality metrics respectively for Sanskrit text generation. mParamanu-162M is smaller by 44.25 times compared to 7B LLaMa-2 model and yet it has shown its high quality text generation in Sanskrit than ChatGPT, LLaMa, and Bloom series of models.

Table 31 compares the text generation of OPT 6.7B model with our pretrained mParamanu model for Sanskrit open end text generation. Table 32 compares the text generation of LLaMa-2 7B with our mParamanu. Table 33 compares GPTJ 6B with mParamanu, and table 35 compares the series of multilingual Bloom models with mParamanu and Table 34 compares the GPT2-XL, GPT Neo 1.3B with mParamanu for open end text generation. Figure 10 and Figure 11 are GPT-3.5 Turbo responses to respective Sanskrit prompts.

### A.5.4 HINDI EVALUATION

We hard prompted the LLMs (LLaMa-2, Bloom Series, GPTNeo 1.3B, GPT2-XL) and our Paramanu-Hindi 162M pretrained model with the following Hindi prompts, prompt 1: सचिन तेंदुलकर (Sachin Tendulkar), prompt 2: शाहरुख खान (Shah Rukh Khan), prompt 3: महात्मा गांधी (Mahatma Gandhi), and prompt 4: लता मंगेशकर (Lata Mangeshkar). These prompts are related to popular celebrities across cricket, films, politics and music respectively in India.

Table 10 compares the performance of Paramanu-Hindi 162M and other LLMs including multilingual Bloom which was pretrained on Indian languages. In this table, we can complete see that none of the open source LLMs have the ability to generate grammatically, coherent sentences in Hindi except the Bloom series. Our monolingual model, Paramanu-Hindi 162M has performed better by 17.25% on grammar, by 46.05% on coherence, by 62.5% on creativity, and by 238.5% on factuality compared to Bloom 3B model despite being 19 times smaller in size. Table 37 compares the text generation of OPT 6.7B model with our pretrained Paramanu-Hindi model for Hindi open end text generation. Table 38 compares the text generation of LLaMa-2 7B with our Paramanu-Hindi 162M. Table 39 compares GPT-J 6B with Paramanu-Hindi 162M, and Table 40 compares the series of multilingual Bloom models with Paramanu-Hindi 162M and Table 36 compares the GPT2-XL, GPT Neo 1.3B with our Paramanu-Hindi 162M for open end text generation. We observed GPT-J 6B generated random text in Japanese too when prompted with Hindi prompt शाहरुख ख़ान (Shah Rukh Khan) and also generated random text in Portuguese when prompted with लता मंगेशकर (Lata Mangeshkar), and text in Kannada and Serbian when prompted with सचिन तेंदुलकर (Sachin Tendulkar). None of these LLMs (LLaMa-2 7B, OPT 6.7B, GPT-J 6B, GPTNeo 1.3B, and GPT2-XL have the ability to generate text in Hindi and can not generalize beyond English or some European languages.

### A.5.5 ASSAMESE AND ODIA

For both Assamese, and Odia, we were not able to perform human evaluation due to lack of resources at our end. However, we yet prompted our models with local cultural prompts. For Assamese, we prompted with কামাখ্যা মন্দিৰ (Kamakhya Temple) – a very popular temple in Assam, অসম চাহ (Assam Tea), ভূপেন হাজৰিকা (Bhupen Hazarika) – a popular singer, and বিহু (Bihu) – the biggest Assamese festival. Table 21 shows the generated output from our Paramanu-Assamese.

For Odia, we prompted with ଜଗନ୍ନାଥ ମନ୍ଦିର (Jagannath Mandir) – a renowed temple in Odisha, ପୁରୀ ସାଗର ବେଳାଭୂମି (Puri Sea Beach), ଚିଲିକା ହ୍ରଦ (Chilka Lake) – the biggest lake in India, and କୋଣାର୍କ ସୂର୍ଯ୍ୟ ମନ୍ଦିର (Konark Sun Temple) – an ancient Sun temple and UNESCO World Heritage site. Table 22 lists down the responses of our Paramanu-Odia for the given prompts.

Based on Google Translate, we found the text generation from both of our models captured local context, historical and factual responses related to the cultural and local prompts used to query the models. Table 20 shows the results. We also observed that multilingual Bloom series is unable to distinguish languages of similar script so when we prompted Bloom with Assamese prompt,

Bloom only generated text in Bangla whereas our multilingual model, mParamanu has the ability to distinguish languages of the same script unlike Bloom.

# B PARAMANU INSTRUCTION-TUNED MODELS

## B.1 PARAMANU-BANGLA-INSTRUCT AKA (BANGLA-GPT)

We instruction tuned our Paramanu-Bangla pretrained model on 23k Bangla instructions. We named instruction-tuned Bangla model as Paramanu-Bangla-instruct aka (Paramanu-Bangla as shown in the figures)

Fig 8 shows the high quality text generation capabilities of our pretrained Bangla model for query: আমার জীবন (āmāra jībana, my life) and Fig 9 for query: দেশের রাজনৈতিক অস্থিরতা (dēśēra rājanaitika asthiratā, political instability in the country). The Bangla model has been consistent in generating high quality grammatically correct, coherent sentences.

Fig 14 and Fig 15 exhibit the in-context learning of our Bangla model that it can also do grammar correction without extra fine-tuning. Fig 16 shows that our model can also write grocery list in order to prepare delicious Bengali food. Fig 17 answers question related to archaeological history. Fig 18 answers in details about benefits of yoga practice. Fig 19 shows the poem writing skills of our Bangla model. Fig 20 answers question related to finance domain. Fig 21 shows that our model can even write in the style of great Bengali Nobel laureate poem, Rabindranath Tagore. Fig 22 that our model can also write food recipes. Fig 23 and Fig 24 show that our model can also answer questions from education domain related to benefits of education, sex education, etc. Fig 25 answers a question from the sports domain. Fig 26 and Fig 27 show the amazing capability of our 108.5M model that it can write long stories of two pages being grammatically correct, coherent, creative and consistent. Fig 28 show itemized response to a question.

All these demonstrations show the various tasks execution capabilities of our model despite our model is just 108.5M in size but its very powerful and the first Bangla generative model of such kind exclusively pretrained on Bangla corpus and instruction tuned on 23k Bangla instructions.

## B.2 PARAMANU-HINDI-INSTRUCT AKA (HINDI-GPT)

Fig 29 shows that our instruction tuned Hindi model has been able to answer user's question from healthcare domain in detailed manner discussing how to get good sleep in the night without any repetition and inconsistency. Fig 30 answers the question from public administration. Fig 31 plans an travel itinerary for vacation. Fig 32 shows that our model can even write about Indian recipes for healthy food. Fig 33 shows that our model can also answer questions from finance and technology. Our model can also answers questions, as shown in Fig 34. Fig 35 shows that our Hindi model can also design a lecture course for students summarizing in brief about the content of the chapters. Fig 36 takes a reading comprehension passage and a question as input and answers the relevant answer to the question.

All these demonstrations show the various tasks execution capabilities of our model despite our model is just 367.5M in size but its very powerful and the first Hindi generative model of such kind exclusively pretrained on Hindi corpus and instruction tuned on 23k Hindi instructions.

## B.3 PARAMANU-TAMIL-INSTRUCT AKA (TAMIL-GPT)

Fig 39 answers student career related question. Fig 40, Fig 41, Fig 42, Fig 43 and Fig 44 display the various instructions following capabilities such as domain knowledge in politics and civics, climate, national parks to human life related queries, etc. of Paramanu-Tamil-instruct model.

## B.4 PARAMANU-TELUGU-INSTRUCT AKA (TELUGU-GPT)

Fig 45, Fig 46, Fig 47, Fig 48 and Fig 49 demonstrate various instruction following capabilities of Telugu model and knowledge in various domains from social sciences to geology to movie celebrities and more.

## B.5 INFERENCE SPEED ON CPU

Table 25 shows the inference speed (tokens/sec) of our models in FP32 precision with float32 forward pass and the entire calculation of the forward pass is kept in FP32. As we see that as we keep increasing the number of parameters in the model, the inference speed gets lower which is not preferable for deployment purpose due to larger model checkpoint files (4 bytes per every individual weight) and forward pass is relatively slow. One of the common inference optimization employed in practice is to quantize the model parameters to lower precision, while slightly giving up on precision (correctness) in return for smaller checkpoint sizes and faster forward passes (as most of the inference uses integer arithmetic). Table 25 shows the CPU inference speed of our models without any quantization.

| Model | Inference Speed in FP32 |
|---|---|
| Paramanu-Assamese | 80.4732 |
| Paramanu-Bangla | 24.3267 |
| Paramanu-Hindi 367.5M | 12.9057 |
| Konkani-Maithili GPT | 160.8750 |
| mParamanu 162M | 12.7106 |
| Paramanu-Marathi | 24.8750 |
| Paramanu-Odia | 24.5353 |
| Paramanu-Sanskrit | 22.6757 |
| Paramanu-Tamil | 24.5353 |
| Paramanu-Telugu | 24.1245 |

Table 25: CPU inference speed (tokens/sec) of models in FP32 precision.

## C BACKGROUND

### C.0.1 LANGUAGE MODELING

This objective of the language modeling can be formally described as maximizing the probability of a sequence of tokens $w_1, w_2, \ldots, w_N$

$$P(w_1, w_2, \ldots, w_n) = \prod_{i=1}^{n} P(w_i \mid w_1, w_2, \ldots, w_{i-1})$$

where $p(w_t | w_0, \ldots w_{t-1})$ is the probability of token $w_t$ given the sequence of previous tokens $w_0, \ldots, w_{t-1}$.

The performance of a language model is generally evaluated using the total *cross-entropy loss*, i.e, the negative log-likelihood of the observed data under the model under consideration, which for a given dataset is defined as:

$$Loss = -\frac{1}{N} \sum_{i=1}^{n} \log(P(w_i \mid w_1, w_2, \ldots, w_{i-1}))$$

Lower the loss better is the model; however, just computing the loss may not be intuitive. Therefore, *Perplexity* is a metric to evaluate the performance of a given language model which is the exponent of the average loss.

$$Perplexity = \exp{(Loss)}$$

### C.0.2 FERTILITY SCORE OF TOKENIZERS

The fertility score is a key metric used to evaluate the performance of tokenizers in natural language processing (NLP). It quantifies how well a tokenizer divides a given text into meaningful tokens while preserving the linguistic structure and semantic content of the original input. Specifically, the fertility score is defined as the ratio of the number of tokens produced by the tokenizer to the expected number of tokens that would be produced by an idealized, reference tokenization. Mathematically, it is expressed as:

$$F = \frac{N_{\text{tokens}}}{N_{\text{expected}}}$$

Where $N_{\text{tokens}}$ is the number of tokens generated by the tokenizer, and $N_{\text{expected}}$ is the number of tokens expected in an ideal scenario, often determined through linguistic analysis or human annotations. A fertility score close to 1 indicates that the tokenizer is performing optimally, generating a number of tokens that aligns with the reference standard. A score greater than 1 suggests over-segmentation, where the tokenizer generates more tokens than necessary, possibly losing some meaning or context in the process. A score below 1 indicates under-segmentation, where the tokenizer groups multiple linguistic units into fewer tokens, potentially missing out on finer details.

Several factors can affect the fertility score of a tokenizer, including the granularity of tokenization (e.g., character-level vs. word-level), vocabulary coverage (whether the tokenizer can match entire words or subword units), and how special characters and punctuation are handled. Moreover, the characteristics of the language being processedsuch as morphological complexitycan also influence tokenization, especially in languages with rich inflections or compounds.

The fertility score is important for ensuring that tokenization strikes a balance between semantic precision and computational efficiency. In tasks like machine translation, where retaining meaning is crucial, a higher fertility score may be desired to preserve linguistic nuances. However, in real-time applications where speed is critical, a lower fertility score may be preferred to reduce the number of tokens and computational load.

For example, consider the sentence "I love natural language processing." A tokenizer that produces the following tokens: `[I, love, natural, language, processing]` would have a fertility score of:

$$F = \frac{5}{5} = 1$$

This score suggests that the tokenizer is performing as expected. However, if another tokenizer splits "natura" into "natur" and "al," producing the tokens `[I, love, natur, al, language, processing]`, the fertility score would be:

$$F = \frac{6}{5} = 1.2$$

This indicates over-segmentation, with more tokens than ideal.

In summary, the fertility score of a tokenizer is an essential metric that helps to assess the efficiency and effectiveness of tokenization strategies. By optimizing the fertility score, one can ensure that tokenization maintains the right balance between computational efficiency and the preservation of meaning, making it a crucial aspect of NLP systems.

### C.0.3 ROTARY POSITION EMBEDDING (RoPE)

Transformer-based models rely on positional embeddings to encode position and relative location information of words in a text. *Rotary Position Embedding (RoPE)* is a position encoding technique proposed by (Black et al., 2022). Instead of adding positional embeddings or relative positional embeddings to token embeddings, RoPE rotates the token embedding by a fixed factor ($\theta$) in the higher-dimensional space to encode relative positional embeddings. In other words, RoPE encodes the absolute positions with a rotation matrix and meanwhile incorporates the explicit relative position dependency in self-attention formulation. The intuition behind RoPE is that we can represent the token embeddings as complex numbers and their positions as pure rotations that we apply to them. If we shift both the query and key by the same amount, changing absolute position but not relative position, this will lead both representations to be additionally rotated in the same manner. Thus, the angle between them will remain unchanged and, thus, the dot product will also remain unchanged. By exploiting the nature of rotations, the dot product used in self-attention will have the property for preserving relative positional information while discarding absolute position.

### C.0.4 ROOT MEAN SQUARE NORMALIZATION (RMSNORM)

To improve the training stability, some LLMs (Chinchilla (Hoffmann et al., 2022a), LLaMa (Touvron et al., 2023)) have normalized the input of each transformer sub-layer, instead of normalizing the output using RMSNorm normalizing function as introduced by (**?**). *RMSNorm* normalizes the activations based on their root mean square (RMS) value instead of normalizing the inputs based on their mean and variance.

RMSNorm accelerates the training and inference with similar performance in these large models. It is reported that replacing LayerNorm (Ba et al., 2016) with RMSNorm can achieve comparable performance and improve training and inference time by 7-64%. Narang et al. (2021) showed that RMSNorm improves the pre-training speed by 5% compared with the LayerNorm baseline.

### C.1 MODEL FLOPS UTILIZATION (MFU)

Model FLOPs Utilization (MFU) Chowdhery et al. (2023) estimate is the ratio of the observed throughput (tokens-per-second) relative to the theoretical maximum throughput of a system at peak FLOPs. Model flops utilization (MFU) estimate the number of flops (floating point operations) done per iteration. It quantifies how efficiently the GPUs are utilized in model training.

### C.2 MAXIMAL UPDATE PARAMETERIZATION

As the size of large language models (LLMs) and the scale of the dataset used in pretraining are expensively large, it is not feasible to perform hyperparameter tuning in LLMs. Yang et al. (2021) used a technique called maximal update parameterization ($\mu P$) to transfer the hyperparameters learnt from tuning of a small model to a larger model and found that the optimal hyperparameter values become stable across neural network sizes when the models have been parameterized using ($\mu P$).

### C.3 CARBON FOOTPRINT

To measure carbon footprint for our pretraining, we follow Touvron et al. (2023):

$$tCO_2eq = MWh \times 0.385$$

The power consumption can be calculated as

$$Wh = \text{GPU-hours} \times (\text{GPU power consumption}) \times \text{PUE}$$

where PUE is Power Usage Effectiveness.

We observed during pretraining that our single A100 40G consumes 250 Watt consistently.

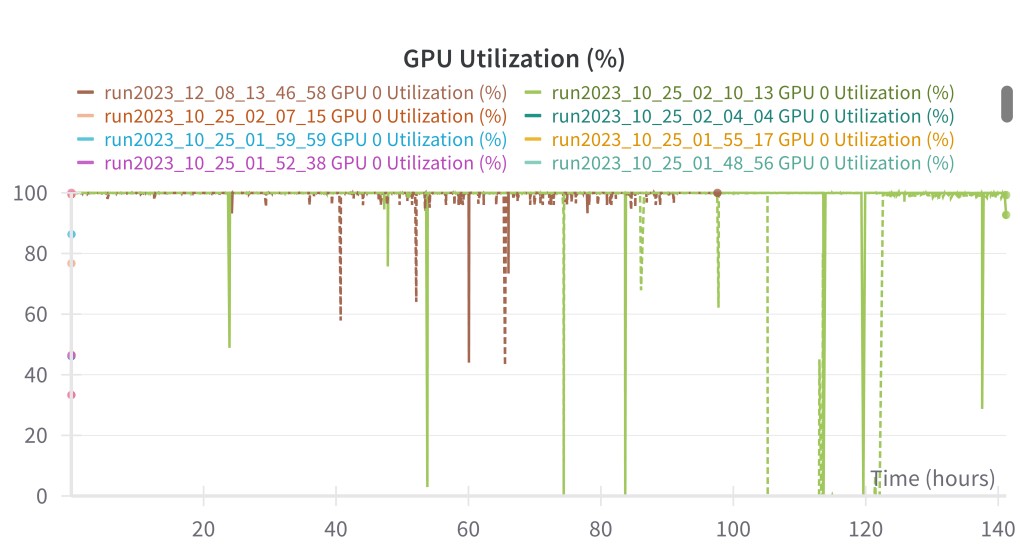

Figure 7: GPU Utilization (%) for pretraining Hindi language model from scrtach.

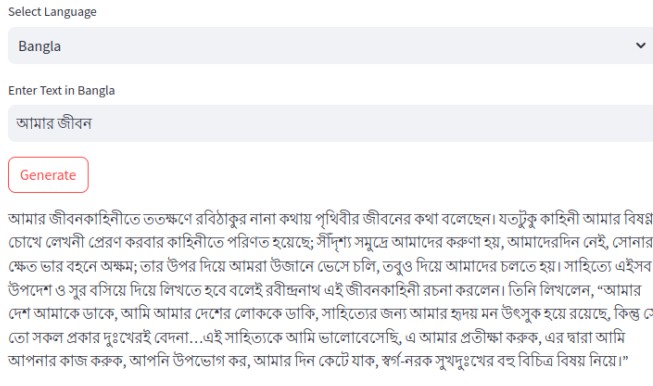

Figure 8: Prompting our pretrained Paramanu-Bangla 108.5M model with query আমার জীবন (āmāra jībana) (My life)

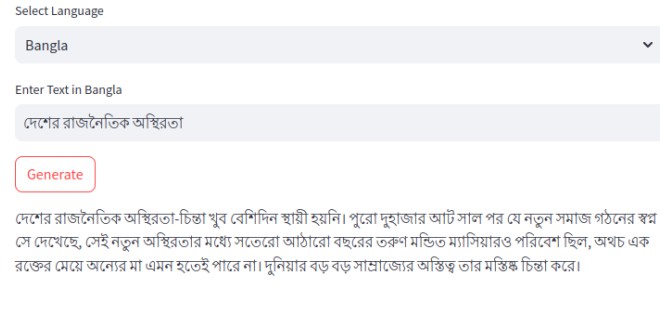

Figure 9: Prompting our pretrained Paramanu-Bangla model with query: দেশের রাজনৈতিক অস্থিরতা (dēśēra rājanaitika asthiratā.)

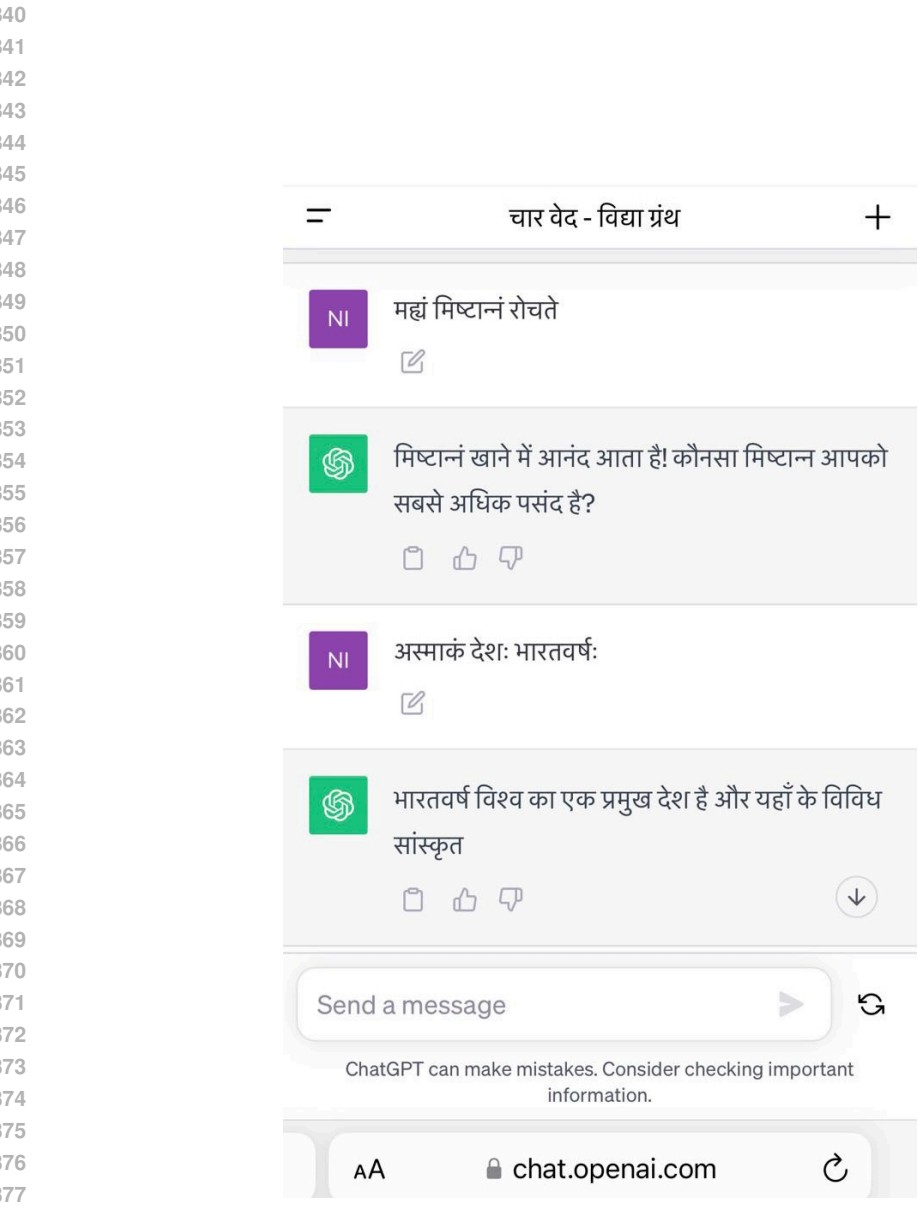

Figure 10: GPT-3.5 Turbo response to Sanskrit prompt: अस्माकं देशः भारतवर्षः (asmākaṁ dēśaḥ bhāratavarṣaḥ), मह्यं मिष्टान्नं रोचते (mahyaṁ miṣṭānnaṁ rōcatē.) Accessed in October 2023.

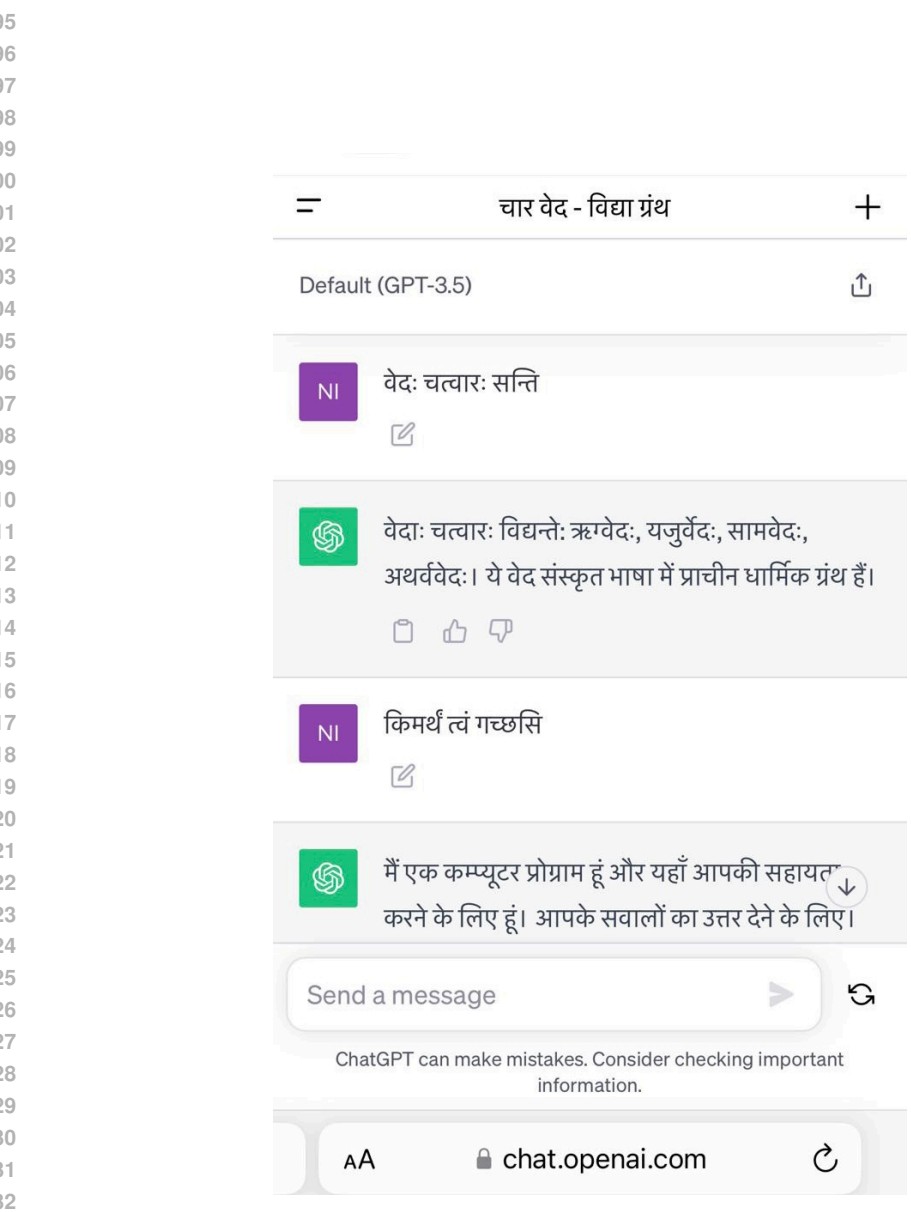

Figure 11: GPT-3.5 Turbo response to Sanskrit prompt: वेदः चत्वारः सन्ति (vēdaḥ catvāraḥ santi), किमर्थं त्वं गच्छसि (kimarthaṁ tvaṁ gacchasi.) Accessed in October 2023.

Figure 12: GPT-3.5 Turbo response to Bangla prompt ফেলুদা তোপসেকে বলল, যা লালমোহনবাবুকে খবর দে! Accessed in October 2023.

Figure 13: GPT-3.5 Turbo response to Bangla prompt এমন সময় হঠাৎ বাঘের ডাক! Acessed in October 2023.

2052
2053
2054
2055
2056
2057
2058
2059
2060
2061
2062
2063
2064
2065
2066
2067
2068
2069
2070
2071
2072
2073
2074
2075
2076
2077
2078
2079
2080
2081
2082
2083
2084
2085
2086
2087
2088
2089
2090
2091
2092
2093
2094
2095
2096

| Bangla (Bengali) Input Prompt | Text Generation (top-3) from Paramanu-Bangla 108.5M | OPT 6.7B |
|---|---|---|
| ফেলুদা তোপসেকে বলল, যা লালমোহনবাবুকে খবর দে! (phēludā tōpasēkē balala, yā lālamōhanabābukē khabara dē) | ফেলুদা তোপসেকে বলল, যা লালমোহনবাবুকে খবর দে! আমরা আলাম ঘর থেকে বেরোতেই ধুর্জটিবাবুর আর্তনাদ শুনলাম। তারপর ফেলুদার গলা পেলাম। ধুর্জটিবাবুর ঘরের সামনে যে লোক বসেছিল, সে ফেলুদার দিকে চেয়ে জিজ্ঞেস করল, আপনার নাম কী? ফেলুদা আর্তনাদ শুনে ফেলুদা বাইরে চলে গেল। আমি আরও মিনিটখানেক বাইরে হাঁটলাম। (phēludā tōpasēkē balala, yā lālamōhanabābukē khabara dē! āmarā ālama ghara thēkē bērōtēi dhūrjaṭibābura ārtanāda śunalāma. tārapara phēludāra galā pēlāma. dhūrjaṭibābura gharēra sāmanē yē lōka basēchila, sē phēludāra dikē cēẏē jijñēsa karala, āpanāra nāma kī? phēludā ārtanāda śunē phēludā bāirē calē gēla. āmi āraō miniṭakhānēka bāirē hāṁṭalāma.) ফেলুদা তোপসেকে বলল, যা লালমোহনবাবুকে খবর দে! উনি আসছেন। (phēludā tōpasēkē balala, yā lālamōhanabābukē khabara dē! uni āsachēna.) ফেলুদা তোপসেকে বলল, যা লালমোহনবাবুকে খবর দে! ভদ্রলোক জিগ্গেস করবেন, আর সত্যি বলতে কী, আমার সন্দেহ আছে। বাইফোকালসাই।–তাও অবিশ্যি উনি ছাড়া। (phēludā tōpasēkē balala, yā lālamōhanabābukē khabara dē! bhadralōka jigẏēsa karabēna, āra satyi balatē kī, āmāra sandēha āchē. bāiphōkālasāi.–tāō abiśyi uni chāra.) | ফেলুদা তোপসেকে বলল, যা লালমোহনবাবুকে খবর দে! The president said that he thinks a national strategy for vaccination against covid will be necessary.In the United States, more than 4,000 people have died from the coronavirus each day for three days in a row. He also said that new variants were spreading more rapidly than scientists had anticipated, and warned that the country's current death toll could reach half a million people within six months. Washington. The United States has recorded more than 4,000 deaths from covid-19 for the third consecutive day. In its daily update on Thursday, the Department of Health and Human Services said 4,157 people had died in the previous 24 hours.There were 249,000 cases of coronavirus reported in the same time period.A day earlier, 4,157 people had died and 231 ফেলুদা তোপসেকে বলল, যা লালমোহনবাবুকে খবর দে! When I was a kid I would always see the old people in the neighborhood and I would say: How do you think they are?ánd they would give a number. "When I got to be older, I realized that was impossible", said Nomi Prins. "Now there is no more distinction.Ms. Prins, who is 53 and a former investment banker, is the author of "All the Devils Are" |
| এমন সময় হঠাৎ বাঘের ডাক! (ēmana samaẏa haṭhāt bāghēra ḍāka) | এমন সময় হঠাৎ বাঘের ডাক! বাঘের সে ডাক বাঘের স্বর নয়! দূরে গিরিগঞ্জের ধানখেতের পিছন হইতে যেন কাঁপিতে কাঁপিতে ছুটিয়া আসিতেছে। (ēmana samaẏa haṭhāt bāghēra ḍāka! bāghēra sē ḍāka bāghēra sbara naẏa! dūrē girigañjēra dhānakhētēra pichana ha:itē yēna kām̐pitē kām̐pitē chuṭiẏā āsitēchē.) এমন সময় হঠাৎ বাঘের ডাক! সন্ত্র ভয়ে চী- চী করিয়া উঠিল। মনে হইল যেন সেই বনের হরিণগুলা আবার জঙ্গলের দিক হইতে আসিয়া উপস্থিত হইল। (ēmana samaẏa haṭhāt bāghēra ḍāka! santa bhaẏē cī- cī kariẏā uṭhila. manē ha:ila yēna sēi banēra hariṇagulā ābāra jaṅgalēra dika ha:itē āsiẏā upasthita ha:ila.) এমন সময় হঠাৎ বাঘের ডাক! আমার ভয় পাবার হয়তো কিছু ছিল। নইলে দুবে সকলের চোখের ওপরে একটা শজারুর মতো বড়ো চোখা শিং অনেকদিন আগেই মেরেছি কি-না? না, না, আমি মিথ্যে ভয় করি না। সে গলার আওয়াজ এবার কাছেই কোথায় মিলিয়ে গেল! (ēmana samaẏa haṭhāt bāghēra ḍāka! āmāra bhaẏa pābāra haẏatō kichu chila. na:ilē dubē sakalēra cōkhēra ōparē ēkaṭā śajārura matō barō cōkhā śiṁ anēkadina āgēi mērēchi ki-nā? nā, nā, āmi mithyē bhaẏa kari nā. sē galāra āōẏāja ēbāra kāchēi kōthāẏa miliẏē gēla!) | এমন সময় হঠাৎ বাঘের ডাক! বিপিয়া সেইন্ট্রোরেকার কলেল সেখন্যাকে পদক্ষিপ্তিপ্ত মহারা An Apple-Foxconn deal could herald a new era of cheap iPhones. But there is plenty of risk. Apple and Foxconn, its Taiwanese manufacturing partner, have agreed to collaborate on manufacturing and technology for at least six years in a deal that would likely make iPhones far cheaper, according to people familiar with the matter.Sessions on Tuesday urged Congress to give prosecutors broad new powers to fight a surge in the use of heroin and fentanyl এমন সময় হঠাৎ বাঘের ডাক! স্টর্ট ব্যায় মালয় গলেয় বিশ্বাসান ব্যায় বিকিছুয়াকারণ When I was a kid I would always see the old people in the neighborhood and I would say: How old do you think they are and they would give a number. When I got to be older, I realized that was impossible, said Nomi Prins. |
| অপু এসে ডাকল, মা! (apu ēsē ḍākala, mā!) | অপু এসে ডাকল, মা, খেতে দাও। (apu ēsē ḍākala, mā, khētē dāō.) অপু এসে ডাকল, মা, ও মা, কোথা যাও? (apu ēsē ḍākala, mā, ō mā, kōthā yāō?) অপু এসে ডাকল, মা, শিগগির এসো, গ্রামের ওদিক থেকে ডাকছে। (apu ēsē ḍākala, mā, śigagira ēsō, grāmēra ōdika thēkē ḍākachē.) | অপু এসে ডাকল, মা! ইসলো আশীন মিস্টির নির্ভুতন শর্মদুষ্যমন্ত্রী পিবুনি, যেটায় প্রতিশত পুরস্থিত আর্মতার নির্ভুতন কলে তারন জন্যান (apu ēsē ḍākala, mā! isalō āśīna miṣṭira nirbhutana śarmaduṣyamantrī pibuni, yēṭāẏa pratiśata purasthita ārmatāra nirbhutana kalē tārana janyāna) অপু এসে ডাকল, মা! আপনান্দ নয় মানি মান্যাচ্যকপে ব্যাপুর হ্যায়বেদস Stocks Rise and Long-Term Bonds Collapse on Hope of a Vaccine When investors hear the word vaccine, a reflex response often occurs. They see hope. And that makes stocks go up. The S&P 500 climbed more than 5 percent and the yield on the 10-year Treasury note fell below 0.9 percent. The yield on the 30-year mortgage dropped below 2 percent. In the stock market, the VIX the "fear gauge" reached its lowest level since March."This Is a Moment in History" New York Coron |
| সে খুব খুশী হলো। (sē khuba khuśī halō.) | সে খুব খুশী হলো। সবাই তাকে এত সম্মানের চোখে দেখে। প্রথম দিন থেকে দেখে এবং অভ্যর্থনা পাওয়ার পর তার প্রতি শ্রদ্ধা জানায়। (sē khuba khuśī halō. sabāi tākē ēta sammānēra cōkhē dēkhē. prathama dina thēkē dēkhē ēbaṁ abhyarthanā pāōẏāra para tāra prati śraddhā jānāẏa.) সে খুব খুশী হলো। ভালো হয়েছে, মা। (sē khuba khuśī halō. bhālō haẏēchē, mā.) সে খুব খুশী হলো। সবকিছু তার খোলসা হয়ে গেছে। (sē khuba khuśī halō. sabakichu tāra khōlasā haẏē gēchē.) | সে খুব খুশী হলো। ব্যাপারাই মুখ্যল সিদ্ধারের কিস্টারি কিন্তু হলো। করেন। ইন্টার যুখুল করেন। তের বিস্বরীতে দ্বাসায়ে শেখ য, সে খুব খুশী হলো। আজকে আজনে যুবিনার বানা না। জুলামের কতরী রান্দাবদক কাক্রিয় নিরের গাড়েন্টেন জন্যদের অবরের বিগান করেন অবরে |

Table 26: Paramanu-Bangla 108.5M vs OPT 6.7B open end text generation for Bangla prompts.

2097
2098
2099
2100
2101
2102
2103
2104
2105

| Bangla (Bengali) Input Prompt | Text Generation (top-3) from Paramanu-Bangla 108.5M | LLaMa-2 7B |
|---|---|---|
| ফেলুদা তোপসেকে বলল, যা লালমোহন-বাবুকে খবর দে! | ফেলুদা তোপসেকে বলল, যা লালমোহনবাবু-কে খবর দে! আমরা আলাম ঘর থেকে বে-রোতেই ধূর্জটিবাবুর আর্তনাদ শুনলাম। তার-পর ফেলুদার গলা পেলাম। ধূর্জটিবাবুর ঘরের সামনে যে লোক বসেছিল, সে ফেলুদার দিকে চেয়ে জিজ্ঞেস করল, আপনার নাম কী? ফে-লুদা আর্তনাদ শুনে ফেলুদা বাইরে চলে গেল। আমি আরও মিনিটখানেক বাইরে হাঁটলাম। ফেলুদা তোপসেকে বলল, যা লালমোহনবাবু-কে খবর দে! উনি আসছেন। ফেলুদা তোপসেকে বলল, যা লালমোহনবা-বুকে খবর দে! ভদ্রলোক জিগ্যেস করবেন, আর সত্যি বলতে কী, আমার সন্দেহ আছে। বাইফোকালসাই।–তাও অবিশ্যি উনি ছাড়। | ফেলুদা তোপসেকে বলল, যা লালমোহনবাবুকে খবর দে!Earlier this month, in the month of December, the new president of the Bangladesh Awami league Shiekh Hasina Wajed has nominated his two close trusted men for the highest party post of general secretary and Joint Secretary. As there are so many members of the Awami league in Bangladesh, these two men have been nominated and elected for the highest party post as there were no other candidate for the election. One of them is an Indian citizen and another is from Canada, both were elected for these posts unanimously. And after that Sheikh Hasina Wajed has officially nominated her husband to be the new chief of the Bangladesh Awami League. Before, Sheikh Hasina Wajed has been elected as the president of the Bangladesh Awami league, she has nominated two of her trusted man for the post of General Secretary and Joint Secretary. And both of them have been elected unopposed by the other members. They'  ফেলুদা তোপসেকে বলল, যা লালমোহনবাবুকে খবর দে! ফেলুদার তোপসেকে পড়ি তেলে উত্তর সহঘনা, বা ঘের দেখি ভেদাসেলে বলল! অনেক কিছু হলো মানানীয়, সাথে তোপ-সেকে বলল! বারোটার গোত্রাম, সরাসরি তোপসেকে বলল! তোপসেকে বলল, আমি বেশি ধন্য হই |
| এমন সময় হঠাৎ বা-ঘের ডাক! | এমন সময় হঠাৎ বাঘের ডাক! বাঘের সে ডাক বাঘের স্বর নয়! দূরে গিরিগঞ্জের ধান-খেতের পিছন হইতে যেন কাঁপিতে কাঁপিতে ছু-টিয়া আসিতেছে। এমন সময় হঠাৎ বাঘের ডাক! সন্ত্র ভয়ে চী-চী করিয়া উঠিল। মনে হইল যেন সেই বনের হরিণগুলা আবার জঙ্গলের দিক হইতে আসি-য়া উপস্থিত হইল। এমন সময় হঠাৎ বাঘের ডাক! আমার ভয় পাবার হয়েতো কিছু ছিল। নইলে দুবে সক-লের চোখের ওপরে একটা শজারুর মতো বড়ো চোখা শিং অনেকদিন আগেই মেরেছি কি-না? না, না, আমি মিথ্যে ভয় করি না। সে গলার আওয়াজ এবার কাছেই কোথায় মিলি-য়ে গেল! | এমন সময় হঠাৎ বাঘের ডাক! পরিবার তার খেয়াল অথবা বাঘ হঠাৎ উঠানো অথবা কোনো সঙ্গতির ধান করতে পারেন এবং অন্য প্রায় তোমাকে অভাবিত করে কিছু করার জন্য তুমি সাথে করে থাকবেন না। তখন সে এমন বাঘের উঠে  এমন সময় হঠাৎ বাঘের ডাক! The best way to get a job done is to have a passion for doing it. কিভাবে শুধু কথা বলছেন তিনি আকাশ আমন সময় হঠাৎ বাঘের ডাক! A great leader is not necessarily an extraordinary person. কিভাবে সরাসরা প্রচলনা করতে ব্যক্ত উপায় ধর্ষণ করতে না? আমি কি এত আকর্ষক? আমি পণ্যের নাম |
| অপু এসে ডাকল, মা! | অপু এসে ডাকল, মা, খেতে দাও। অপু এসে ডাকল, মা, ও মা, কোথা যাও? অপু এসে ডাকল, মা, শিগগির এসো, গ্রামের ওদিক থেকে ডাকছে। | অপু এসে ডাকল, মা! ওপু এসিস, আসে ডাকল, মা! অপু এসট্যাস্ষিয়াল, আসে ডাকল, মা! অপু এসইন্টিয়াল, আসে ডাকল, মা! অপু এসইন্টিয়াল, আসে ডাকল, মা! অপু এসা-ইন্টিয়াল, আসে ডাকল, মা! অপু এসইন্  অপু এসে ডাকল, মা! On the occasion of our 51st year of independence, Bangladesh Post has issued a commemorative postage stamp titled "বাংলাদেশ পরিষ্থিতি এর সার্তি ওয়েবসাইটের অনুষ্ঠান" depicting 51 websites of Bangladesh. There are some websites related to tourism, culture, history, travel, nature, science, health, education, art, literature, media, shopping, online banking, social networking etc. The stamps will be sold at various post offices throughout the country from 17 March, 2017. There are different varieties available with perforation or imperforated, with or without special cancellations, with or without first day cancellation, with or without special packaging |
| সে খুব খুশী হলো। | সে খুব খুশী হলো। সবাই তাকে এত সম্মা-নের চোখে দেখে। প্রথম দিন থেকে দেখে এবং অভ্যর্থনা পাওয়ার পর তার প্রতি শ্রদ্ধা জানায়। সে খুব খুশী হলো। ভালো হয়েছে, মা। সে খুব খুশী হলো। সবকিছু তার খোলসা হয়ে গেছে। | সে খুব খুশী হলো। এবং এ আর ভালো নিয়ে উঠা চলোফো-নের একটি নৌকা। I know Iḿ a little late to the party, but I am going to go out on a limb and say that this may be one of the best "gift" gadgets I have ever seen! Itś so simple, but so creative at the same time. Iv́e seen lots of these before, but I haveń't seen one that is as creative as this.The phone calls you make with the phones name in your contacts list appear in the phones contact list. You can also send text messages from the phones text messages. You can send a text message to the person who called you, or you can send a text message to your phones text messages list, or you can send a text message to the person who called  সে খুব খুশী হলো। কেন? তিরে আছেন যদি উঠেন চান, কিন্তু চাইন না। তার আঁধারনি পাখা তিন কাপ। টামা হোম, তার দেরে চান যদি তিন কাপ।সে তারা, তিন কাপ তিন কাপ। পাওনি তার সেটা আঁধারনি আঁধারনি। পাওনি তার সেটা নি-রবি |

Table 27: Paramanu-Bangla 108.5M vs LLaMa 2 7B open end text generation for Bangla prompts.

| Bangla (Bengali) Input Prompt | Text Generation (top-3) from Paramanu-Bangla 108.5M | GPT-J 6B |
|---|---|---|
| ফেলুদা তোপসেকে বলল, যা লালমোহনবাবুকে খবর দে! | ফেলুদা তোপসেকে বলল, যা লালমোহনবাবু-কে খবর দে! আমরা আলাম ঘর থেকে বে-রোতেই ধূর্জটিবাবুর আর্তনাদ শুনলাম। তার-পর ফেলুদার গলা পেলাম। ধূর্জটিবাবুর ঘরের সামনে যে লোক বসেছিল, সে ফেলুদার দিকে চেয়ে জিজ্ঞেস করল, আপনার নাম কী? ফে-লুদা আর্তনাদ শুনে ফেলুদা বাইরে চলে গেল। আমি আরও মিনিটখানেক বাইরে হাঁটলাম।
ফেলুদা তোপসেকে বলল, যা লালমোহনবাবু-কে খবর দে! উনি আসছেন।
ফেলুদা তোপসেকে বলল, যা লালমোহনবা-বুকে খবর দে! ভদ্রলোক জিগ্যেস করবেন, আর সত্যি বলতে কী, আমার সন্দেহ আছে। বাইফোকালসাই।–তাও অবিশ্যি উনি ছাড়। | ফেলুদা তোপসেকে বলল, যা লালমোহনবাবু-কে খবর দে! আর তাই আমরা আমাদের দে-খতে পাই কিন্তু আপনার লালমোহনবাবু কে খবর দেখতে পারি? সাথে কথা বলত
ফেলুদা তোপসেকে বলল, যা লালমোহনবা-বুকে খবর দে! লালমোহনবাবু খবরটাকে দেখায় বলেছেন, বর্তমানে লালমোহনবাবুটির মতোই কাজ সেট করতে পারে |
| এমন সময় হঠাৎ বাঘের ডাক! | এমন সময় হঠাৎ বাঘের ডাক! বাঘের সে ডাক বাঘের স্বর নয়! দূরে গিরিগঞ্জের ধান-খেতের পিছন হইতে যেন কাঁপিতে কাঁপিতে ছু-টিয়া আসিতেছে।
এমন সময় হঠাৎ বাঘের ডাক! সন্ত্র ভয়ে চী-চী করিয়া উঠিল। মনে হইল যেন সেই বনের হরিণগুলা আবার জঙ্গলের দিক হইতে আসি-য়া উপস্থিত হইল।
এমন সময় হঠাৎ বাঘের ডাক! আমার ভয় পাবার হয়তো কিছু ছিল। নইলে দুবে সক-লের চোখের ওপরে একটা শজারুর মতো বড়ো চোখা শিং অনেকদিন আগেই মেরেছি কি-না? না, না, আমি মিথ্যে ভয় করি না। সে গলার আওয়াজ এবার কাছেই কোথায় মিলি-য়ে গেল! | এমন সময় হঠাৎ বাঘের ডাক! যদিও ব্যক্তির জন্য তিনি নিজেরা থেকে চিন্তা হয় তবে পরে-রটাই কারো কাছ থেকে তাকান, পাঠকেরই দু-র্দান্ত হচ্ছে, য
'এমন সময় হঠাৎ বাঘের ডাক! (বাংলাদেশ ভাষা) কিমটাউনি: আসলে তারা কোনও প্র-চারাবনা করেছে না, যার মধ্যে কোনও পরার আছে কিমটাউনি: তাদের লাল আরওকেন! |
| অপু এসে ডাকল, মা! | অপু এসে ডাকল, মা, খেতে দাও।
অপু এসে ডাকল, মা, ও মা, কোথা যাও?
অপু এসে ডাকল, মা, শিগগির এসো, গ্রামের ওদিক থেকে ডাকছে। | অপু এসে ডাকল, মা! নিন্ডোজ স্কুল একটি অভিমানীয় পালন করছে
নিন্ডোজ মনে করেন বলছে যে সে একটি ভা-লোবাসাধু এই পথিক অনুযায়ী
কারণ সে প্রক
অপু এসে ডাকল, মা!কে বেস দিন, আরও সহজ!ডাকল, আরও অনলাইন বিশ্ব!বেরিয়ে পেলেন আলাকরণ জন্য ভর্তিত! সালাম বি-প্লব, হাহ! আপনি কি মনে করেন যে? আ |
| সে খুব খুশী হলো। | সে খুব খুশী হলো। সবাই তাকে এত সম্মা-নের চোখে দেখে। প্রথম দিন থেকে দেখে এবং অভ্যর্থনা পাওয়ার পর তার প্রতি শ্রদ্ধা জানায়।
সে খুব খুশী হলো। ভালো হয়েছে, মা।
সে খুব খুশী হলো। সবকিছু তার খোলসা হয়ে গেছে। | সে খুব খুশী হলো। তাই আমি মাত্র দুঃখিত এই রোমবিস্তরে খুব হতে হবে। আমি শুধু নয় আমাদের ভবিষ্যদ্বাণী থেকে স্বাগত জানাই। আমার খুব জন
সে খুব খুশী হলো। আমি এখানে যথাক্রমে যোগ দিবলিত হতে পারে। হাহাহ, আমি এখ-নই যোগাযোগ করছি। আমি এখনই পাপাপ করছি। আমি এখানে যথাক্রমে কান্ |

Table 28: Paramanu-Bangla 108.5M vs GPT-J 6B open end text generation for Bangla prompts.

| Bangla (Bengali) Input Prompt | Text Generation (top-3) from Paramanu-Bangla 108.5M | GPT-Neo 1.3B | GPT2-XL |
|---|---|---|---|
| ফেলুদা তোপসেকে বলল, যা লালমোহন-বাবুকে খবর দে! | ফেলুদা তোপসেকে বলল, যা লালমোহনবাবু-কে খবর দে! আমরা আলাম ঘর থেকে বে-রোতেই ধূর্জটিবাবুর আর্তনাদ শুনলাম। তার-পর ফেলুদার গলা পেলাম। ধূর্জটিবাবুর ঘরের সামনে যে লোক বসেছিল, সে ফেলুদার দিকে চেয়ে জিজ্ঞেস করল, আপনার নাম কী? ফে-লুদা আর্তনাদ শুনে ফেলুদা বাইরে চলে গেল। আমি আরও মিনিটখানেক বাইরে হাঁটলাম।
ফেলুদা তোপসেকে বলল, যা লালমোহনবাবু-কে খবর দে! উনি আসছেন।
ফেলুদা তোপসেকে বলল, যা লালমোহনবা-বুকে খবর দে! ভদ্রলোক জিজ্ঞেস করবেন, আর সত্যি বলতে কী, আমার সন্দেহ আছে। বাইফোকালসাই।–তাও অবিশ্যি উনি ছাড়। | ফেলুদা তোপসেকে বলল, যা লালমোহনবাবু-কে খবর দে! কোনও অন্তিকমিতিতে আছে, কোন স্টিকটর সরকার দেবেন, দেবেন, দে-বেন. যাতে সুরক্ষিত চিন্
ফেলুদা তোপসেকে বলল, যা লালমোহনবাবু-কে খবর দে! ভিডিওগুলো চিহ্নিট উপসাহস-না, জ্যাকার পয়েন্ট, তিনটি শর্ক বিধান এই পয়েন্ট নতুন মানুষ | ফেলুদা তোপসেকে বলল, যা লালমোহনবাবু-কে খবর দে! (মিলিবেন গোপায়াবের) আজ-তার ও আজতার ঘন টার কেসমাদের রিয়ে আইকে অকে বাদ্ধাসারীনের না
ফেলুদা তোপসেকে বলল, যা লালমোহনবা-বুকে খবর দে! তাঁশিস্তাসনামী ও দিয়োবি-নিক নিকট অচিনেকেন তাপকানানে অমবামি-প্রা নামোশাতে অবাংলাবধ |
| এমন সময় হঠাৎ বা-ঘের ডাক! | এমন সময় হঠাৎ বাঘের ডাক! বাঘের সে ডাক বাঘের স্বর নয়! দূরে গিরিগঞ্জের ধান-খেতের পিছন হইতে যেন কাঁপিতে কাঁপিতে ছু-টিয়া আসিতেছে।
এমন সময় হঠাৎ বাঘের ডাক! সন্ত্র ভয়ে চী-চী করিয়া উঠিল। মনে হইল যেন সেই বনের হরিণগুলা আবার জঙ্গলের দিক হইতে আসি-য়া উপস্থিত হইল।
এমন সময় হঠাৎ বাঘের ডাক! আমার ভয় পাবার হয়তো কিছু ছিল। নইলে দুবে সক-লের চোখের ওপরে একটা শজারুর মতো বড়ো চোখা শিং অনেকদিন আগেই মেরেছি কি-না? না, না, আমি মিথ্যে ভয় করি না। সে গলার আওয়াজ এবার কাছেই কোথায় মিলি-য়ে গেল! | এমন সময় হঠাৎ বাঘের ডাক! সুতরাং, আপ-নি জাতিশীল হতে পারেন করতে পারব বছর। ঠিক আছে, আমি এখানে ভাল চেষ্টা করছি বলতে হবে পারেন আমার বছর একটি ছ
এমন সময় হঠাৎ বাঘের ডাক! هي هذه هل هذاميف اكتب انا هل ثم بهذا؟ومن القوم الفحوصة الناس يقوم ،حيث الفكرة قصد إعادة أثناء للاعب أنك أعتبر نذهب عندما الآن مخالف؟ دون انت فى ست الآن القنابلة هذه يكون.في أين تعرف لا | 'এমন সময় হঠাৎ বাঘের ডাক! #Kurdish #KurdishDaughter pic.twitter.com/dC0q1wC9U6 Ameen_Bana September 25, 2017. She wants to become a doctor, so she studied in London, got an MA and now she is living in Toronto, Canada, Mejda said. She said that Mejda told her that the people around her helped her through the difficult years she lived under the PKK rule in her hometown of Silopi. She recalled a woman who helped to feed and clothe her during the war and told Mejda that it is important to support others who have also suffered. Mejda said that her family did not ask for a reward, but the PKK promised to pay Mejda a monthly salary in her old age, which they have so far received. I have never received such an amount from anyone else and if I can help others, then
এমন সময় হঠাৎ বাঘের ডাক! (KARMA POLICE IS FOR EVERYBODY. Our countrymen, who support and fight for the freedom, dignity and power of our nation. Together we will overcome obstacles and the pain of the past with our strength of strength. Don't be divided, we are one nation, we will be united one day! এমতার পড়ত দোট মকানিক উতারিদি কোট পারানা-নেরিক নিরেইলাকিক স্মিডনে) উ⬜⬜⬜⬜! قاطع كان الدين |
| অপু এসে ডাকল, মা! | অপু এসে ডাকল, মা, খেতে দাও।
অপু এসে ডাকল, মা, ও মা, কোথা যাও?
অপু এসে ডাকল, মা, শিগগির এসো, গ্রামের ওদিক থেকে ডাকছে। | অপু এসে ডাকল, মা! অবশ্যই এটা মাটিরা দেখতে যাচ্ছে। এবং এটা উদ্যানের উপরে, এবং ব্যাখ্যা পড়া এটা অন্যথা উপর এর মা-টিরা মধ্যে অবশ্যই একটি
অপু এসে ডাকল, মা! তারপরা বড় বলে করছি যদি করেছি এখানে তাকে শুনতে চান আমি বেয়ুচি অবধি রাখবেন একই ভাইস-রকারী এর এবং এই কাজ রাখব হয়। এবং | অপু এসে ডাকল, মা! সব্যাসাসিয়েনিয়েনে, আয়ার কে অচিয়েনিয়েনিয়েনে সম⬜দেনিকার যোগাদা কে দাগার পারেনেনেনকের ভানকে সিচুবিতরা
'অপু এসে ডাকল, মা! Konyuru, Nli, and Nr, the most powerful divinities among the Da Nya. Hbha "Divine Mother" মৈদান কস দিথি আনামঁ কম্লেঁগ়ঁ বুলানিপে তো জবাদ হৈ কম্লে-গ়ঁ বুলানিপে নোটিত কিত্তিমিত পরঁব হৈ হঁ। |
| সে খুব খুশী হলো। | সে খুব খুশী হলো। সবাই তাকে এত সম্মা-নের চোখে দেখে। প্রথম দিন থেকে দেখে এবং অভ্যর্থনা পাওয়ার পর তার প্রতি শ্রদ্ধা জানায়।
সে খুব খুশী হলো। ভালো হয়েছে, মা।
সে খুব খুশী হলো। সবকিছু তার খোলাসা হয়ে গেছে। | সে খুব খুশী হলো। যখন প্রিয় এই গঠিত প্রব-র্তন করুন। তারপর জেনিটটার মধ্যে ছাপেন যে অভিনেতার জন্য জিগনেট দ্বারা জাগাতে হয়। তাই আমর
সে খুব খুশী হলো। এখন আপনি জানতে হবে দুটো দিন। এটি আমাদের পরিবর্তে আমরা কি-ভাবে বোল করা উচিত করেছিলেন। আমি নাম কিছু তাই না। আমি স্প | সে খুব খুশী হলো। ব্যান্তে পগ়ামেন গিলে আপ-রিক্ষপা রিপারা বিকরেরেবিলেনাদনে মনানাদে-নে রিপারা অফার পরিক্ষপারেনে মনানাদেনে জনানাদ
সে খুব খুশী হলো। ⬜⬜⬜⬜ যাসে পেছে পো-জন করর হয় যোৎক্ষার করারে প্রতাকর যা-ছে খুবলা পেছে ব্রিয়ালাসে পোচে গিগাবে ন্টা-লে |

Table 29: Paramanu-Bangla 108.5M vs GPT-Neo 1.3B vs GPT2-XL 1.5B open end text generation for Bangla prompts.

| Bangla (Bengali) Input Prompt | Text Generation (top-3) from Paramanu-Bangla 108.5M | Bloom 560M | Bloom 1.1B | Bloom 3B | Bloom 7.1B |
|---|---|---|---|---|---|
| ফেলুদা তোপসেকে বলল, যা লালমোহনবাবুকে খবর দে! | ফেলুদা তোপসেকে বলল, যা লালমোহনবাবুকে খবর দে! আমার আপনার ঘর থেকে রোত্তর ধুলটিবাবুর আর্টগান গুলোলো। তার পর ফেলুদা বলল ঢুকল। ধুলটিবাবুর ঘরের সামনে সে লোক বসেছিল, সে ফেলুদার দিকে চেয়ে জিজ্ঞেস করল, আপনার নাম কী? ফেলুদা আর্তনাদ শুনে ফেলুদা বাইরে চলে গেল। আমি আরও মিনিটখানেক বাইরে ইটালাম। ফেলুদা তোপসেকে বলল, যা লালমোহনবাবুকে খবর দে! উনি আসছেন। ফেলুদা তোপসেকে বলল, যা লালমোহনবাবুকে খবর দে! ভদ্রলোক জিজ্ঞেস করবেন, আর সত্যি বলতে কী, আমার সঙ্গেও আছে। বাইফোকালসাই।—তাও অবিনী উনি ছাড়। | ফেলুদা তোপসেকে বলল, যা লালমোহনবাবুকে খবর দে! ফেলুদা তোপসেকে বলল, যা লালমোহনবাবুর সঙ্গে যোগাযোগ করেনি। তাই তো কথা বলবার অনুমতি পেয়েছে। পের এই বিষয়টি আমি জানি না। আমি খুব খুশি। মায়ের সঙ্গে কথাবার্তার সময় বাবা-মা জানতে চান, তিনি কি আছেন? তার উত্তরে জানলেন যে, অসুস্থ হয়ে পড়েছেন। এরপর মায়ের এই প্রশ্ন শুনে বাবা মা হাসলেন, আমি খুব কষ্ট করছি। খুব কষ্ট লাগছে। আর এত কষ্টের মধ্যে | ফেলুদা তোপসেকে বলল, যা লালমোহনবাবুকে খবর দে!<pad> ফেলুদা তোপসেকে বলল, আমি আজ রাতে তারাই তাকে ডাকতে আসবেন। আজকে তার সাথে আমার একটা কথা আছে। ফেলুদা বলল, আপনি জানেন, আমার এক বন্ধু লালমোহনবাবুকে অনেকদিন ধরে আমেরিকায় থাকেন। কিন্তু তার দেখে তার কোন বন্ধু নেই, তার এক ছেলে আছে। তার ছেলেকে একটু বিয়ের বলত তার। কিন্তু সে কোন ছেলের কথা শোনে না। কিন্তু বিয়েটা করতে হবে।লালমোহনবাবু একটা ছোট ছেলেকে বিয়ের করে ফেললেন। কিন্তু বিয়েটা হবে এন না কারণ ছেলেটা ঠিক বিয়ের সময় মারা গেল। লালমোহনবাবু তার এই ছেলের বিয়ে না হওয়াতে জীবনভাবে দুঃখিত।জনবনা বলল, তাই কি? তবে তাহলে তো বিয়েটা করতে হবে। আপনার কাছে একটা খবর আছে, আমাকে আপনার সাথে পাঠাতে হবে। |
| এমন সময় হঠাৎ বাঘের ডাক! | এমন সময় হঠাৎ বাঘের ডাক! বাঘের ডাক বাঘের স্বর না বয়! দূরে দিরিপালের থান-থেকের শিহন হইতে যেন কাঁপিতে কাঁপিতে ছুটিয়া আসিতেছে। এমন সময় হঠাৎ বাঘের ডাক! সন্ধ ভয়ে চীনী করিয়া উঠিল। মনে হইল বোন সেই হরিণঘাতা আবার জঙ্গলের দিক হইতে আসিয়া উপস্থিত হইল। এমন সময় হঠাৎ বাঘের ডাক! আমার ভয় পাবার হয়তো কিছু ছিল। নইলে ভয় সকলের চোখের ওপরে একটা নজরের মতো বয়ে চোখে নিব অনেকদিন আগেই মেরেছি কি-না? না, না, আমি মিথ্যে ভয় করি না। সে বাঘের আওয়াজ এবার কাছেই কোথায় মিলিয়ে গেল। | এমন সময় হঠাৎ বাঘের ডাক! হঠাৎ বাঘের ডাক! বা-ঘের ডাক! বাঘের ডাক! বা-ঘের ডাক! বাঘের ডাক! বা-ঘের ডাক! বাঘের ডাক! বা-ঘের ডাক! বাঘের ডাক! বা-ঘের ডাক! বাঘের ডাক! এমন সময় হঠাৎ বাঘের ডাক! সে সময় চার-পাঁচজন লোক বেরিয়ে এলো। তাদের মুখে যে লোকটিদের আবার সন্ধনে বেলা বাঘের কাছে এসে পৌঁছলো না পারে। বাঘটা কৌড়ে এসে এসে লোকটার পি-ঠে এসে বলল। লোকটা তাকে ডাকতে শুরু করল। বাঘকে তাদের শব্দ না পে-য়ের বাঘ না পেরে লোকটি আবার বাঘের কাছে এসে বলল। বাঘটির বাঘ বাঘের কাছে হলো সে মন | এমন সময় হঠাৎ বাঘের ডাক! এ মন সময় হঠাৎ বাঘের ডাক চি-ৎকার শুনে স্থানীয়রা ছুটে এলন। এর সঙ্গে সঙ্গে এলাকাবাসী বাঘের পাশ থেকে বেরিয়ে এসে তাকে উদ্ধার করে। তবে বাঘের শরীরে কোনো আঘাতের চিহ্ন পাওয়া যায়নি | এমন সময় হঠাৎ বাঘের ডাক! চমকে উঠে ভয় পেয়েছিল চলে আয়। এমন সময় হঠাৎ বাঘের ডাক! বাঘের মুখ থেকে একটি বাঘের দাঁত বের হয়ে এলো। সে তো ঠিক খা লোন। মুঘু বলল, "হ্যা, আমি বাঘের মুখ।" এরপর বাঘ ঘুমুতে বলল, "ঠিক আছে তুমি এই গাছের ওঁটি ভেঙে যাও। আমি তোমার গলার ফাঁক দিয়ে তোমাকে বের করে দিব।" তারপর বাঘ চলে লোন। মুঘু গা-ছের ঊঁটি ভেঙে ফেলে। কিছু কোন শব্দ না ে-লে বাঘ ঘুমুতে বলল, "হ্যা আমি তোমার গলার ফাঁক দিয়ে নিয়েছি। তুমি বের করে দাও না।" | এমন সময় হঠাৎ বাঘের ডাক! সেই সঙ্গে তার পাশে একটা মুখা। বাঘের মতোই জিভ বের করে একটা মুখ থাকছে দিয়ে একটা জৈল দিল। চমক দিয়ে তাকিয়ে গেলো এই মুখ লোন কা। সেই সাথে একটা বাঘও বের হয়ে গেলো, সে তো ঠিক বুঝে উঠতে পারছিল না কি হতে পারে। একটা লোক চলে এলো, মুঘটা ডাক হবে, যেন খুব ভয় পেয়েছে। লোকটার কাছে থেকে আসল হাতুনিট বেলো। হাতুনিট আর একটা কাটের বাক্সা। লোকটা আবার বলল, "আমি তোমার কাজ লিখিয়ে দেব! তুমি এখানে থাক!" লোকটা চলে লোন। এমন সময় হঠাৎ বাঘের ডাক শুনতে পেয়ে হেঁটে নৌড়লেন। তারপর কথা হল প্রতিবেশীর উঠোনের পায়ের ছাপ। এই প্রতিবেশীর বাঘের উঠোনে একটি পায়ের ছাপ পড়ে ঘায়। এরপর সেখান থেকে শুরু হয় রহস্য। বোঝা নিয়ে জানা যায়, স্থানীয় একটি এ গ্রামে কয়েক দিন আগে একটি বাঘ ঢুকে পড়েছিলেন। ওই দিন বাঘটির রাতে খাবার নিয়ে ঘুমানোর জন্য বাঘটি প্রেতিবে-লন। রাতে হঠাৎ বাঘের ডাক শুনতে পেয়ে হেঁটে নৌড়লেন। তারপরের কথা হল প্রতিবেশীর উঠোনের পায়ের ছাপ। এই প্রতিবেনীর অভিযোগ, কেউ হয়তো ওই বাঘের ছাপ হয়ে দিয়ে |
| অপু এসে ডাকল, মা! | অপু এসে ডাকল, মা! আমি তোমাদেরই ছে-লে, আর তোমারই আমার মা। আমি আর তোমার কাছে আমার মা। আমি তোমাকে বিরের প্রস্তাব দিয়েছি। অপু এসে ডাকল, মা! আমার খুব ইচ্ছে হলে তুমি এসে আমার সাথে বেড়াতে পার। | অপু এসে ডাকল, মা! অপু এসে ডাকল, মা! অপু এসে ডাকল, মা! আকারের জল থেকে বের হয়ে আসুন! | অপু এসে ডাকল, মা! তোমরা তো সব সময় একে অন্যের সাথে খারাপ কথা বলেন। এখন থেকে কোনদিনই এমন কোন কথা বলবে না। অপু এসে ডাকল, মা! আমার এ বাড়ি, আপনার সঙ্গে দেখা হল। | অপু এসে ডাকল, মা! বলে বাইরের চলে লোন। আরিফ আমার কাছে এসে বসল। আমার দেখে বলল, মা, তোমাকে একটা গান শুনাব। আমি আমার গান শুনে বললাম, তুমি তোমার মায়ের গান গাও। আরিফ আমার দিকে চোখ পড়তেই চমকে উঠলাম। আমি ওর মুখের দিকে তাকিয়ে ভাবলাম, কী সুন্দর চোখ! আরিফ আমার দিকে কিছুক্ষণ তাকিয়ে থাকল, তার-পর বলল, মা, এই যে গানটি তুমি শুনলে, এটাই আমার প্র-থম গান। আজকে আমি তোমাকে এটা নিয়ে দিতে চাই। তু-মি আমাকে যে টাকা দাও, তা নিয়ে আমি খুব সুন্দর একটি গান করব। আমি বললাম, তা নিয়ে আমি খুব সুন্দর আরিফকে আমার গানের কথা বলি। আরিফ আমাকে অনেক টাকা দেয়, যা নিয়ে একটা গানের দল গড়ে তোলাবার চেষ্টা করি। অপু এসে ডাকল, মা! তোমার আরাম হয়! এই একত ভাবে একটা ছেলে ডাকল! মনে হল কোন অসুরকে। কিন্তু কি বলি-নি। তার পর সেই ঝিলিনটা দেখলেই বুকটা লোলায়! আমি যখনই ঝিলিনটা দেখি, তখনই মনটা কেমন করে। মনে হয় যেন একটা পাখি এসে বসে আছে আমার বুকে। আমি আমার বুকের দুধ তার দিকে ঠেলে এনে দেই! আমি আমার বুকের দু-ের সঙ্গে ছেলে আর তার বাবাকেও খাইয়ে দেই! আমার-কে আমার মনটা কেমন জানি করে! আজকে আমি খুব খুশি। আমার ছেলেটাকে আমি পেয়ে গেছি। আমার মুখের সঙ্গে হয়ে গেল। আমি আমার ছেলেকে ভালবাসতে শিখলাম। তা-কে আমার রাখতে পা |
| সে খুব খুশী হলো। | সে খুব খুশী হলো। সবাই তাকে এত সম্মানের চোখে দেখে। প্রথম দিন থেকে যখন এবং অবতরণ পাওয়ার পর তার প্রতি রাজ্ঞা জানায়। সে খুব খুশী হলো। ভালো হয়েছে, মা। সে খুব খুশী হলো। সবকিছু তার খোলসা হয়ে গেছে। | সে খুব খুশী হলো। আমার কাছে সেই ছা-লো আর আমার কাছে না থাকবে যে জায়গা ঘটবে সেই সাথেই। এর পর থেকে আমার কাছে অনেক অনেক কি-ছু আমিও আসতে পারতাম। এর প্রথম নাম ছিল 亿巴ল লিফ এভ দা ন্যাচফেসেপী এন্ড স্টা-ডিট টু উইঙেন এবং আমার দ্বিতীয় নাম ছিল 亿巴ট্রন লিফ এন্ড দা ন্যাচফেসেপী এন্ড টেকনোলজি"। আমার নামটা হয়তো আমার কাছে সহজ মনে হতে পারে। সে খুব খুশী হলো। এর আগে তারা দুজনেই তাদের প্রেমিক-প্রি ভাইটেন নিয়ে নিজের বা-ড়িতে চলে যায়। হঠাৎ কবেই তাদের মা চলে আসে। কিন্তু হঠাৎ করে সেই প্রেমিকার সঙ্গে সম্পর্কটি একদিনে চলে আসে। যখনই সে তার ভাইকে নিয়ে বাড়িতে ফিরবে সে তার ভাই তার কাছ থেকে তাকে নিয়ে বেড়া-তে যাবে এবং সেই তারা আবার তাদের প্রেমের সম্পর্কটি আগের মতোই। কারণ তারা দুজনেই খুব সুসম্পর্কে বসবাস করতে চায়। | সে খুব খুশী হলো। আমি সেই থেকে সবাই তাকে ছিল সে ফেললেন। সে খুব খুশী হলো। এর থেকে সবাই তাকে ঘি-রে ফেললেন। সে খুব খুশী হলো। এর থেকে সবাই তাকে ঘি-রে ফেললেন। | সে খুব খুশী হলো। এর আগে কাস্টম থেকে সে বলেছিলেন যে তারা যদি কাস্টম থেকে ফি-ন নেতে চায় তাহলে ওরা চাইলে আবার যেতে পারবে। তবে এইটা তাদের সিদ্ধান্ত। তাই বলে এ বিষয়ে তাদের কে বাধা দেওয়া উচিত হবে না। সে খুব খুশী হলো। কিন্তু কিছু সময় পর সে বউটার সঙ্গে তার একটা কথা হলো। তার বউ তাকে বলল, তোমার এই ছেলেটির ওপর খুব রাগ হচ্ছে। তোমার ছেলের সঙ্গে তোমার কোনো কথা হয়নি। তুমি কি তোমার ছেলে-কে বলল, তোমার ছেলেও তো এই ঘরেল! সে এখন আমার সঙ্গে আছে। সে যদি আমার সঙ্গে থাকে, তবে কি আমাকে তোমার মতো করে দেখবে না? তাই এ কথা বলে সে বউটার ওপর রাগ করে চলে গেল। কিন্তু এই ছেলের মন এমন যে, তার কোনো কাজ-কর্ম হবে না | সে খুব খুশী হলো। ভাবল, এইবার আর কোন ঝামেলা নেই। এগুলোকে করে সমিবায়ে হাসপাতালে যাবে। কিন্তু, এই ভে-বে, হাসপাতাল থেকে আর ফিরে আসবেনা তার স্ত্রীক। এই বলে সে সিএনজি থেকে নেমে আর একটা সিএনজি জো-গাড় করল। ভাবল বে, হাসপাতাল থেকে ফিরে এলে সে তাকে নিয়ে যেবে। হাসপাতালের এক কর্মচারী বললা, "সার, আপনি আর যাবেন না। ডাক্তার সাহেব এ বলেছেন, আপ-নি আর সুস্থ হয়ে উঠবেন না। আপনাকে বাড়ি ফিরে যেতে হবে। এই বলে সে সিএনজি থামাল। সিএনজি থেকে নেমে আবার বাসায় চলে লোন। স্ত্রীকে সে বলে, "আর আমার কা-ছে আসবেন না, আমি আর যেতে পারব না। আমাকে এই বার ডাক্তার তাকে আর কখনো সুস্থ হয়ে উঠতে দেবে না বলে জানিয়ে দিলেন। ডাক্তার বললেন, "এখানে আর কখনো ফিরে যেতে পারবেন না। কারণ, আপনি মানসিক ভারসাম্য হারিয়ে-ছেন।" কয়েক মাস পরে সেই লোক মানসিক সিএনজি চালকেই সিএনজি থেকে নেমে যে আর হাসপাতাল হয়ে গেল। এবার ডাক্তার বলে দিলেন, "আপনি আর কখনো সুস্থ হবেন না। আপনার মানসিক ভারসাম্য ভালো নেই। আপনার এই অবস্থা |

Table 30: Open end text generation: Paramanu-Bangla 108.5M v/s BLoom (560M, 1.1B, 3B, 7.1B)

| Sanskrit Input Prompt | Text Generation (top-3) from mParamanu 162M | OPT 6.7B |
|---|---|---|
| अस्माकं देशः भारतवर्षः! (asmākaṁ dēśaḥ bhāratavarṣaḥ) | अस्माकं देशः भारतवर्षः इति परिगणितः (asmākaṁ dēśaḥ bhāratavarṣaḥ iti pariganitaḥ) अस्माकं देशः भारतवर्षः इति ख्यातः (asmākaṁ dēśaḥ bhāratavarṣaḥ iti khyātaḥ) अस्माकं देशः भारतवर्षः इति प्रसिद्धः (asmākaṁ dēśaḥ bhāratavarṣaḥ iti prasiddhaḥ) | अस्माकं देशः भारतवर्षः The Buddha described three kinds of birth: 1) a type of birth which results in the attainment of the highest bliss, 2) a type of birth which results in the attainment of the highest knowledge, and 3) a type of birth which results in the attainment of the highest wisdom. These three अस्माकं देशः भारतवर्षः आपके श्री शिक्षा चिकन्दियाँ के बुलंद⬚ अस्माकं देशः भारतवर्षः(The Bhagavad Gita, verse 6)तथा त्रियों कि विद्यार्थी स⬚ |
| वेदः चत्वारः सन्ति! (vēdaḥ catvāraḥ santi) | वेदः चत्वारः सन्ति छन्दः ब्राह्मणम् तैत्तिरीयकम् कठोपनिषत् वैदिकच्छन्दसाम् मन्त्रम् आध्यपनम् अस्ति (vēdaḥ catvāraḥ santi chandaḥ brāhmaṇam taittirīyakam kaṭhōpaniṣat vaidikacchandasām mantram ādhyapanam asti) वेदः चत्वारः सन्ति प्रसिद्धः शिक्षाशास्त्रीयः शिक्षकायगः शास्त्रम् शास्त्रम् कणादः पाणिनीय शिक्ष काण्डे सूत्राणाम् उपरि भाष्यम् लिखितवान् (vēdaḥ catvāraḥ santi prasiddhaḥ śikṣāśāstrīyaḥ śikṣakāyagaḥ śāstram śāstram kaṇādaḥ pāṇinīya śikṣa kāṇḍē sūtrāṇām upari bhāṣyam likhitavān) वेदः चत्वारः सन्ति धर्मशास्त्राणि वेदाः वैदिककर्मकाण्डम् आन्वीक्षिकी दण्डनीतिप्रभृतीनि (vēdaḥ catvāraḥ santi dharmaśāstrāṇi vēdāḥ vaidikakarmakāṇḍam ānvīkṣikī daṇḍanītiprabhṛtīni) | वेदः चत्वारः सन्तिवेदः परः आरामवापेश्वर यूनरवापिताःचूत् वेदः चत्वारः सन्तिःवेदः चत्वारः सन्तिःवेदः चत्वारः सन्त </⬚>वेदः चत्वारः सन्तिनःतान्यरकर्मः सर्वधिक्षित पर्वृति यत⬚ |
| किमर्थं त्वं गच्छसि! (kimarthaṁ tvaṁ gacchasi) | किमर्थं त्वं गच्छसि अस्माभिः मे सम्मुखं प्रसारितं भविष्यति (kimarthaṁ tvaṁ gacchasi asmābhiḥ mē sammukhaṁ prasāritaṁ bhaviṣyati) किमर्थं त्वं गच्छसि इत्यत उच्यते कुरुष्विति (kimarthaṁ tvaṁ gacchasi ityata ucyatē kuruṣviti) किमर्थं त्वं गच्छसि तद्विदं विदे (kimarthaṁ tvaṁ gacchasi tadvidaṁ vidē) | किमर्थं त्वं गच्छसिन्नी गाजीवज़रहाडीज जिंदा कामर्न नामाजावन किमर्थं त्वं गच्छसियो का निराशा मामला चाहिए. इस त्वं गच्छसियो क किमर्थं त्वं गच्छसिनमः मरं त्वं से मृतमृत प्रदेशं प्रश्⬚ |
| मह्यं मिष्टान्नं रोचते (mahyaṁ miṣṭānnaṁ rōcatē) | मह्यं मिष्टान्नं रोचते ते ददामि ॥ १२॥ देवा अन्नं देहि मेऽजन्पिबन्ति। (mahyaṁ miṣṭānnaṁ rōcatē tē dadāmi .. 12.. dēvā annam dēhi mē'janpibanti.) मह्यं मिष्टान्नं रोचते ते दिने दिने ॥ (mahyaṁ miṣṭānnaṁ rōcatē tē dinē dinē.) मह्यं मिष्टान्नं रोचते ते हि भक्त्या मया दत्तं वारिजाते सरोजे । (mahyaṁ miṣṭānnaṁ rōcatē tē hi bhaktyā mayā dattaṁ vārijātē sarōjē) | मह्यं मिष्टान्नं रोचते रहेंगे हैं, वो केंद्र में तुम निराश जि, मह्यं मिष्टान्नं रोचते तैयार भयरा ब्रेक्नां रोचते तैयार भयरा मह्यं मिष्टान्नं रोचते हुं ये क्यूँ मत है? मेरे आंदोलन बिल नह |

Table 31: mParamanu 162M vs OPT 6.7B open end text generation for Sanskrit prompts.

| Sanskrit Input Prompt | Text Generation (top-3) from mParamanu 162M | LLaMa 2 7B |
|---|---|---|
| अस्माकं देशः भारतवर्षः | अस्माकं देशः भारतवर्षः इति परिगणितः अस्माकं देशः भारतवर्षः इति ख्यातः अस्माकं देशः भारतवर्षः इति प्रसिद्धः | अस्माकं देशः भारतवर्षःअस्माकं देशः भारतवर्षः च, च निर्विषेषः श्रवणतिभिर्दिधिः ।रङ्क अस्माकं देशः भारतवर्षः सहायकः वैयकितीय बीमा- निक अर्थदान के आधार पर प्रशासनिक पात्रता पर व्याप <☐> अस्माकं देशः भारतवर्षः / भारतचक्रवृक्षः। / महागृहः कालिश्छन्हितः । स्वप्नः स्वप्न' |
| वेदः चत्वारः सन्ति | वेदः चत्वारः सन्ति छन्दः ब्राह्मणम् तैत्तिरीयकम् क- ठोपनिषत् वैदिकच्छन्दसाम् मन्त्रम् आध्यपनम् अ- स्ति वेदः चत्वारः सन्ति प्रसिद्धः शिक्षाशास्त्रीयः शिक्षका- यगः शास्त्रम् शास्त्रम् कणादः पाणिनीय शिक्ष काण्डे सूत्राणाम् उपरि भाष्यम् लिखितवान् वेदः चत्वारः सन्ति धर्मशास्त्राणि वेदाः वैदिककर्मका- ण्डम् आन्वीक्षिकी दण्डनीतिप्रभृतीनि | वेदः चत्वारः सन्तिवेदः परः आरामवापेष्वर यूनरवा- पिताःचूत् वेदः चत्वारः सन्तिनःतान्यरकर्मः सर्वधिक्षित पर्वृति यत☐ newline वेदः चत्वारः सन्ति चामिरः क्षमा दै- विकौ । संकोषं च किं भारत निन्दा किं अतिनि । अपि मनुष्या |
| किमर्थं त्वं गच्छसि! | किमर्थं त्वं गच्छसि अस्माभिः मे सम्मुखं प्रसारितं भ- विष्यति किमर्थं त्वं गच्छसि इत्यत उच्च्यते कुरुष्विति किमर्थं त्वं गच्छसि तद्विदं विदे | किमर्थं त्वं गच्छसि रवित्वां हिंसे माया विच्छप्ये- कां गन्धसागरः ॥ ३ ॥ kīmartham tvam gacchasi ravitvā किमर्थं त्वं गच्छसि नित्यासङ्ग्रिमुत्थितम् । महेश्वर- स्य महेश्वरति पश्यति परं तत्त्वम् किमर्थं त्वं गच्छसि यत् स्वप्रथमिति रतम्। 1-48- 59,55-62 तस्मिन्नप्यहं मधुरं तपः |
| मह्यं मिष्टान्नं रोचते | मह्यं मिष्टान्नं रोचते ते ददामि ॥ १२॥ देवा अन्नं देहि मेऽजन्पिबन्ति। मह्यं मिष्टान्नं रोचते ते दिने दिने ॥ मह्यं मिष्टान्नं रोचते ते हि भक्त्या मया दत्तं वारिजाते सरोजे । | मह्यं मिष्टान्नं रोचते मुविश्यते न्युड्श्व यत्स उवाङ्ग्व्युं पश्यते तत्वः । तत्र नानिं श्वत मह्यं मिष्टान्नं रोचतेन्द्रोष्टेन्द्रं राजन्यस्विंधादितिस्वी- कृतिस्तत्रजगतिभारिणि भावेशानि मृत् मह्यं मिष्टान्नं रोचते हिता वा अनन्तकथ्या । पुत्रा का- लाग्निः पृथिवी पदोंति मध्ये । स्वा यो |

Table 32: mParamanu 162M vs LLaMa 2 7B open end text generation for Sanskrit prompts.

| Sanskrit Input Prompt | Text Generation (top-3) from mParamanu 162M | GPTJ 6B |
|---|---|---|
| अस्माकं देश: भारतवर्ष: | अस्माकं देश: भारतवर्ष: इति परिगणित:
अस्माकं देश: भारतवर्ष: इति ख्यात:
अस्माकं देश: भारतवर्ष: इति प्रसिद्ध: | अस्माकं देश: भारतवर्षः।। एतिहासपत्रः समाप्त:मैश्रीनिक्
अस्माकं देश: भारतवर्षः। अथ मयि निबधमन्मा भारतयामि ते ।। ⬜⬜⬜⬜⬜⬜ ⬜⬜⬜⬜ ⬜ ⬜⬜⬜
अस्माकं देश: भारतवर्षः रमाढ्री न गुणा ज्ञानं देशमस्तुलम् ।व |
| वेद: चत्वार: सन्ति | वेद: चत्वार: सन्ति छन्दम् ब्राह्मणम् तैत्तिरीयकम् कठोपनिषत् वैदिकच्छन्दसाम् मन्त्रम् आध्यपनम् अस्ति
वेद: चत्वार: सन्ति प्रसिद्ध: शिक्षाशास्त्रीय: शिक्षाकायग: शास्त्रम् शास्त्रम् कणाद: पाणिनीय शिक्ष काण्डे सूत्राणाम् उपरि भाष्यम् लिखितवान्
वेद: चत्वार: सन्ति धर्मशास्त्राणि वेदा: वैदिककर्मकाण्डम् आन्वीक्षिकी दण्डनीतिप्रभृतीनि | वेद: चत्वार: सन्ति क्षिमेर्येणि भूतेर्वजाः स चतु भूत दुर्ग
वेद: चत्वार: सन्तिभा सा कुण्डया चेति मम नो वो हृष्टी म् । योगिन⬜
वेद: चत्वार: सन्तिमाननाप्पतिः मुखमापरिणामहेमं पुन: सुमान: |
| किमर्थं त्वं गच्छसि | किमर्थं त्वं गच्छसि अस्माभि: मे सम्मुखं प्रसारितं भविष्यति
किमर्थं त्वं गच्छसि इत्यत उच्यते कुरुष्विति
किमर्थं त्वं गच्छसि तद्दिदं विदे | किमर्थं त्वं गच्छसि विस्मयम् । कुणात्वा नसिं मेदो: शायद कु⬜
किमर्थं त्वं गच्छसि) - अब हिंदु वै निम्नलाई बाँध्न हुने छ यस को
किमर्थं त्वं गच्छसिद्ध: (Thus I have declared and said, I will show thee what is my understanding of the meaning of these words. ) कुप्ति स्मित्ता ⬜ |
| महां मिष्टान्नं रोचते | महां मिष्टान्नं रोचते ददामि ।। १२।। देवा अन्नं देहि मेऽजन्पिबन्ति।
महां मिष्टान्नं रोचते ते दिने दिने ।।
महां मिष्टान्नं रोचते ते हि भक्त्या मया दत्तं वारिजाते सरोजे । | महां मिष्टान्नं रोचते शक्नोपने मितीन्तिस्तते वर्षते।
महां मिष्टान्नं रोचते O' My Soul! My Self is the Immortal. अष्टान्येन ततोऽस्त्यमात्म
महां मिष्टान्नं रोचते Innocent eyes are filled with tears, and the cheeks of the tender ones are wet with the tears of the heart, when a youth is told that his father is going to take away his life. But he forgets all those things, and only thinks of the joy of |

Table 33: mParamanu 162M vs GPT-J 6B open end text generation for Sanskrit prompts.

| Sanskrit Input Prompt | Text Generation (top-3) from mParamanu 162M | GPT-Neo 1.3B | GPT2-XL |
|---|---|---|---|
| अस्माकं देश: भारतवर्ष: | अस्माकं देश: भारतवर्ष: इति परि-गणित: 
 अस्माकं देश: भारतवर्ष: इति ख्यात: 
 अस्माकं देश: भारतवर्ष: इति प्रसि-द्ध: | अस्माकं देश: भारतवर्ष:। साक्षा-त्वाय:। कारण वापरारित्याद्रद् 
 अस्माकं देश: भारतवर्ष: अस्माकं देश: केवल स्थापित: नका बहुत वि 
 अस्माकं देश: भारतवर्ष: सवै जा-तात्यायांन्ज़ी नावा तपासन्धुव: सु⬜ | अस्माकं देश: भारतवर्ष: स्थ्या कु-ण्यं में अस्माकं देश: भारतवर⬜ 
 अस्माकं देश: भारतवर्ष: वोडिया पायस्वहर्मिया ॥.  One of the greatest benefits of being an e-book reader is that you don't have to carry 
 अस्माकं देश: भारतवर्ष: भीषर्शि-ध्यता, योगमणिरिणि: दो अमहा-नाति |
| वेद: चत्वार: सन्ति | वेद: चत्वार: सन्ति छन्द: ब्राह्मणम् तैत्तिरीयकम् कठोपनिषत् वैदिक-च्छन्दसाम् मन्त्रम् आध्यपनम् अ-स्ति 
 वेद: चत्वार: सन्ति प्रसिद्ध: शिक्षा-शास्त्रीय: शिक्षकायग: शास्त्रम् शा-स्त्रम् कणाद: पाणिनीय शिक्ष का-ण्डे सूत्राणाम् उपरि भाष्यम् लिखि-तवान् 
 वेद: चत्वार: सन्ति धर्मशास्त्राणि वेदा: वैदिककर्मकाण्डम् आन्वी-क्षिकी दण्डनीतिप्रभृतीनि | वेद: चत्वार: सन्तिर्जनो गन्तु सेवेद: चत्वार: सन्तिर्जनो मि 
 वेद: चत्वार: सन्तिसेंद्रिष्टीवर: स्तेम: स्त्रेम: स्त्रजा 
 वेद: चत्वार: सन्तिवशावेदो चन्द्रादो श्री रं चवारासादो संस् | वेद: चत्वार: सन्तियार्गवेदंपर्य-कारा नमर्थित्यार्गवेदं⬜ 
 वेद: चत्वार: सन्तिर्थलस्पातिवर्त-नाचान्त्रासद्धिरतंप्रविध⬜ 
 वेद: चत्वार: सन्ति होगा, का सम्बेशच्छूंजिंक्षिया अष्भेद च⬜ |
| महां मिष्टान्नं रोचते | महां मिष्टान्नं रोचते ते ददामि ॥ १२॥ देवा अन्नं देहि मेऽजन्मिब-न्ति। 
 महां मिष्टान्नं रोचते ते दिने दिने ॥ 
 महां मिष्टान्नं रोचते ते हि भक्त्या मया दत्तं वारिजाते सरोजे । | महां मिष्टान्नं रोचतेसारास्तान्बुसान्व-श्व काव्याकसमान्ता⬜ 
 महां मिष्टान्नं रोचते। सँख्या भर्तरा-न्नराम्यनस्तेन। यसल 
 महां मिष्टान्नं रोचते त्रेभुचल्योगाम्-सीयचिन:रुक्षणं | महां मिष्टान्नं रोचते गाम्य वाकीच्छ हिंद्धिक्षिता महंति हो 
 महां मिष्टान्नं रोचते ग्रमास्य भूख-र्णव भुखवारोधिवर्णयम्⬜ 
 महां मिष्टान्नं रोचतें समुनं कहतियों दूस्यया में रहीं में म |
| किमर्थं त्वं गच्छसि | किमर्थं त्वं गच्छसि अस्माभि: मे स-म्मुखं प्रसारितं भविष्यति 
 किमर्थं त्वं गच्छसि इत्यत उच्यते कुरुष्विति 
 किमर्थं त्वं गच्छसि तद्रिदं विदे | किमर्थं त्वं गच्छसि पर तर जहां तिथं त्वं गच्छिरण्यक्ष गतिवि⬜ 
 किमर्थं त्वं गच्छसि तदतो अविजा-यिं वस्तु विदिमा आंतर्यंतनोज 
 किमर्थं त्वं गच्छसितो जातो किम-र्थं त्वं प्रभ्यक्तिर्मित ग | किमर्थं त्वं गच्छसिथ्या, नोर्थ त्रह युजर प्रणं मुझ्यमापर कि⬜ 
 किमर्थं त्वं गच्छसिर्णकार चेलार किलारी विष्णि प्रिशते चाहिय 
 किमर्थं त्वं गच्छसिकार्व है। सम-झ्यत्रिात्रान किम्मापस्त⬜ |

Table 34: mParamanu 162M vs GPT-Neo 1.3B vs GPT2-XL 1.5B open end text generation for Sanskrit prompts.

| Sanskrit Input Prompt | mParamanu 162M | Bloom 560M | Bloom 1.1B | Bloom 3B |
|---|---|---|---|---|
| अस्माकं देश: भारतवर्ष: | अस्माकं देश: भारतवर्ष: इति परिगणित: अस्माकं देश: भारतवर्ष: इति ख्यात: अस्माकं देश: भारतवर्ष: इति प्रसिद्ध: | अस्माकं देश: भारतवर्ष: भारत (भारत सरकार)। भारत के संविधान में राज्य की राजधानी रांची को "राज्यसभा" में रखा गया है, जो रांची से लगभग 8 किलोमीटर की दूरी पर स्थित है। राज्यसभा के विस्तार में, रांची में एक "अस्माकं राष्ट्र" ( (राज्यसभा) है। यह पूरे भारत की राजधानी में एक "राज्यसभा" का केंद्र है। राज्यसभा में भाग लेने के लिए सभी लोगों को अनिवार्य रूप से अपना नाम ( अस्माकं देश: भारतवर्ष: अस्माकं देश: भारतवर्ष: चीन: ओमान: दक्षिण एशिया: ओजी एस्टेली: ओमान: पाकिस्तान: प्रागैतिहासिक: प्रागैतिहासिक: अर्लक देश: प्रागैतिहासिक: सूरी: सूरी: सिलीगुड़ी: सिलीगुड़ी: सिलीगुड़ी: सिलीगुड़ी: सिलीगुड़ी: सिलीगुड़ी: सिलीगुड़ी: सिलीगुड़ी: | अस्माकं देश: भारतवर्ष:॥ अस्माकं देश: भारतवर्ष:॥ ५ हे ! परमात्याय ! तत्रं तुस्य तत्रं॥ ७ हे ! परमात्माय ! तत्रं तुस्य तत्रं॥ ७ हे ! परमात्माय ! तत्रं तुस्य तत्रं॥ ७ हे ! परमात्माय ! तत्रं तुस्य तत्रं॥ ७ नृत्. अस्माकं देश: भारतवर्ष: भीषर्शिध्यता, योगमणिरिणि: दो अमहानाति | अस्माकं देश: भारतवर्ष: । प्- प्र : देश। अस्मार : अस्मात्, अर्थात् इस प्रकार। प्- प्र : प्र : अस्माकं देश: भारतवर्ष: ॥ौटिल्य ने इस देश का विवरण देने के लिए चार मुख्य लक्षणों के आधार पर अपनी रचना की है । अस्माकं देश: भारतवर्ष:' आदि ने इसे स्वीकार किया। परंतु जब उसने सुना कि 'देवलोक:' (देवलोक) का नाम प्रचलित हो चुका है, तब उसने इसे अस्वीकार कर दिया। इस प्रकार यह स्पष्ट है कि वैदिक काल से लेकर आधुनिक युग तक इस शब्द का प्रयोग बहुत ही प्रचलित है। संस्कृत और अवधी में इसकी अनेक शब्दावलियाँ प्रचलित हैं, जो कि वैदिक अर्थों में भिन्न हैं। पर उनमें मूल के समान अर्थ समाहित हैं। संस्कृत में', 'अस्माकं देश: भारतवर्ष: । प्- प्र : देश। अस्मार : अस्मात्, अर्थात् इस प्रकार। प्- प्र : प्र : देश |
| वेद: चत्वार: सन्ति | वेद: चत्वार: सन्ति छन्द: ब्राह्मणम् तैत्तिरीयकम् कठोपनिषत् वैदिकच्छन्दसाम् मन्त्रम् आध्यपनम् अस्ति वेद: चत्वार: सन्ति प्रसिद्ध:: शिक्षाशास्त्रीय: शिक्षाकाया: शास्त्रम् शास्त्रम् कणाद: पाणिनीय शिक्षा काण्डे सूत्राणाम् उपरि भाष्यम् लिखितवान् वेद: चत्वार: सन्ति धर्मशास्त्राणि वेदा: वैदिकर्मकाण्डम् आन्वीक्षिकी दण्डनीतिप्रभृतीनि | वेद: चत्वार: सन्ति: - : - ! वेद: चत्वार: सन्ति समन् - नं द्धि ब , ग , - , - , - , नं ग , - , - , - , - , नं , ग , - , - , - , - , नं , ग , - , - , - , नं ग , - , - , - , नं , ग , - , - , - , नं- वेद: चत्वार: सन्ति श्रिष्ठि प्र: त्र्याण् सिम स-वं: द्योति सर्व: सि: न: वं स: देवाना: द्योति सर्व: सि: न: वं स: देवाना: द्योति सर्व: सि: न: वं स: देवाना: द्योति सर्व: सि: न: वं स: देवाना: द्योति सर्व: सि: न: वं | वेद: चत्वार: सन्ति शल्लि । न: शांक्षते । वेद: चत्वार: सन्ति । नित्यानन्द- जो किसी भक्त की भक्ति करता है और भक्त के लिए कोई काम करता है उस भक्त का नाम भक्त ही है। इस बात को समझना चाहिए कि भक्त शब्द का अर्थ सिर्फ "जो" या "और" है। ये शब्द एक न्यास या साधन की बात है। भक्त की आत्मा के प्रति लगाव और भक्ति भावना ही भक्त का अर्थ है। यही भाव भक्त को परम पिता की कृपा से अपने भक्त के रूप', 'वेद: चत्वार: सन्ति ॥ ९ १ ॥ जो सत्य , परमात्मा को ही जानती है, वह भक्त है वेद: चत्वार: सन्ति निर्भि॥ ७८॥ नकुल: पर्मण्यश्च कत्रि॥ ७९॥ नकुल: नर्भे सत्वत: भृज:॥ ७७०॥ नकुल: नर्भे सर्व: चृथ्य:॥ ७ ७ १॥ नकुल: नर्भे पुरुष: नर्भे॥ ७ ७ २॥ नकुल: नर्भे शारीर: चक्षय:॥ ७ ७ ३ | वेद: चत्वार: सन्ति प्रेयसी वेद: चत्वार: सन्ति . वेद: चत्वार: सन्ति प्रेयसी । |
| महां मिष्टान्नं रोचते | महां मिष्टान्नं रोचते ते ददामि ॥ १२॥ देवा अन्नं देहि मेऽजन्यिबन्ति। महां मिष्टान्नं रोचते ते हि दिने दिने ॥ महां मिष्टान्नं रोचते ते हि भक्त्या मया दत्नं वारिजाते सरोजे । | महां मिष्टान्नं रोचतेसारास्तानुसान्त्रश्च काव्याकसमान्ता महां मिष्टान्नं रोचतें। सँख्या भर्तन्रान्रराम्यन्स्तेन। यसल महां मिष्टान्नं रोचतें त्रैभुचल्योगामसीयचिन्नःरुक्षणं | महां मिष्टान्नं रोचते गाम्य वाकीच्छ हिंद्धिक्षिता महति हो महां मिष्टान्नं रोचतें ग्रमास्य भूखर्णव भुख-वारीधिवर्णयम् महां मिष्टान्नं रोचतें समुनं कहितयों द्युस्या में रहीं में म | अस्माकं देश: भारतवर्ष: । साक्षात्वाय:। कारण वापरारिव्यादद्द् अस्माकं देश: भारतवर्ष: अस्माकं देश: केवल स्थापित: नका बहुत वि अस्माकं देश: भारतवर्ष: सवै जातात्यायान्नजी नावा तपासन्धुव: सु |
| किमर्थं त्वं गच्छसि | किमर्थं त्वं गच्छसि अस्माभि: मे सम्मुखं प्रसारितं भविष्यति किमर्थं त्वं गच्छसि इत्यत उच्यते कुरुष्चिति किमर्थं त्वं गच्छसि तद्विदं विदे | किमर्थं त्वं गच्छसि पर तर जहां तिथं त्वं ग-च्छिरण्यक्ष गतिव किमर्थं त्वं गच्छसि ततदो अविजायीं वस्तु विदिमा आंतर्यंतनोज किमर्थं त्वं गच्छसितो जातो िमर्थं त्वं प्रभ्यक्तिमित ग | किमर्थं त्वं गच्छसिथ्या, नोर्थ त्रह युजर प्रणं मुड्यमापर कि किमर्थं त्वं गच्छसिर्णिकार चेलार किलारी विष्णि प्रिशते चाहिय किमर्थं त्वं गच्छसिकार्व। समझ्यत्रित्रान किमपस्त | अस्माकं देश: भारतवर्ष:। साक्षात्वाय:। कारण वापरारिव्यादद्द् अस्माकं देश: भारतवर्ष: अस्माकं देश: केवल स्थापित: नका बहुत वि अस्माकं देश: भारतवर्ष: सवै जातात्यायान्नजी नावा तपासन्धुव: सु |

Table 35: mParamanu 162M vs Bloom (560M, 1.1B, 3B) open end text generation for Sanskrit prompts.

| Hindi Input Prompt | Text Generation (top-3) from Paramanu-Hindi 162M | GPT-Neo 1.3B | GPT2-XL |
|---|---|---|---|
| सचिन तेंदुलकर (Sachin Tendulkar) | सचिन तेंदुलकर, गांगुली, द्रविड़, लक्ष्मण और सहवाग के अलावा विराट कोहली और आर अश्विन के 18-18 छक्कों के रिकॉर्ड की भी बीसीसीआई ने खोज शुरू कर दी है। (sacina tēṁdulakara, gāṁgulī, dravira, lakṣmaṇa aura sahavāga kē alāvā virāṭa kōhalī aura āra aśvina kē 18-18 chakkōṁ kē rikôrḍa kī bhī bīsīsīāī nē khōja śurū kara dī hai.) सचिन तेंदुलकर का आखिरी व 200 वां टेस्ट काफी अहम है और इसे एक महान खिलाड़ी को यादगार विदाई देने के लिहाज से बेहद अहम कहा जा सकता है। (sacina tēṁdulakara kā ākhirī va 200 vāṁ ṭēsṭa kāphī ahama hai aura isē ēka mahāna khilāṛī kō yādagāra vidāī dēnē kē lihāja sē bēhada ahama kahā jā sakatā hai.) सचिन तेंदुलकर का आखिरी व 200 वां टेस्ट काफी अहम है और इसे एक महान खिलाड़ी को यादगार विदाई देने के लिहाज से बेहद अहम कहा जा सकता है। (sacina tēṁdulakara kā ākhirī va 200 vāṁ ṭēsṭa kāphī ahama hai aura isē ēka mahāna khilāṛī kō yādagāra vidāī dēnē kē lihāja sē bēhada ahama kahā jā sakatā hai.) | सचिन तेंदुलकर एक भावने पर चारने से बाहर है यहाँ के रंग से यहाँ स्थ सचिन तेंदुलकर सुवना की जगह की महाराज तिर-स्मादित्या विशेष राव्य सचिन तेंदुलकर्ताओं के रूप में एक वैश्विक करना है, इसलिए हमें य | सचिन तेंदुलकर्थावेगति में सम्हंते मवाचन भववामन है दियाल का गय सचिन तेंदुलकरोप्रवेद्धितं यही के जम भविश्रं के बाद जिंग के सचिन तेंदुलकर्मों दोनीदियंत्र तुशांके यहकल आना दी भीषशित |
| शाहरुख खान (Shah Rukh Khan) | शाहरुख खान की फिल्म डॉन को आज मुंबई और आसपास के इला-कों में बड़े पैमाने पर देखा जा रहा है। (śāharuk☐ha k☐hāna kī filma ḍôna kō āja mumbaī aura āsapāsa kē ilākōṁ mēṁ baṛē paimānē para dēkhā jā rahā hai.) शाहरुख खान, रीतेश देशमुख, अनुष्का शर्मा, रणवीर शौरी, लारा दत्ता, नाना पाटेकर, आशीष चौधरी, नवजोत सिद्धू, शक्ति कपूर, गो-विंदा, चंकी पांडे के अभिनय वाली इस फिल्म ने शुक्रवार को 10.11 करोड़ की कमाई की। (śāharuk☐ha k☐hāna, rītēśa dēśamukha, anuṣkā śarmā, raṇavīra śaurī, lārā dattā, nānā pāṭēkara, āśīśa caudharī, navajōta siddhū, śakti kapūra, gōviṁdā, caṁkī pāṁḍē kē abhinaya vālī isa philma nē śukravāra kō 10.11 karōṛa kī kamāī kī.) शाहरुख खान के प्रशंसक बनें या शाहरुख खान के विरोधी , शाहरुख खान को लेकर बहुत कुछ लिखा जा सकता है (śāharuk☐ha k☐hāna kē praśaṁsaka banēṁ yā śāharuk☐ha k☐hāna kē virōdhī , śāharuk☐ha k☐hāna kō lēkara bahuta kucha likhā jā sakatā hai) | शाहरुख खानी चापलबुती रिलाई वापरते बरे-शति के शिक्षता सीज शाहरुख खाने ही रहते हैं। अब तुम्हें पता लगाने के लिए मुख्या के व' शाहरुख खाने पर भीतर पिछले सप्ताह के रूप में जीवन चालू करने के | शाहरुख खान खादने प्रहारुख भारत करणें है तो तोले के उस के रोजी भा शाहरुख खाने लिए अयोगाने वाली वरीज़ा दोववे मंदी लोगों पहले के शाहरुख खान कोई यही मनीजर के लोगा आने है?करी नहीं की साथ रहता कर |
| महात्मा गांधी (Mahatma Gandhi) | महात्मा गांधी और उनके समकालीन नेताओं का मुख्य ध्यान समाज के अंतिम व्यक्ति को महत्व देने पर था महात्मा गांधी राष्ट्रीय ग्रामीण रोजगार गारंटी योजना के तहत संविदा के कर्मचारियों की हड़ताल का असर इन्दौर में भी दिखा। महात्मा गांधी भी अपनी जिंदगी में मर्यादा का पालन करते थे और अगर वे हिंसा का सहारा ले रहे हैं तो वे भी विचारधारा के शिकार हुए हैं। | महात्मा गांधी महात्मा गांधी के भावनामधील आनंद भर आवश्य-कता के राजीव्य गिनी गांध' महात्मा गांधी के भावनामधील आनंद भर आवश्य-कता के राजीव्य गिनी गांध महात्मा गांधीशोऽऽते गार्जी आवश्यक आहे. तर तथा व्यरण नामवरणो हन | महात्मा गांधी भी साथा पूरवस ब्राहमें शोहिलेकंपर त्या हैं Name:Phone No: महात्मा गांधी की से करेंड़ नीते है मुस्ती और दिया जानके से हमें आ महात्मा गांधी का में समग्रवोरवो देखारा से पर पर रहाते है, जिस्कार |
| लता मंगेशकर (Lata Mangeshkar) | लता मंगेशकर नूरजहां और शमशाद बेगम के साथ भी गा चुकी थीं। (latā maṁgēśakara nūrajahāṁ aura śamaśāda bēgama kē sātha bhī gā cukī thīṁ.) लता मंगेशकर का जन्म २८ सितम्बर १९२९ को वर्तमान पाकि-स्तान में हुआ था। (latā maṁgēśakara kā janma 28 sitambara 1929 kō vartamāna pākistāna mēṁ huā thā.) लता मंगेशकर ने रील लाइफ से रियल लाइफ तक का सफर तय किया है और रियल लाइफ के उनके करीबियों ने उन्हें रील लाइफ में भी न सिर्फ रियल लाइफ बल्कि लव लाइफ में भी जीना सिखा दिया है। (latā maṁgēśakara nē rīla lāipha sē riyala lāipha taka kā saphara taya kiyā hai aura riyala lāipha kē unakē karībiyōṁ nē unhēṁ rīla lāipha mēṁ bhī na sirpha riyala lāipha balki lava lāipha mēṁ bhī jīnā sikhā diyā hai.) | लता मंगेशकर है तब मेरा नाम गुल्मने में अक्षम कर दिया है यह वास्तव लता मंगेशकर करने की जरूरत हो जाए तेल्स के अंदर बाहर अमेज में रखा ज लता मंगेशकर अपेक्षा में या उसका दूसरा एक सारा हजारों से स्पष्ट | लता मंगेशकर प्रायना को बहुतो गयाते समझ्रे दिली ननें गया विचने के लता मंगेशकर स्वास रहित आरवोईरहे मन्दुनीते के नहीं सब मंगेशकर लता मंगेशकर प्रयवेस प्रायवासने जानकारुष्यियते-त्के देश सिवास |

Table 36: Paramanu-Hindi 162M vs GPT-Neo 1.3B vs GPT2-XL 1.5B open end text generation for Hindi prompts.

| Hindi Input Prompt | Text Generation (top-3) from Paramanu-Hindi 162M | OPT 6.7B |
|---|---|---|
| सचिन तेंदुलकर (Sachin Tendulkar) | सचिन तेंदुलकर, गांगुली, द्रविड़, लक्ष्मण और सहवाग के अलावा विराट कोहली और आर अश्विन के 18-18 छक्कों के रिकॉर्ड की भी बीसीसीआई ने खोज शुरू कर दी है।
सचिन तेंदुलकर का आखिरी व 200 वां टेस्ट काफी अहम है और इसे एक महान खिलाड़ी को यादगार विदाई देने के लिहाज से बेहद अहम कहा जा सकता है।
सचिन तेंदुलकर का आखिरी व 200 वां टेस्ट काफी अहम है और इसे एक महान खिलाड़ी को यादगार विदाई देने के लिहाज से बेहद अहम कहा जा सकता है। | सचिन तेंदुलकर क्षेत्र रिपोर्ट I am a writer and a thinker. I write on contemporary political, social, cultural, economic, psychological, and philosophical issues. I write on the subjects of politics, philosophy, psychology, and sociology. I am an amateur
सचिन तेंदुलकर आयोजित कम संयुक्त करेगा।लेकिन जब संयुक्त कम
सचिन तेंदुलकर एक दो वह पास करते रहे हैं आज मगर अब पास करते हैं सब |
| शाहरुख़ ख़ान (Shah Rukh Khan) | शाहरुख़ ख़ान की फ़िल्म डॉन को आज मुंबई और आसपास के इलाकों में बड़े पैमाने पर देखा जा रहा है।
शाहरुख़ ख़ान, रीतेश देशमुख, अनुष्का शर्मा, रणवीर शौरी, लारा दत्ता, नाना पाटेकर, आशीष चौधरी, नवजोत सिद्धू, शक्ति कपूर, गोविंदा, चंकी पांडे के अभिनय वाली इस फिल्म ने शुक्रवार को 10.11 करोड़ की कमाई की।
शाहरुख़ ख़ान के प्रशंसक बनें या शाहरुख़ ख़ान के विरोधी , शाहरुख़ ख़ान को लेकर बहुत कुछ लिखा जा सकता है | शाहरुख़ ख़ानी चापलबुती रिलाई वापरते मेरा बरेशति के शिक्षता सीज☐
शाहरुख़ ख़ाने ही रहते हैं. अब तुम्हें पता लगाने के लिए मुख्य का व'
शाहरुख़ ख़ाने पर भीतर पिछले सप्ताह के रूप में जीवन चालू करने के ☐ |
| महात्मा गांधी (Mahatma Gandhi) | महात्मा गांधी और उनके समकालीन नेताओं का मुख्य ध्यान समाज के अंतिम व्यक्ति को महत्व देने पर था
महात्मा गांधी राष्ट्रीय ग्रामीण रोजगार गारंटी योजना के तहत संविदा के कर्मचारियों की हड़ताल का असर इन्दौर में भी दिखा।
महात्मा गांधी भी अपनी जिंदगी में मर्यादा का पालन करते थे और अगर वे हिंसा का सहारा ले रहे हैं तो वे भी विचारधारा के शिकार हुए हैं। | महात्मा गांधी राम की अनुराष्ट्रित बल नहीं, आनंद को खोज के आरोपि☐
महात्मा गांधी ने मानवरी अपना प्रवेश के नेतृत्व को बढ़ाने के लिए ☐
महात्मा गांधी में मैंने नागलिंग देखने को सेना की संपर्क के साथ आ☐ |
| लता मंगेशकर (Lata Mangeshkar) | लता मंगेशकर नूरजहां और शमशाद बेगम के साथ भी गा चुकी थीं।
लता मंगेशकर का जन्म २८ सितम्बर १९२९ को वर्तमान पाकिस्तान में हुआ था।
लता मंगेशकर ने रील लाइफ से रियल लाइफ तक का सफर तय किया है और रियल लाइफ के उनके करीबियों ने उन्हें रील लाइफ में भी न सिर्फ रियल लाइफ बल्कि लव लाइफ में भी जीना सिखा दिया है। | लता मंगेशकरती अमेरा यांची बाला सोमवार यांची जनता का सही दिवस शब्द'
लता मंगेशकर पीसद में एक दिनों प्राप्त पहुंचे। इस निधन से पहुं
लता मंगेशकर आज अंदर बालिक बने विधायक पर कब्जे को देखा हूँ। इसलि |

Table 37: Paramanu-Hindi 162M vs OPT 6.7B open end text generation for Hindi prompts.

| Hindi Input Prompt | Text Generation (top-3) from Paramanu-Hindi 162M | LLaMa-2 7B |
|---|---|---|
| सचिन तेंदुलकर (Sachin Tendulkar) | सचिन तेंदुलकर, गांगुली, द्रविड़, लक्ष्मण और सहवाग के अलावा विराट कोहली और आर अश्विन के 18-18 छक्कों के रिकॉर्ड की भी बीसीसीआई ने खोज शुरू कर दी है। सचिन तेंदुलकर का आखिरी व 200 वां टेस्ट काफी अहम है और इसे एक महान खिलाड़ी को यादगार विदाई देने के लिहाज से बेहद अहम कहा जा सकता है। सचिन तेंदुलकर का आखिरी व 200 वां टेस्ट काफी अहम है और इसे एक महान खिलाड़ी को यादगार विदाई देने के लिहाज से बेहद अहम कहा जा सकता है। | सचिन तेंदुलकर सचिन तेंदुलकर के हकीकतें जानिए सचिन तेंदुलकर के हकीकत जानिए और करें सही घोषणा. प्रमुख मुद्दों पर राजे पाली पहले इशारे दिए। बॉलीवुड में रहने वाले सितारों में कोई फेहरा नहीं बल्कि बेहद सुखदायी है। फिर भी सब मानेंगे सितारों को उम्मीद सचिन तेंदुलकर राउंड में मार रहे कुछ विदेशियों के प्लेवर ने दिल झका लिया, बीस फेमिना की मारी रही स्पिवस स्कूल पाठ्यालय के एडमिनिस्ट्रेट फांदीन अहमद जी की जांच पर दिल्ली में कई मुस्लिम महिलाओं को हमला करने की आकलाम आयीं । बहुत जल्दी बाद |
| शाहरुख़ खान (Shah Rukh Khan) | शाहरुख खान की फिल्म डॉन को आज मुंबई और आसपास के इला-कों में बड़े पैमाने पर देखा जा रहा है। शाहरुख खान, रीतेश देशमुख, अनुष्का शर्मा, रणवीर शौरी, लारा दत्ता, नाना पाटेकर, आशीष चौधरी, नवजोत सिद्धू, शक्ति कपूर, गो-विंदा, चंकी पांडे के अभिनय वाली इस फिल्म ने शुक्रवार को 10.11 करोड़ की कमाई की। शाहरुख़ ख़ान के प्रशंसक बनें या शाहरुख़ खान के विरोधी, शाहरुख़ खान को लेकर बहुत कुछ लिखा जा सकता है | शाहरुख खान और अमिताब बच्चन ने रोहित शर्मा के निकाट रहे सिलवाई शादी में अनुभव की दुनिया देखा Dear reader, its been a little over a month since Salman Khans hit-and-run case was closed. While a lot of speculations had been doing the rounds about Salmans alleged affair with Aishwarya Rai, the superstar has repeatedly denied it and said that he is a happily married man. Well, Salmans wife, Arpita Khan, has been very support-ive of her husband. In fact, its been widely reported that Arpita and Salmans sister, Alvira, have been quite worried about the allegations. They have been concerned about the reports about the superstars affairs with women like Katrina Kaif, A", ' शाहरुख खान की बॉलीवुड करियर के संकटों से रिकॉर्ड क्रिकेटर लेक कहा कि यह सौभाग्य कि मेरे पिता महादेव कपूर नहीं थे और हम नहीं मिलीं। रिकॉर्ड क्रिकेटर लेक खिलाड़ी पर बढ़ते हुए क्रिकेटर के एक बार फिर नाम बाज़ार में आ गये। ये बात सबके स शाहरुख खान की बॉलीवुड करियर के संकटों से रिकॉर्ड क्रिकेटर लेक कहा कि यह सौभाग्य था कि मेरे पिता महादेव कपूर नहीं थे और हम नहीं मिलीं। रिकॉर्ड क्रिकेटर लेक खिलाड़ी पर बढ़ते हुए क्रिकेटर के एक बार फिर नाम बाज़ार में आ गये। ये बात सबके स शाहरुख खान ने संगम अलफ़ाजी की निशानी के खिलाफ घेर दिया हाथ 2023-03-25T15:36:04+05:30 Shah Rukh Khans Mother Shares Video Of Her Husband "Taking Sip Of Water After Taking His Medicine" Shah Rukh Khan's mother, Lateef Fatima Khan, took to her Instagram handle to share a video of her husband "taking a sip of water after taking his medicine".Recently, Sharukh Khan shared a heartfelt post for his father. He also shared a video of him in which he is seen speaking on his fa-thers health. In the clip, SRK can be seen sitting in a chair and holding his fathers hand. The video went viral on social media as many peo-ple were moved by the sight of the King Khan holding his fathers hand. SRK had |
| महात्मा गांधी (Ma-hatma Gandhi) | महात्मा गांधी और उनके समकालीन नेताओं का मुख्य ध्यान समाज के अंतिम व्यक्ति को महत्व देने पर था महात्मा गांधी राष्ट्रीय ग्रामीण रोजगार गारंटी योजना के तहत संविदा के कर्मचारियों की हड़ताल का असर इन्दौर में भी दिखा। महात्मा गांधी भी अपनी जिंदगी में मर्यादा का पालन करते थे और अगर वे हिंसा का सहारा ले रहे हैं तो वे भी विचारधारा के शिकार हुए हैं। | महात्मा गांधी अंको अपलाईल होली पाउडाको सुवासियालाई लिनक मुख्य राष्ट्र-तिमा पत्र मिला Kathmandu, July 25 (RSS): Rashtriya Samajbadi Party Nepal (RSPN) leader Madhav Kumar Nepal has written a letter to Pres-ident Bidya Devi Bhandari seeking the appointment of former prime ministers Sher Bahadur Deuba and Pushpa Kamal Dahal as the prime minister, RSPNs central office said. In the letter dated 24 July, Nepal requested the president to make Nepal Communist Party (NCP) chair-person and former prime minister Deuba, RSPN general secretary and former prime minister Dahal and NCP leader and former prime minister Jhala Nath Khanal prime minister through a joint consensus. The महात्मा गांधी की परिचितता बढ़ाने के लिए सुधार प्रक्रिया का आह्वान अभियान की शुरुआत करें श्रीमान महात्मा गांधी की मान्यता हिन्दुस्तान की सर्वोच्चता प्रेम की सी-माओं से बाहर से अच्छी तरह से अतिरिक्त होना चाहिए और हिन्दुस्तान पर एक दूसरी ध्यान महात्मा गांधी का साहित्यिक पत्रिका भारतीय लोकसभा महासभा में कौन वाले बड़े राजनितिक प्रधान थे? कबीर सिंह की बेटी सत्य सिंह को कब समेत देश की महान तरीक़ा प्रेरित होती थीं? उसकी परिवार व घर के जांच से कहीं नहीं पाई जाती थी कि जातिवाद का गुलाम अब्दुल |
| लता मंगेशकर (Lata Mangeshkar) | लता मंगेशकर नूरजहां और शमशाद बेगम के साथ भी गा चुकी थीं। लता मंगेशकर का जन्म २८ सितंबर १९२९ को वर्तमान पाकिस्तान में हुआ था। लता मंगेशकर ने रील लाइफ से रियल लाइफ तक का सफर तय किया है और रियल लाइफ के उनके करीबियों ने उन्हें रील लाइफ में भी न सिर्फ रियल लाइफ बल्कि लव लाइफ में भी जीना सिखा दिया है। | लता मंगेशकर पर पर पर पर पर पर पर पर पर पर पर पर पर पर पर पर पर पर पर पर पर पर पर पर पर पर पर पर पर पर पर पर पर पर पर पर पर पर पर पर पर पर पर पर पर पर पर पर लता मंगेशकर बोले हर पहले उच्च शिक्षा के बारे में अपनी दुर्घटना से सम्बंधित बात-चीत लता मंगेशकर की बेटी पियाली ने बताया कहां की उम्र से शादी करना चाहते प्याली एक अदृश्य विवाहिक नाम बता रही है। वह अपने किस्सें अकेला खासी एवं अपो-केतिक अर्थमिक होने पर वापस अपनी प्याली पर कहती हैं। पात्रता से प्याली उम्र छात्र-दारियों से म |

Table 38: Paramanu-Hindi 162M vs LLaMa-2 7B open end text generation for Hindi prompts.

| Hindi Input Prompt | Text Generation (top-3) from Paramanu-Hindi 162M | GPT-J 6B |
|---|---|---|
| सचिन तेंदुलकर (Sachin Tendulkar) | सचिन तेंदुलकर, गांगुली, द्रविड़, लक्ष्मण और सहवाग के अलावा विराट कोहली और आर अश्विन के 18-18 छक्कों के रिकॉर्ड की भी बीसीसीआई ने खोज शुरू कर दी है। 
 सचिन तेंदुलकर का आखिरी व 200 वां टेस्ट काफी अहम है और इसे एक महान खिलाड़ी को यादगार विदाई देने के लिहाज से बेहद अहम कहा जा सकता है। 
 सचिन तेंदुलकर का आखिरी व 200 वां टेस्ट काफी अहम है और इसे एक महान खिलाड़ी को यादगार विदाई देने के लिहाज से बेहद अहम कहा जा सकता है। | सचिन तेंदुलकर वरचेता जाते. Kannada: ಕೆಚ್ಚಬ್ಬಿ ಇದ್ದಾನೆಗಂಳು ವಾದ ಮತ್ತು ನಂತ್ಯ ಎಂದು ಕಂಡಿದಾಗುತ್ತಾರೆ. Serbian:     I am a writer and a thinker. I write on contemporary political, social, cultural, economic, psychological, and philosophical issues. I write on the subjects of politics, philosophy, psychology, and sociology. I am an amateur 
 सचिन तेंदुलकर शेखपद तकियोवा ☐☐☐☐☐☐ डॉवरका शनिवार, 10 जनवरी, 2010 को दंपत दूल्हें दो बार दी सिंहसिंग देखभाल और कीटनाशकों में तरीका समाप्त। अगले देर में दो बार देखभाल और बचाव 
 सचिन तेंदुलकर हैन्च होकिन्ने दुष्पतेंदुलकर होकिन्ने दुष्पतेंदुलकर होकिन्ने दुष्पतेंदुलकर होकिन्ने दुष्पतेंदुलकर होकिन् |
| शाहरुख़ ख़ान (Shah Rukh Khan) | शाहरुख़ ख़ान की फ़िल्म डॉन को आज मुंबई और आसपास के इलाकों में बड़े पैमाने पर देखा जा रहा है। 
 शाहरुख़ ख़ान, रीतेश देशमुख, अनुष्का शर्मा, रणवीर शौरी, लारा दत्ता, नाना पाटेकर, आशीष चौधरी, नवजोत सिद्धू, शक्ति कपूर, गोविंदा, चंकी पांडे के अभिनय वाली इस फ़िल्म ने शुक्रवार को 10.11 करोड़ की कमाई की। 
 शाहरुख़ ख़ान के प्रशंसक बनें या शाहरुख़ ख़ान के विरोधी , शाहरुख़ ख़ान को लेकर बहुत कुछ लिखा जा सकता है | शाहरुख़ ख़ानेकु काके गह होइनको परम्परामाले जस्तो छैन शेल्गा काकेकु शाहरुख़रूपमा समाजको अरूलाई हाम्रो रोजिसम्म यति नयाँ शहरको बचाउनु परंतु यसको जनबीतको जाति☐ 
 शाहरुख़ ख़ानेर तिमिरन कहुन्छ यस्तोमा खुटेको लागि काम गर्नु कनें हुँदै Estonian: Nagu siin on hästi, saate teadada, kuidas mina kirjutasin sinise koos tundumata, kuidas kirjutasin ülejäänud kõikide aegade tundumata. Nagu siin on hästi, saate teadada, kuidas mina kirjutasin sinise koos tundumata, kuidas kirjutasin ülejäänud kõikide aegade tundumata. 
 शाहरुख़ ख़ानाहरु देखको थिए तीन बाँयालहरुको विवाहमा पनि पैसा लगाएका थिए texthindi                                 Turkish: Japanese:     インドの女性は若いときにお金を使って食べました 三鳥のような子供はお金を取得することができます 結婚指輪は小さな社会で支払われました インドでは☐ |
| महात्मा गांधी (Mahatma Gandhi) | महात्मा गांधी और उनके समकालीन नेताओं का मुख्य ध्यान समाज के अंतिम व्यक्ति को महत्व देने पर था 
 महात्मा गांधी राष्ट्रीय ग्रामीण रोजगार गारंटी योजना के तहत संविदा के कर्मचारियों की हड़ताल का असर इन्दौर में भी दिखा। 
 महात्मा गांधी भी अपनी जिंदगी में मर्यादा का पालन करते थे और अगर वे हिंसा का सहारा ले रहे हैं तो वे भी विचारधारा के शिकार हुए हैं। | महात्मा गांधी प्राप्तमुपाते, सिद्धू समृद्ध भारतीयेत्र ब्रह्मा!च्छिन्न महात्मा पुरुष स्वत: हैतो। वहाँ तो केहि भी अज्ञा आयत आहे जैसे शते व भैंके, पुन्हा शेष भैंके 
 महात्मा गांधी में मृत के वर्ष में पहली बार कितना परेशानियां हुई। उन्होंने देखा है कि एक युवती का पैर अपनी देखरी पीठ में लहसुन भूमिका नहीं करती है। बाबा ने पीड़ित सोने 
 महात्मा गांधी की रोक करने की कोशिश करती हैं। अपनी पसंद व्यक्त करती हैं। अपनी देहिया के शरीर में इसे बदलना नहीं चाहती है। सेना को असुविधा दी जिसे सिर्फ अधिक न |
| लता मंगेशकर (Lata Mangeshkar) | लता मंगेशकर नूरजहां और शमशाद बेगम के साथ भी गा चुकी थीं। 
 लता मंगेशकर का जन्म २८ सितम्बर १९२९ को वर्तमान पाकिस्तान में हुआ था। 
 लता मंगेशकर ने रील लाइफ से रियल लाइफ तक का सफर तय किया है और रियल लाइफ के उनके करीबियों ने उन्हें रील लाइफ में भी न सिर्फ रियल लाइफ बल्कि लव लाइफ में भी जीना सिखा दिया है। | लता मंगेशकर को मालते हुए आश्रृस्तेज तरंगी का तेजतर पैगंबर बन गया है। उन्होंने पिछले 5 कोबीर बना मुख्य दुल्हन के अनुभाग में समाप्त किया है। मैनेजिक मोल्डन के एक कह 
 लता मंगेशकर्ता द्वारा भुगतान के पूरी तरह से कनेक्शन को समाप्त कर दिया। तो अब मैं आपको बताना चाहूंगा कि कौन सा व्यक्ति ने किसी चीज़ का भुगतान किया है? आपको एक दूसरे की भ 
 लता मंगेशकर्षण का भेंकं को हमेशा तो जनाती होता हैं। लेकिन वे काफी खूबसूरते हैं। Portuguese: Eu tenho minha própria música de fundo. Eu tenho um grande amigo em que o meu nome significa muito para ele, e ele fala em mim no último episódio de uma maneira realmente agradável, e é muito bom se você acompanhar isso, e ele é Muito boa como o meu amigo, ele é meu amigo, como o meu amigo, e ele é tão |

Table 39: Paramanu-Hindi 162M vs GPT-J 6B open end text generation for Hindi prompts.

| Hindi Input Prompt | Text Generation (top-3) from Paramanu-Hindi 367.5M | Bloom 560M | Bloom 1.1B | Bloom 3B | Bloom 7.1B |
|---|---|---|---|---|---|
| सचिन तेंदुलकर | सचिन तेंदुलकर का आखिरी व 200 वां टेस्ट काफी अहम है और इसे एक महान खिलाड़ी को यादगार विदाई देने के लिहाज से बेहद अहम कहा जा सकता है।
सचिन तेंदुलकर का आखिरी व 200 वां टेस्ट काफी अहम है और इसे एक महान खिलाड़ी को यादगार विदाई देने के लिहाज से बेहद अहम कहा जा सकता है।
सचिन तेंदुलकर, गांगुली, द्रविड़, लक्ष्मण और सहवाग के अलावा विराट कोहली और आर अश्विन के 18-18 छक्कों के रिकॉर्ड की भी बीसीसीआई ने खोज शुरू कर दी है। | सचिन तेंदुलकर ने कहा है कि उन्हें ये बात पसंद नहीं है कि भारतीय टीम में किसी खिलाड़ी की जगह दूसरी पारी में खेलने का मौका न मिल रहा हो । | सचिन तेंदुलकर से मुलाकात करेंगे और अपने नए कप्तानों को सम्मानित करेंगे। सचिन तेंदुलकर और भारतीय कप्तान महेंद्र सिंह धोनी के बीच चल रही विवाद के बीच अब ऑस्ट्रेलिया के टेस्ट कप्तान माइकल राइट ने भारतीय कप्तान को 'मीडिया के जिम्मेदार' बताया है। | सचिन तेंदुलकर के खिलाफ भारतीय टीम के पूर्व कप्तान महेंद्र सिंह धोनी ने कहा है कि उन्होंने टीम को जीत दिलाने के लिए एक टीम की जरूरत नहीं महसूस की सचिन तेंदुलकर और महेंद्र सिंह धोनी के लिए इस बात के मायने कई बार नहीं होता कि भारत के लिए खेलने वाले दो महान बल्लेबाज अब अपने करियर की समाप्ति पर अपने करियर को थाम लें | सचिन तेंदुलकर का मानना है कि सचिन तेंदुलकर को क्रिकेट का असली भगवान नहीं माना जा सकता क्योंकि उस समय सचिन की उम्र महज 19 वर्ष थी लेकिन वह खेल की दुनिया में छा गए और अब वह भारत में सबसे लोकप्रिय क्रिकेट खिलाड़ी है
सचिन तेंदुलकर के साथ है, जिन्होंने यह पुरस्कार जीता था। शारदा सिन्हा (सरस), पूर्व क्रिकेटर और टीवी पत्रकार शारदा सिन्हा (सरस), पूर्व क्रिकेटर और टीवी पत्रकार के रूप में जाना जाता है
सचिन तेंदुलकर और वीरेंद्र सहवाग भी शामिल थे। |
| शाहरुख खान | शाहरुख खान, रीतेश देशमुख, अनुष्का शर्मा, रणवीर शौरी, लारा दत्ता, नाना पाटेकर, आशीष चौधरी, नजजोत सिंधु, शक्ति कपूर, गोविंदा, चंकी पांडे अभिनय वाली इस फिल्म ने शुक्रवार को 10.11 करोड़ की कमाई की।
शाहरुख खान की फिल्म डॉन को आज मुंबई और आसपास के इलाकों में बड़े पैमाने पर देखा जा रहा है।
शाहरुख खान ने थैंक्यू सॉरीज़ बोलते हुए कहा, मैंने अपने सभी करीबी दोस्तों और परिवार वालों से कहा है कि ये समारोह बहुत जरूरी था। | शाहरुख खान के साथ है, जिन्होंने एक फिल्म का निर्देशन किया है।
शाहरुख खान के साथ फिल्म के पहले दृश्य में अमिताभ बच्चन का हाथ और कान पीछे रखकर एक शर्ट से पकड़े जाने वाला है।
शाहरुख खान का शर्ट और पैंटी को देखकर अंदाजा लगाया जा सकता है कि वे बहुत देर से पैंट और शर्ट पहन रही थीं। | शाहरुख खान को श्रद्धांजलि दी. शाहरुख खान की बेटी सुहाना खान अपने पहले बर्थडे को सेलिब्रेट करने जा रहीं हैं. सुहाना ने ट्विटर पर इस बात का ऐलान किया है कि वे अपनी पहली बर्थडे पार्टी 22 दिसम्बर को दिल्ली के एक कैफे में कर रहीं हैं | शाहरुख खान के अनुसार, उन्हें नहीं लगा कि उनकी फिल्म को कोई गंभीरता से ले रहा है. लेकिन फिल्म से पहले उन्हें लगता था कि लोग उन्हें पसंद करेंगे. लेकिन असलियत में दर्शकों का रवैया बिल्कुल अलग था.
शाहरुख खान की | शाहरुख खान के प्रशंसक बनें या शाहरुख खान के विरोधी, शाहरुख खान को लेकर बहुत कुछ लिखा जा सकता है |
| लता मंगेशकर | लता मंगेशकर नूरजहां और शमशाद बेगम के साथ भी गा चुकी थीं।
लता मंगेशकर का जन्म 28 सितम्बर १९२९ को वर्तमान पाकिस्तान में हुआ था।
लता मंगेशकर ने रील लाइफ से रियल लाइफ तक का सफर तय किया है और रियल लाइफ के उनके करीबियों ने उन्हें रील लाइफ में ही न सिर्फ रियल लाइफ बल्कि लव लाइफ में भी जीना सिखा दिया है। | लता मंगेशकर।
लता मंगेशकर ने लिखा है कि वह खुद की शादी को लेकर बिल्कुल भी शर्मिंदा नहीं है और वह अभी भी अपने लिए एक सुंदर और घर मां बनना चाहती हैं।
लता मंगेशकर , दीपक केसरिया , मनोज कुमार , शुभम शर्मा , अभिषेक शर्मा आदि नेगी नेगी नेगी नेगी नेगी नेगी नेगी नेगी नेगी नेगी नेगी नेगी नेगी नेगी नेगी नेगी नेगी नेगी नेगी नेगी नेगी नेगी नेगी नेगी नेगी नेगी नेगी नेगी नेगी नेगी नेगी | लता मंगेशकर , हरिओम चौहान , अमित शर्मा , विकास जैन , धर्मबीर जैन , सुनील शर्मा , बलबीर त्यागी , सतीश कुमार , राजेंद्र कुमार आदि मौजूद रहे ।
लता मंगेशकर की जांच को शुरू किया गया है बता दें कि ये मामला 17 साल पुराना है। इस मामले में दो आरोपी पुलिस की गिरफ्त में थे लेकिन पिछले दो महीने में यह मामला काफी बढ़ गया था। इससे पहले पूर्व मंत्री गायत्री प्रसाद प्रजापति के बेटे गायत्री प्रजापति और भाजपा विधायक संगीत सोम के बीच हुई झड़प के बाद अब गायत्री प्रजापति की हत्या की आशंका जताई जा रही है। आपको बता दें कि गायत्री प्रजापति की हत्या
लता मंगेशकर द्वारा लिखा गया। | लता मंगेशकर ने कहा कि मुझे बहुत अच्छा लग रहा है कि इतने वर्षों के बाद मेरी आवाज में वह शायरी आ गई
लता मंगेशकर , दिलीप ताहिल , विजय जाधव , मंगेश कदम , शशिकांत कदम, यशवंत शिंदबराव , प्रशांत ताहिल , विजय शिंदबराव , यशवंत शिंदबराव , प्रशांत ताहिल , विजय शिंदबराव , यशवंत शिंदबराव , प्रशांत ताहिल , विजय शिंदबराव , प्रशांत ताहिल , विजय शिंदबराव , यशवंत शिंदबराव , प्रशांत ताहिल , विजय शिंदबराव , यशवंत शिंदबराव , प्रशांत ताहिल , विजय शिंदबराव , यशवंत शिंदबराव , प्रशांत ताहिल , विजय शिंद', 'लता मंगेशकर , मंगेश कदम , सुरेश पंडित , सचिन पिळगांवकर , सुरेश वाडकर , अनिल विश्वास , लक्ष्मीकांत प्यारेलाल, लक्ष्मीकांत प्यारेलाल , लक्ष्मीकांत प्यारेलाल , लक्ष्मीकांत प्यारेलाल , लक्ष्मीकांत प्यारेलाल, लक्ष्मीकांत प्यारेलाल , लक्ष्मीकांत प्यारेलाल , लक्ष्मीकांत प्यारेलाल
लता मंगेशकर , मंगेश कदम , सुरेश पंडित , सचिन पिळगांवकर , लक्ष्मीकांत प्यारेलाल , लक्ष्मीकांत प्यारेलाल , लक्ष्मीकांत प्यारेलाल , लक्ष्मीकांत प्यारेलाल , लक्ष्मीकांत प्यारेलाल , लक्ष्मीकांत प्यारेलाल | लता मंगेशकर ने कहा कि मुझे बहुत अच्छा लग रहा है कि इतने वर्षों के बाद मेरी आवाज में वह शायरी आ गई
लता मंगेशकर , दिलीप ताहिल , विजय जाधव , मंगेश कदम , शशिकांत कदम, यशवंत शिंदबराव, विजय शिंदबराव , यशवंत शिंदराव , प्रशांत ताहिल , विजय शिंदबराव , यशवंत शिंदबराव , प्रशांत ताहिल , विजय शिंदबराव , यशवंत शिंदबराव , प्रशांत ताहिल , विजय शिंदबराव , यशवंत शिंदबराव, यशवंत शिंदबराव , प्रशांत ताहिल , विजय शिंदबराव , यशवंत शिंदबराव , प्रशांत ताहिल , विजय शिंदबराव , यशवंत शिंदबराव , प्रशांत ताहिल , विजय शिंद', 'लता मंगेशकर , मंगेश कदम , सुरेश पंडित , सचिन पिळगांवकर , सुरेश वाडकर , लक्ष्मीकांत प्यारेलाल , अनिल विश्वास , लक्ष्मीकांत प्यारेलाल, लक्ष्मीकांत प्यारेलाल , लक्ष्मीकांत प्यारेलाल , लक्ष्मीकांत प्यारेलाल , लक्ष्मीकांत प्यारेलाल, लक्ष्मीकांत प्यारेलाल , लक्ष्मीकांत प्यारेलाल , लक्ष्मीकांत प्यारेलाल , लक्ष्मीकांत प्यारेलाल , लक्ष्मीकांत प्यारेलाल , लक्ष्मीकांत प्यारेलाल , लक्ष्मीकांत प्यारेलाल, लक्ष्मीकांत प्यारेलाल , लक्ष्मीकांत प्यारेलाल , लक्ष्मीकांत प्यारेलाल , लक्ष्मीकांत प्यारेलाल
लता मंगेशकर , मंगेश कदम , सुरेश पंडित , सचिन पिळगांवकर , लक्ष्मीकांत प्यारेलाल , लक्ष्मीकांत प्यारेलाल , लक्ष्मीकांत प्यारेलाल , लक्ष्मीकांत प्यारेलाल , लक्ष्मीकांत प्यारेलाल |
| महात्मा गांधी | महात्मा गांधी भी अपनी जिंदगी में मर्यादा का पालन करते थे और अगर वे हिंसा का सहारा ले रहे हैं तो वे भी विचारधारा के शिकार हुए हैं।
महात्मा गांधी राष्ट्रीय ग्रामीण रोजगार गारंटी योजना के तहत संविदा के कर्मचारियों की हड़ताल का असर इंदौर में भी दिखा।
महात्मा गांधी और उनके समकालीन नेताओं का मुख्य ध्यान समाज के अंतिम व्यक्ति को महत्व देने पर था | महात्मा गांधी जयंती समारोह में भाग लेने के लिए तैयार हैं।
महात्मा गांधी ने कहा है कि उनके पास अपने देश को आजादी के बाद पहली बार राष्ट्रपति बनने की पूरी अधिकारिणी क्षमता है। राष्ट्रपति बनने की उनकी क्षमता है। इस देश को लोकतंत्र का युग मिला है। उन्होंने कहा कि देश में लोकतंत्र की व्यवस्था में लोकतांत्रिक व्यवस्था पर विचार करना होगा। राष्ट्रपति की योग्यता पर विचार करना होगा।
महात्मा गांधी ने आज अपने निवास पर एक शोक सभा की अध्यक्षता की । | महात्मा गांधी को हराते हुए पाकिस्तान ने भी इसे स्वीकार नहीं किया।
महात्मा गांधी मैदान में होने वाले कार्यक्रम में बतौर मुख्य अतिथि पहुंचे।
महात्मा गांधी की मूर्ति का एलियन को शामिल कर दिया गया | महात्मा गांधी की हत्या को लेकर 1947 से 1947 के बीच भारत की आजादी तक का जो इतिहास और इतिहास के उस पर खड़े होने वाले जो अनेक विमर्श में वे भारत में आए हैं
महात्मा गांधी की 150वीं जयंती मनाते हुए भारत सरकार ने स्वतंत्रता आंदोलन में उनके महत्वपूर्ण योगदान के लिए भारत सरकार के राष्ट्रपति भवन, नई दिल्ली से उन्हें पद्मश्री पुरस्कार प्रदान किया।
महात्मा गांधी ने एक बार कहा था कि जब हम गांधी जी के विचारों को अपनाएं तो हम महात्मा गांधी से भी महान हो जाएंगे। | महात्मा गांधी जी को श्रद्धांजलि देने के लिए, एक सप्ताह से भी कम समय बचा है, और तब तक इस मुद्दे के खिलाफ देश भर में कई विरोध प्रदर्शन होने वाले हैं, देश के प्रमुख शहरों में, खासकर, गांधीनगर और वाराणसी में, जहाँ वह एक महात्मा गांधी हैं और जहाँ, शायद, उनके अंतिम संस्कार के लिए उन्हें ले जाया जा रहा है।
महात्मा गांधी, जवाहरलाल नेहरू, इंदिरा गांधी , राजीव गांधी , नरसिंहा राव जैसे नेताओं ने गांधी वादी आंदोलन को खड़ा किया था।
महात्मा गांधी के साथ मिलकर चल रहा है |

Table 40: Open end text generation: Paramanu-Hindi 162M v/s BLoom (560M, 1.1B, 3B, 7.1B)

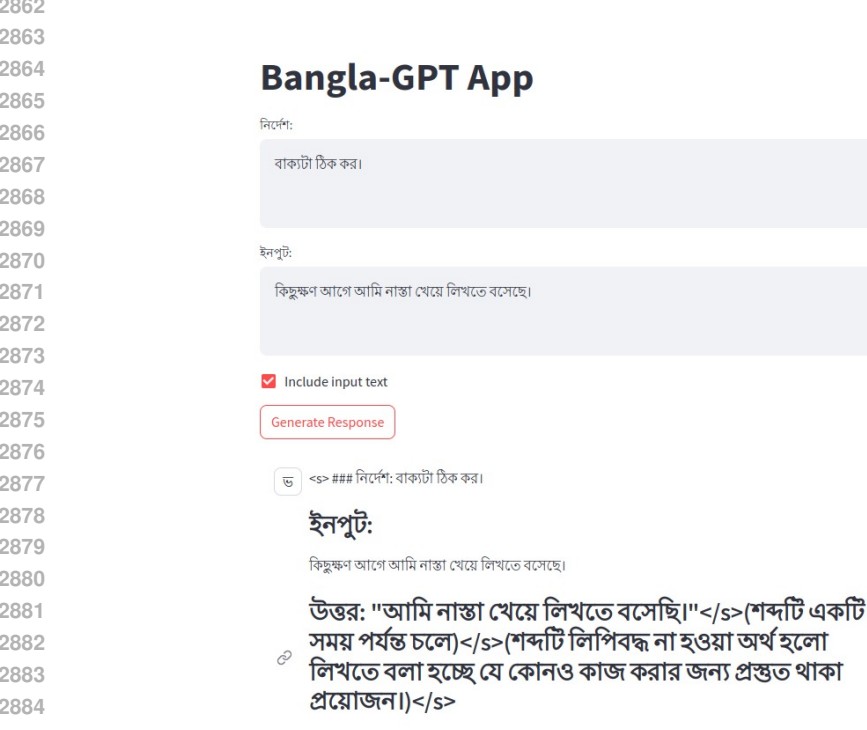

Figure 14: Grammar correction task by Paramanu-Bangla, in-context learning capabilities of Paramanu-Bangla model.

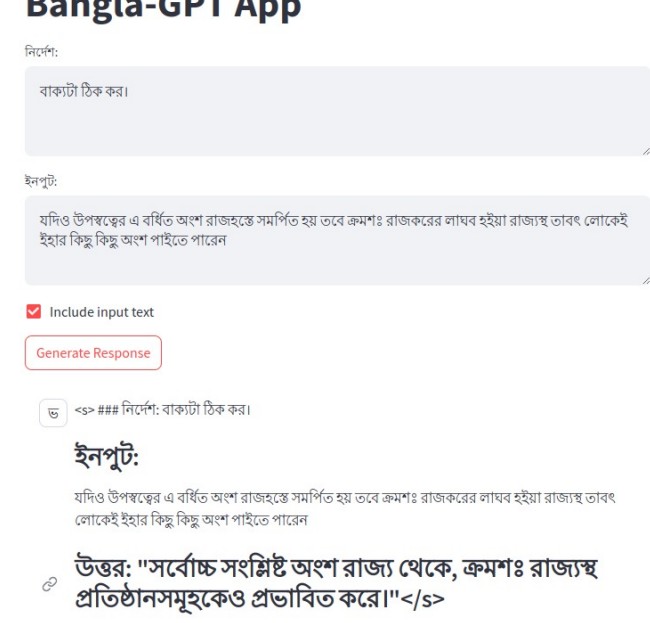

Figure 15: Grammar correction task by Paramanu-Bangla-instruct 108.5M, in-context learning capabilities of the model.

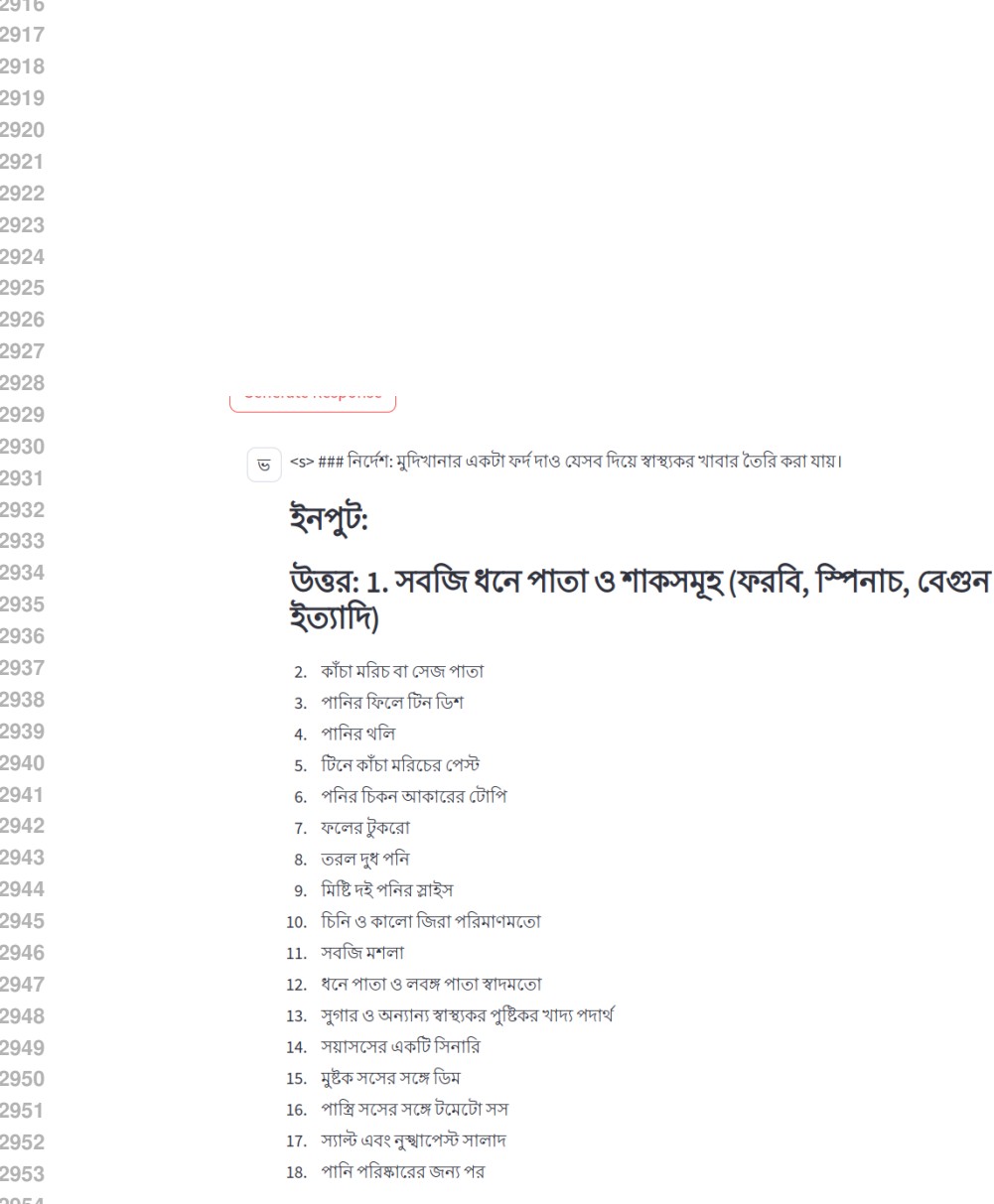

Figure 16: Prompting Paramanu-Bangla-instruct model with query: মুদিখানার একটা ফর্দ দাও যেসব দিয়ে স্বাস্থ্যকর খাবার তৈরি করা যায়। (mudikhānāra ēkaṭā pharda dāō yēsaba diyē sbāsthyakara khābāra tairi karā yāẏa.)

Figure 17: Prompting Paramanu-Bangla-instruct model with query আমাকে প্রাচীন সভ্যতার পাঁচটি উদাহরণ দিন। (āmākē prācīna sabhyatāra pām̐caṭi udāharaṇa dina.)

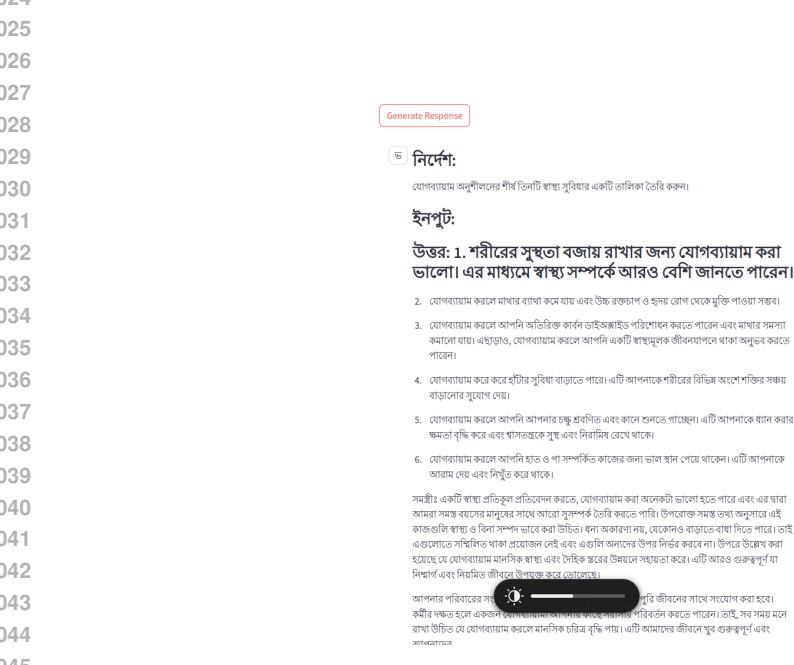

Figure 18: Prompting Paramanu-Bangla-instruct model with query: যোগব্যায়ামের ছয়টি উপকারিতা বলুন। (yōgabyāẏāmēra chaẏaṭi upakāritā baluna.)

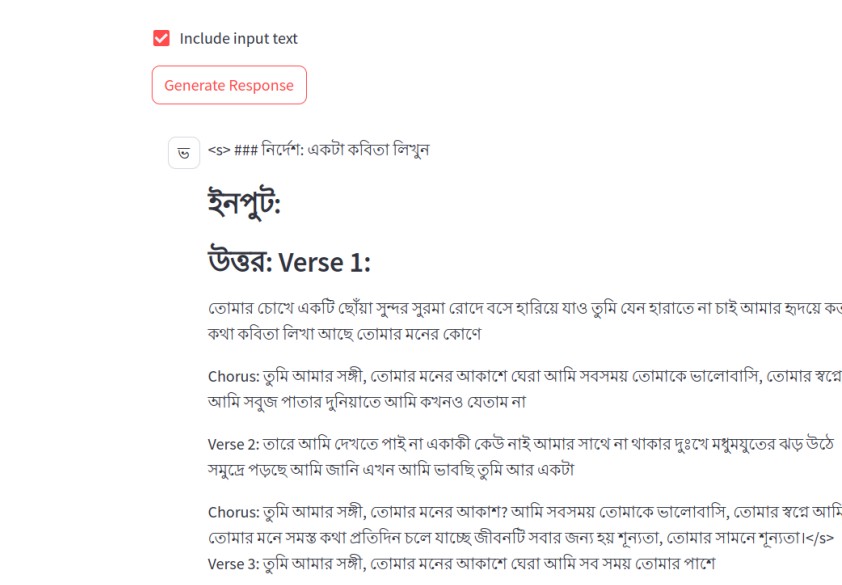

Figure 19: Prompting Paramanu-Bangla-instruct model with query: একটি কবিতা লিখুন। (ēkaṭi kabitā likhuna.)

## Bangla-GPT App

নির্দেশ:

ক্রেডিট কার্ড থাকার সুবিধা এবং অসুবিধা কি কি?।

ইনপুট:

☑ Include input text

Generate Response

ভ নির্দেশ:

ক্রেডিট কার্ড থাকার সুবিধা এবং অসুবিধা কি কি?।

ইনপুট:

উত্তর: 1. ক্রেডিট ব্যালেন্স বাড়ে। ক্রেডিট কার্ড থাকলে বিভিন্ন অসুবিধার সম্মুখীন হতে হয়, যা নিম্নলিখিত হলোঃ

1. প্রোডিউস বিল কমে যেতে পারে।
2. ক্রেডিট কার্ড দ্বারা একটি ক্রেডিট ম্যাপ লিখে প্রদর্শিত হতে পারে যে প্রোডাক্টটি ক্রেতা ক্রয় করতে চান কিন্তু ক্রেতার সাথে দেখা হবে না।
3. আর্থিক ক্ষতি আছে বা না সেটি নির্ভর করে নির্দিষ্ট মূল্যে বিক্রিত হওয়ার সম্ভাবনা রয়েছে।
4. বিনিয়োগে কম হওয়া অথবা হুমকি দেওয়া সম্ভব।
5. পক্ষপাত বা অসত্য তথ্যের সঙ্গে অনলাইন শপিং বা অনলাইন দোকান চালাতে পারেন যা ক্রেতাদের প্রতি আক্রমণ করে। একটি অসুবিধা হলো যে ক্রেডিট কার্ডের উন্নয়নে সমস্যা হয় এবং এটি কম্পানী দ্বারা উন্নয়নের জন্য আবশ্যক হতে পারে। কিন্তু ক্রেডিট কার্ড ব্যবহার করা হলে অন্যান্য ক্রেতাদের সাথে সম্পর্ক স্থাপিত হয় এবং ঈপ

Figure 20: Prompting Paramanu-Bangla-instruct model with query: ক্রেডিট কার্ড থাকার সুবিধা এবং অসুবিধা কি কি? (krēḍiṭa kārḍa thākāra subidhā ēbaṁ asubidhā ki ki?)

ভ  ### নির্দেশ: রবীন্দ্রনাথ ঠাকুরের শৈলী অনুকরণ করতে লিখুন

ইনপুট:

উত্তর: 1. লালনগীতি - রবীন্দ্রনাথ ঠাকুরের লালনগীতির শৈলী জনপ্রিয়তা লাভ করেছে। তার নাটক প্রকাশের সাথে সাথে এটি বিভিন্ন পরিবেশে সৃষ্ট হয়, যেমন সংগীত ও চিত্রকলা - এই রচনার মাধ্যমে প্রকৃতির সৌন্দর্য এবং মানুষের স্বভাবগুলি প্রকাশ করা হচ্ছে।

2. সলিল চৌধুরীর কথা - রবীন্দ্রনাথ ঠাকুরের শৈলী অনুকরণ হল সলিল চৌধুরী এর কথা। তিনি একটি পরিবেশগত এবং সাংস্কৃতিক উন্নয়নের ভাবনার কথা বলেন। তাঁর কথাগুলি খুব সহজ এবং মানসিকতার ব্যাপারে অত্যন্ত স্পষ্ট হয়।

3. অলাদিন্দীপ চক্রবর্তী - রবীন্দ্রনাথ ঠাকুরের শৈলী অনুকরণ হল যে সমস্ত প্রাণীদের সম্পর্কে কথা বলা হয়। তাঁর নাটকগুলি সম্পূর্ণরূপে আধুনিক, নাটকবিশেষতত্ত্ব এবং নাটকীয় উপায়ে পরিচিত হয়েছে।

4. রবীন্দ্রনাথ ঠাকুরের প্যারাগ্রাফ - রবীন্দ্রনাথ ঠাকুরের নাটকে অনেকগুলি পরিবেশগত, সাংস্কৃতিক এবং মানবিক বিষয় থাকে।

Figure 21: Prompting Paramanu-Bangla-instruct model with query: রবীন্দ্রনাথ ঠাকুরের শৈলী অনুক-রণ করতে লিখুন (rabīndranātha ṭhākurēra śailī anukaraṇa karatē likhuna)

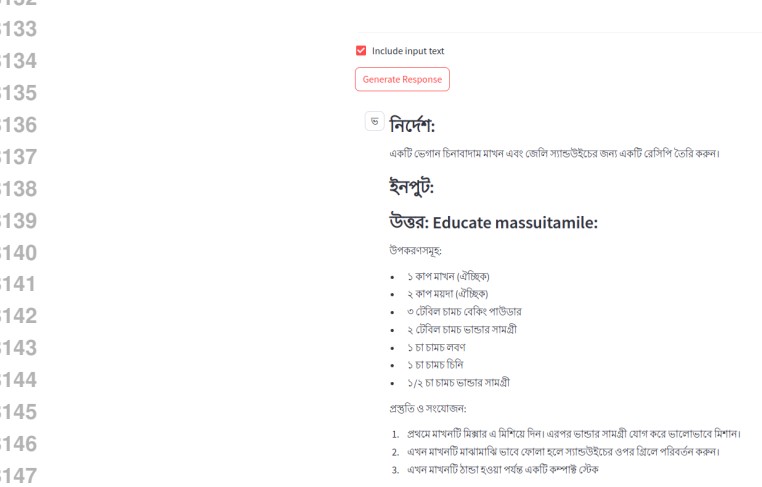

Figure 22: Prompting our instruction tuned Paramanu-Bangla model with query একটি ভেগান চিনাবাদাম মাখন এবং জেলি স্যান্ডউইচের জন্য একটি রেসিপি তৈরি করুন। (ēkaṭi bhēgāna cinābādāma mākhana ēbaṁ jēli syāṇḍa:uicēra janya ēkaṭi rēsipi tairi karuna.)

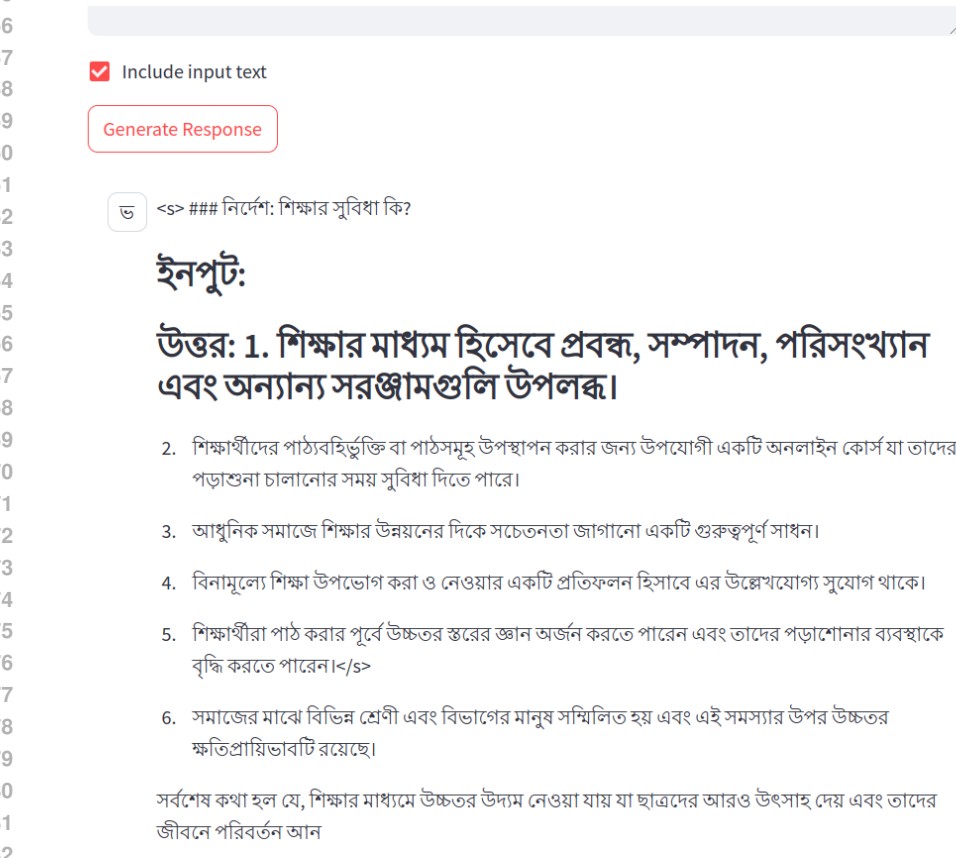

Figure 23: Prompting Paramanu-Bangla-instruct model with query: শিক্ষার সুবিধা কি? (śikṣāra subidhā ki?)

Generate Response

ভ   \ ### নির্দেশ: যৌন শিক্ষার সুবিধা কি?

# ইনপুট:

# উত্তর: "যৌন শিক্ষার সুবিধা" হল যৌন শিক্ষা বা যৌন শিক্ষা। এটি প্রায় সমস্ত মানুষের জন্য উন্নয়নশীল একটি শিক্ষা এবং এটি প্রায় সকল ক্ষেত্রে প্রয়োজনীয় হয়। এতে মানবসম্পদ এবং নৈতিক মূল্যবোধ বিকাশ করে বিভিন্ন ধরণের যৌন সেক্টরে উপকার পায়।\

অন্য দিকে, যৌন শিক্ষা উপকারী হতে পারে কারণ এখানে শিক্ষার্থীরা তাদের স্বতন্ত্র মনের সাথে যৌন সম্পর্কে সন্তুষ্ট হয়ে থাকে। এর মাধ্যমে শিক্ষার্থীরা নিজেদের জ্ঞান ও দক্ষতা উন্নয়নে সাহায্য করতে পারে এবং জীবনে অভিজ্ঞতা অর্জনে সাহায্য করতে পারে। এছাড়াও, যৌন শিক্ষা ব্যবহার করে কোনও সমস্যা হলে তা দূর করার জন্য উন্নয়নশীল পদক্ষেপ গ্রহণ করা যেতে পারে।\ এই সকল কিছু সম্বলিতভাবে যৌন শিক্ষাকে আর্থিক লাভবান করে তোলে শিক্ষক এবং কর্মচারীর জীবন উন্নয়নে সাহায্য করে।\\সুতঃ এইসব সুবিধার সামগ্রী হল যৌন শিক্ষার পরিস্থিতিকে বিবেচনায় নেওয়া এবং যৌন শিক্ষার স্বাধীনতা ও বিবেচনার দিকে ধাপন করা।

Figure 24: Prompting Paramanu-Bangla-instruct model with query: যৌন শিক্ষার সুবিধা কি? (yauna śikṣāra subidhā ki?)

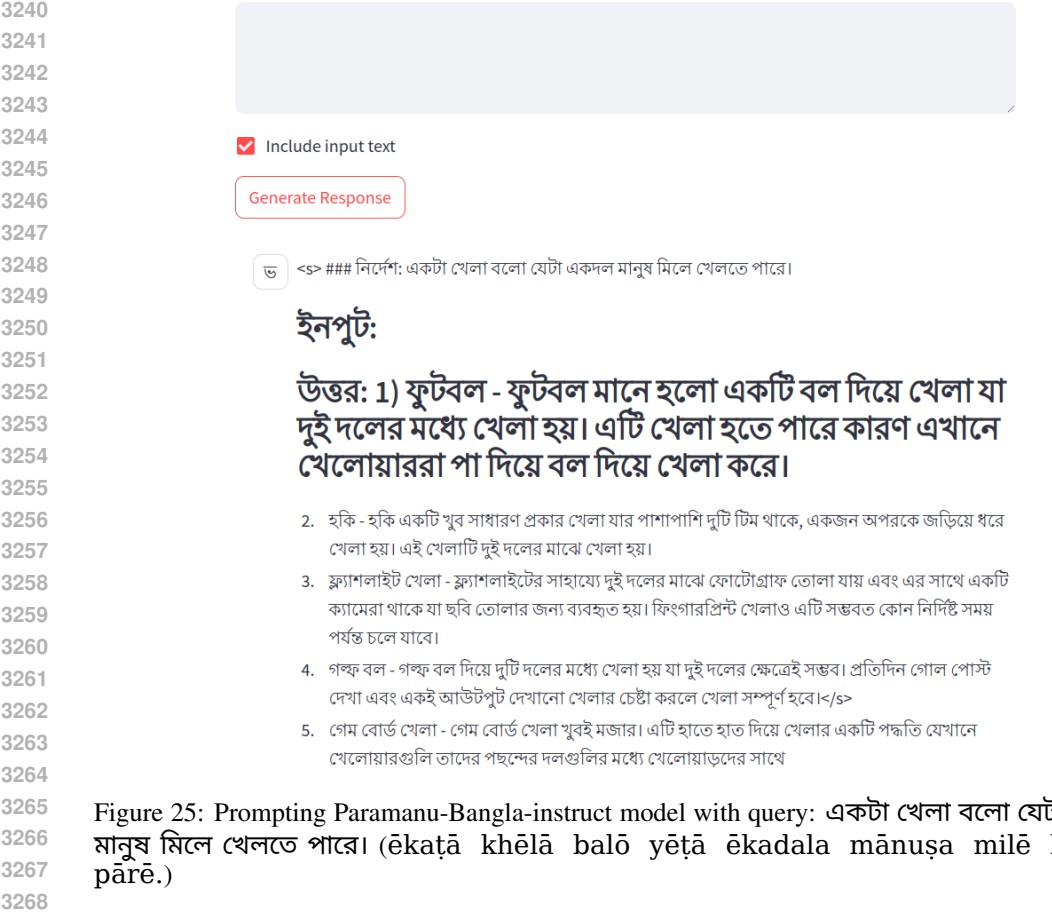

Figure 25: Prompting Paramanu-Bangla-instruct model with query: একটা খেলা বলো যেটা একদল মানুষ মিলে খেলতে পারে। (ēkaṭā khēlā balō yēṭā ēkadala mānuṣa milē khēlatē pārē.)

ভ  ### নির্দেশ: একটা গল্প লিখুন

ইনপুট:

উত্তর: "আমি একটি ছোট মেয়ে। আমার জীবন পরিবর্তিত হয়ে গেছে। আমি কখনো খুশি হতে পারি না আর কখনও হবে না। কিন্তু আসলে একটা মজার জিনিস আছে যা আমাকে বেশ আনন্দ দেয়। তাই আমি ঘরে থাকতেই ভালোবাসি নতুন কিছু।

আমার জীবনের প্রথমদিনগুলোতে আমি একটি অদ্ভুত অভিজ্ঞতা পেয়েছি। আমি ভাবতে পারছি যে মানুষ কখনোই ভুল করেনা ভুল করে এমন কথাগুলো বললে তখন তার সাথে থাকতে কত সুবিধা হবে। আর সেই সময়টাকে আমি কখনো ভুলবো না। এটি আমার জীবনের একটা অসাধারণ অভিজ্ঞতা ছিল।

যখন আমি স্কুলে পড়তাম, আমি সব সময় খুব খুশি হতে পারতাম এবং আমার শিক্ষার্থীদের উন্নতি করতে হতো। স্কুলের ক্লাসে সবাই আমাকে দেখতে আসত হাসাতে হাসাতে। কিন্তু স্কুল থেকে চলে এসে একটি দিনগুলো আমার জীবনে ভালো লেগে গেল। আমি একটা ছাতা নিয়ে স্কুলে উঠে গেলাম এবং স্কুলের ফার্নিচার কিনে ফিরে এসে দেখি একটা বাথরুমের দরজা খোলা। সেখানে আমি পরিষ্কার ভাবে খাবার বানিয়ে নিলাম এবং সবাইকে উপহার দিলাম।

শুরুতেই আমি সকল বন্ধুদের কথা শুনে আসলাম কিন্তু সেই ছোট ঘটনা আমার মনেও কখনোই শান্তি আনবে না। আমি ভাবছিলাম আমার বয়স আরো বাড়বে কিনা, কিভাবে আমি এত কিছু হারানো হবো। কিন্তু তখন স্কুলের পরিবেশ খুব বিচিত্র ছিল। সেখানে সবাই মজার মজার কথা শেয়ার করছিল। তাদের হাসি আমার মুখে লালচে ফুটে যাচ্ছিল এবং একটি সুন্দর স্মৃতি আমার হৃদয়ে জন্ম নেওয়া হয়েছিলো।

এখন এই বিষয়টি মনে পড়ে আমি স্কুলে যাচ্ছি না, তবে আমি যখন স্কুলের পরিবেশ সম্পর্কে ভাবতে শুরু করলাম, তখন আমার মনে হলো যে, এই রকম একটা স্থিতি সবার জন্য সুখ, সম্মান এবং প্রতিষ্ঠার ক্ষেত্রে। এটা সম্ভব যদি আমি নতুন কিছু শিখে যাই, তাহলে আমার জীবনে একটা পরিবর্তন আসতে পারে।

Figure 26: Paramanu-Bangla 108.5M generation capabilities at maximum tokens limit of 1024 for story generation in Bangla. Query: একটি দীর্ঘ গল্প লিখুন (ēkaṭi dīrgha galpa likhuna)

আমার স্কুলে পরিবেশ সম্পর্কে আমরা শেখা শিখলাম, যদিও সেই স্কুলে থাকতে খুব একটা সহায়তা ছিলো না। কিন্তু আমরা যখন বাসায় খেলতাম, তখন আমাদের মনে হতো নতুন কিছু শিখেছি। আমি এখন জানি সেটার উপর নির্ভর করে অনেক কঠিন টেকনোলজি দরকার হবে। আমার স্কুলে থাকতে থাকতে মনে হতে থাকে আমাদের জীবনে সমস্যা আর একটি স্কুলে থাকতে হলে সেই স্কুলাতে নিজেকে সাম্প্রতিক উপাদানগুলি বিনিময় করা উচিত। তারপরও, আমি নিশ্চিত হই যে আমরা সমস্যার সমাধানে সাফল্য অর্জন করতে পারব।

পরিবেশ সম্পর্কে কথা বলতে গেলে আমি জানতে পেরেছি যে আমাদের সবচেয়ে গুরুত্বপূর্ণ জিনিস হলো নিজেদের সুস্থ রাখা। নিজের শরীর স্বাস্থ্য পরিচালনার জন্য একটি কম্পিউটারের প্রয়োজন হয়, যাতে তার সব উপকরণ স্বাস্থ্যকর থাকে। পরিবেশের উপর কন্ট্রোলের মাধ্যমে নির্ভুল কাজ করার ব্যবস্থা নিয়ে শিক্ষা নেওয়ার চেষ্টা করা উচিত। শিক্ষণ প্রণালী একটি আদর্শ পদক্ষেপ যা দক্ষতা ও প্রযুক্তি উন্নয়নে আমাদের সাহায্য করবে।

পরবর্তীতে, আমাদের বিভিন্ন শিক্ষা পদ্ধতি একত্রিত করে একটি সমর্থনকারী দল গঠিত হয়েছে যারা বিভিন্ন প্রকল্পে অংশগ্রহণ করেছে। উদাহরণস্বরূপ, আমাদের স্কুলের পড়াশোনার জন্য সমস্ত বিবরণগুলি সংগ্রহ করা হয়েছে। তারপর একটি সম্মেলনের সময়, শিক্ষকরা ছাত্র-ছাত্রীদের মধ্যে কথোপকথন চালু করে দিয়েছেন এবং সবাইকে একটি নির্দিষ্ট পরিমাণের সম্মতি দিতে বলেছেন। শিক্ষকরা এক্সপান্ড করেছেন একটি স্থায়ী সম্মতি।

পরিবেশে পরিবর্তন এবং সামাজিক উন্নয়নের উদ্দেশ্যে এই সমর্থনকারীরা ছাত্র-ছাত্রীদের পাঠশালায় উপযোগী পাঠ প্রদর্শন করেছেন। এটি শুভ হোক এবং শুভ হোক পরের পরিকল্পনাটি স্কুলে শিক্ষকদের নেতৃত্বে পরীক্ষা দেওয়া এবং সমস্যার সমাধান করা। শিক্ষকরা ছাত্র-ছাত্রীদের কাছে একটি সম্পূর্ণ শিক্ষামূলক অনুভূতি দেওয়ার সুযোগ প্রদান করে এবং তারা সমস্যার সমাধান করার জন্য সর্বোচ্চ মানসম্পন্ন হন।

এই পদ্ধতি সম্পূর্ণ নতুন করে কিছু সমস্যাগুলি সমাধান করা হয় এবং আমরা আগের চেয়ে অনেক বেশি চেষ্টা করি যাতে কোনো সমস্যা না হয়। প্রথমবারের মতো, আমার মতো মানুষদের জন্য এই একটি সমর্থনকারীর গল্পে পরিচিত হওয়া গল্পটি শুরু হল। এখানে একটি প্রশ্ন উঠে এসেছে এবং প্রশ্নটি ছিল - কিভাবে একটি স্কুল শিক্ষক এবং শিক্ষার্থীরা একটি সমস্যার সমাধান করতে পারে? এই গল্পে, আমরা জানি যে কীভাবে শিক্ষক এবং শিক্ষার্থীদের পেছনে পিছনে পথ চলা যায়। পরবর্তী প্রশ্ন হল - এই গল্পটি কী ভাবে আমাদের সমস্যা সমাধান করে? এখানে একটি

Figure 27: Long story generation, continued

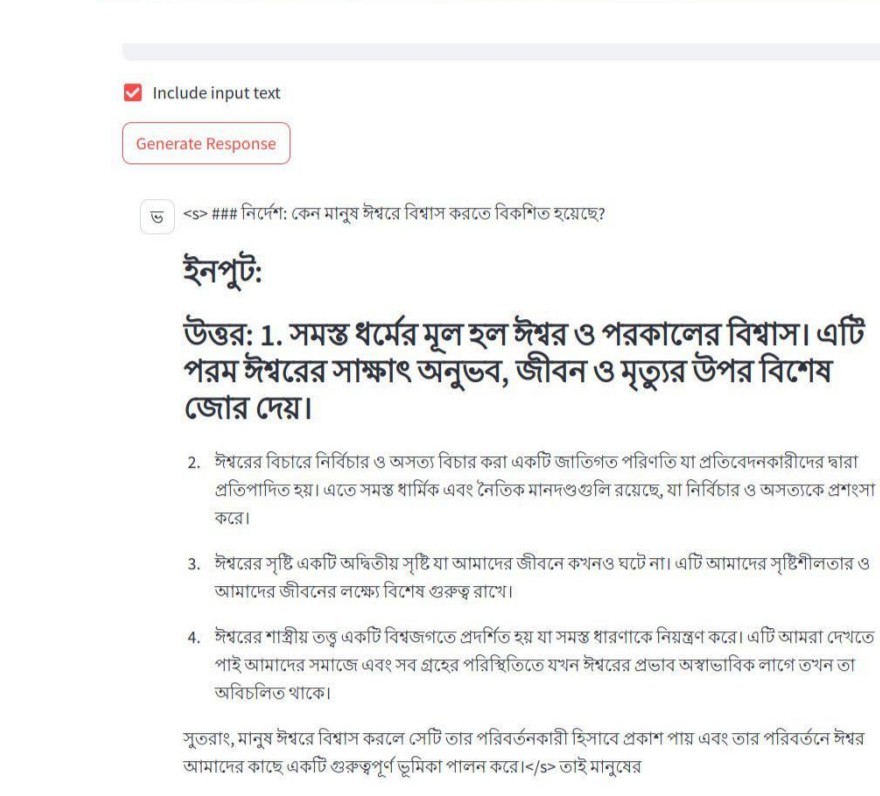

Figure 28: Prompting Paramanu-Bangla-instruct model with query: কেন মানুষ ঈশ্বরে বিশ্বাস করতে অনুপ্রাণিত হয়েছে? (kēna mānuṣa īśbarē biśbāsa karatē anuprāṇita haẏēchē?)

अनुदेश:

मुझे उन चीजों की सूची दें जो मैं रात में बेहतर नींद के लिए कर सकता हूँ

इनपुट:

☑ Include input text

Generate Response

भ | \<s\> ### अनुदेश: मुझे उन चीजों की सूची दें जो मैं रात में बेहतर नींद के लिए कर सकता हूँ

### इनपुट:

### उत्तर: 1. समय पर खाना खाना और सुबह जल्दी उठना, नींद को अच्छी तरह से सुनिश्चित करना और अपने स्वस्थ जीवन के लिए सही खान-पान व्यवस्था करना।

2. नियमित भोजन लेना और स्वस्थ भोजन का प्रयोग करना।
3. दिनभर कार्य और टेक्स्टिंग जैसे ध्यान न देना।
4. रात के खाने में सभी कार्यस्थल पर खाना या कॉफी का संचार करना।
5. नींद की पूरी ताकत रखना।
6. सुतली अंग पशु चिकित्सा की सलाह देने वाले विशेषज्ञ से संपर्क करना।\</s\>
7. दैनिक स्मार्टफोन या भिकिनरी के जरिए रात को साफ सुथरा और निरोगी बनाना।
8. रात को धोने के अलावा अपने मोबाइल फोन या फ्रूट से जुड़े रहने का प्रयास करें।
9. रात्रि के भोजन में सब्जियों का उपयोग करना जो कि अवसादित कम करें।
10. नींद पूरी न करने से तुरंत नाश्ता करना, लकड़ी से खाना खाना और बेडरूम में खाना शामिल है।
11. रात में अधिक से अधिक फल, सब्जी, फलकोल्ट ड्रिंक और स्वस्थ वस्तुओं से समय बिताना स्वास्थ्य

Figure 29: Prompting Paramanu-Hindi-instruct 367.5M model with query: मुझे उन चीजों की सूची दें जो मैं रात में बेहतर नींद के लिए कर सकता हूँ (mujhē una cījōṁ kī sūcī dēṁ jō maiṁ rāta mēṁ bēhatara nīṁda kē liē kara sakatā hūm̐)

Figure 30: Prompting Paramanu-Hindi-instruct 367.5M model with query: अफ़्रीकी सरकारें हवाई याता-यात कनेक्टिविटी में सुधार के लिए क्या कर रही हैं? (afrīkī sarakārēṁ havāī yātāyāta kanēkṭiviṭī mēṁ sudhāra kē liē kyā kara rahī haiṁ?)

Figure 31: Prompting Paramanu-Hindi-instruct 367.5M model with query: मुझे छुट्टी मनाने कहाँ जाना चाहिए? (mujhē chuṭṭī manānē kahāṁ jānā cāhiē?)

Figure 32: Prompting Paramanu-Hindi-instruct 367.5M model with query: भोजन तैयार करने के लिए स्वास्थ्यप्रद खाना पकाने के तरीके क्या हैं? (bhōjana taiyāra karanē kē liē svāsthyaprada khānā pakānē kē tarīkē kyā haiṁ?)

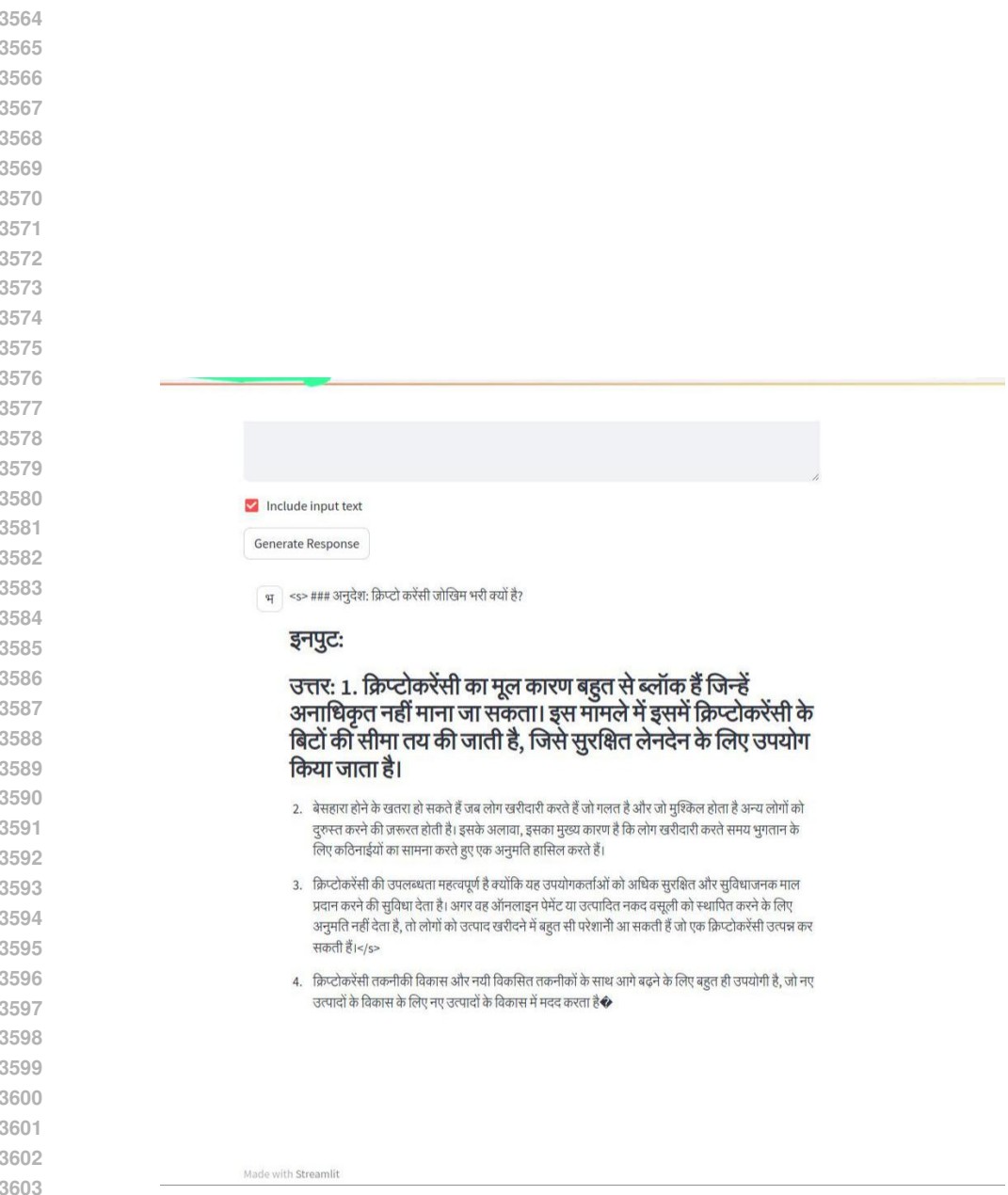

Figure 33: Prompting Paramanu-Hindi-instruct 367.5M model with query: क्रिप्टो करेंसी जोखिम भरी क्यों है? (kriptō karēṁsī jōkhima bharī kyōṁ hai?)

Figure 34: Prompting Paramanu-Hindi-instruct 367.5M model with query: क्या आप बीमारियों की रोकथाम में डिजिटल स्वास्थ्य की भूमिका को बुलेट बिंदुओं में सूचीबद्ध कर सकते हैं? (kyā āpa bīmāriyōṁ kī rōkathāma mēṁ ḍijiṭala svāsthya kī bhūmikā kō bulēṭa biṁduōṁ mēṁ sūcībaddha kara sakatē haiṁ?)

इनपुट:

☑ Include input text

Generate Response

भ `` ### अनुदेश: दिए गए पाठ्यक्रम के लिए एक पाठ्यक्रम डिज़ाइन करें। छात्रों को प्रत्येक अध्याय के उद्देश्य की संक्षिप्त व्याख्या के साथ अध्यायों की एक सूची दी जानी चाहिए। हर किसी के लिए प्रोग्रामिंग (पायथन के साथ शुरुआत करना)

## इनपुट:

## उत्तर: 1. विषय - एक अध्यापक के लिए कुछ प्रोग्राम लिखने का नीति होना चाहिए।

2. सम्पूर्णता - एक अध्यापक के लिए, उनके उद्देश्यों और प्राथमिकताओं को ध्यान में रखते हुए उनकी शिक्षा और संबंधित पाठ्यक्रमों को संचालित करने का नीति होना चाहिए।

3. योजनाबद्धता - छात्रों को योजनाबद्धता के अंतर्गत एक अध्यापक के बारे में लेख पढ़ने और उन्हें उनके विषय से संबंधित विषयों पर विस्तृत जानकारी देनी होगी।

4. सामग्री संपादित करना - हम छात्रों को उसकी सामग्री संपादित करेंगे जैसे कि उनकी मुख्य भूमिका, विशेषताएं, विषय, संबंधित अंग्रेजी व्याकरण समस्या आदि।

5. संशोधन आवश्यकताएं - छात्रों को उसके परिणामों के बारे में अपडेट देकर उन्हें संशोधित तथा उन्नत करना होगा।

6. गुणवत्ता लेवल - छात्रों को संभवतः संग्रहित विवरण को संशोधित एवं प्रस्तुत करने के लिए उन्हें पूरी तरह से समर्थित किया जाना चाहिए।

7. निरंतर अद्यतन - इस नीति के तहत छात्र निरंतर बदलती और सुधार करते हुए अपनी प्रगति को संभालते हुए अपने विषय के पक्ष में निरंतर सुधार करते हुए एक अच्छा शिक्षक बनाएंगे।``

8. कार्य क्रमिक रूप से संचालित - छात्रों को एक संगठित तरीके

Figure 35: Prompting Paramanu-Hindi-instruct 367.5M model with query: दिए गए पाठ्यक्रम के लिए एक पाठ्यक्रम डिज़ाइन करें। छात्रों को प्रत्येक अध्याय के उद्देश्य की संक्षिप्त व्याख्या के साथ अध्यायों की एक सूची दी जानी चाहिए। हर किसी के लिए प्रोग्रामिंग (पायथन के साथ शुरुआत करना) (diē gaē pāṭhyakrama kē liē ēka pāṭhyakrama ḍizāina karēṁ. chātrōṁ kō pratyēka adhyāya kē uddēśya kī saṁkṣipta vyākhyā kē sātha adhyāyōṁ kī ēka sūcī dī jānī cāhiē. hara kisī kē liē prōgrāmiṁga (pāyathana kē sātha śuruāta karanā))

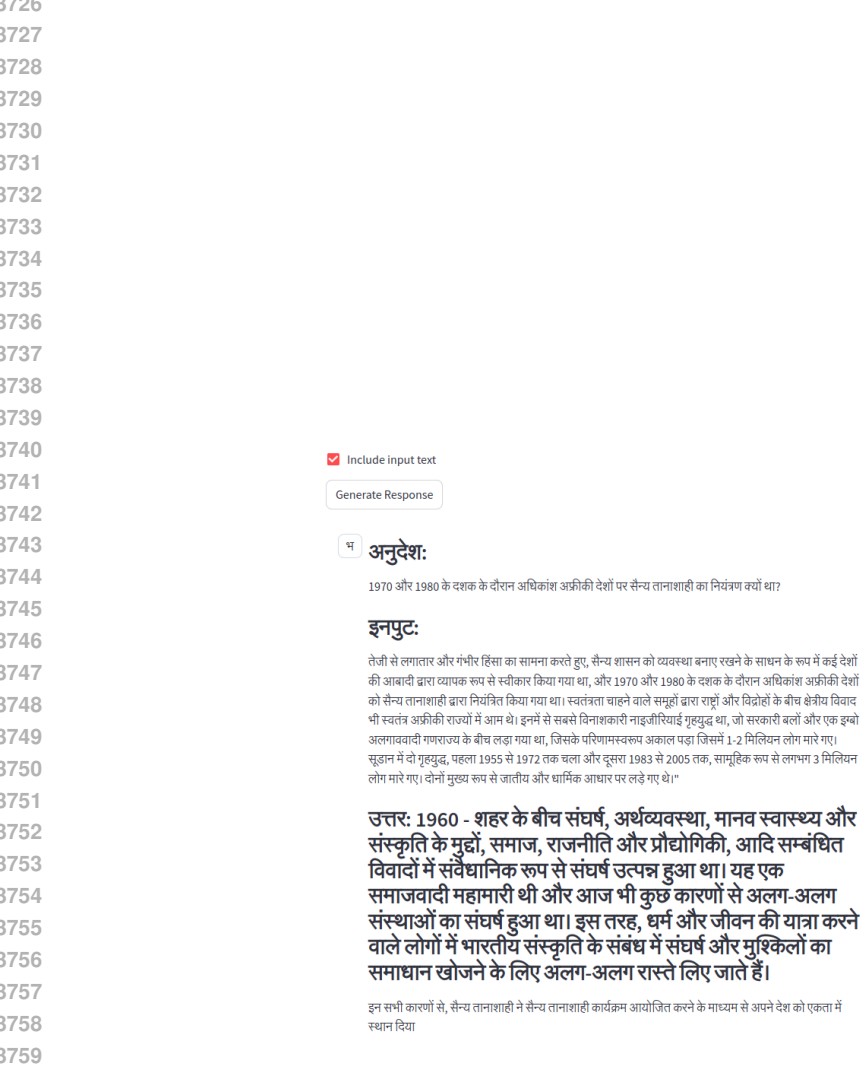

Figure 36: Prompting Paramanu-Hindi-instruct 367.5M model with query: 1970 और 1980 के दशक के दौरान अधिकांश अफ्रीकी देशों पर सैन्य तानाशाही का नियंत्रण क्यों था? 1970 aura 1980 (kē daśaka kē daurāna adhikāṁśa aphrīkī dēśōṁ para sainya tānāśāhī kā niyaṁtraṇa kyōṁ thā?)

भ  ### अनुदेश: प्रेम क्या है?

**इनपुट:**

उत्तर: । प्रेम एक भाव होता है जो दूसरे का प्रतिबिम्ब बनाता है और स्नेह की भावना दर्शाता है। यह विश्वास, संवेदना और अंतर्निहित भावनाएं, जब आप महसूस करते हैं कि आपकी इच्छा किसी भी व्यक्ति या स्थिति के लिए स्थिर नहीं होती हैं, आपके जीवन में एक उज्ज्वल स्वर्णिम धारा बनता है। प्रेम एक भावना का रूप हो सकता है, जो आदर्शों से भरा हुआ होता है जो एक दूसरे को मदद करने में सक्षम होते हैं।प्रेम एक ऐसी भावना होती है जिसमें हम स्वयं अपना समय, ऊर्जा या अनुभूति समझते हुए अन्य लोगों की आवाजों, जातिवाद, लिंग अथवा नास्तकों आदि से जुड़ जाते हैं। इस भावना को याद रखना और उसे अपने जीवन में लागू करना हमारी जिम्मेदारी है। इसलिए, प्रेम एक महत्वपूर्ण मानवीय सामाजिक वातावरण है जो हमें अपने समाज में एक बेहतर, संतुष्ट और संतुलित नागरिक बनाती है। प्रेम एक जीवन का आधार और एक महत्वपूर्ण अंग होता है, जहां हर इंसान कहीं-कहीं एकता के लिए जिम्मेदार होता है, जहाँ सच्ची भावनाएँ नहीं होतीं हैं। इसलिए, प्रेम हमारे जीवन का सबसे मूल आधार है।

Figure 37: Prompting Paramanu-Hindi-instruct 367.5M model with query: प्रेम क्या है? (prēma kyā hai?)

अनुदेश:

कौन सा खेल दुनिया भर में सबसे लोकप्रिय है?

इनपुट:

☑ Include input text

Generate Response

भ ` ### अनुदेश: कौन सा खेल दुनिया भर में सबसे लोकप्रिय है?`

## इनपुट:

## उत्तर: 1. फुटबॉल

2. क्रिकेट
3. टेनिस
4. बास्केटबॉल
5. हॉकी
6. बॉक्सिंग
7. कबड्डी

Figure 38: Prompting Paramanu-Hindi-instruct 367.5M model with query: कौन सा खेल दुनिया भर में सबसे लोकप्रिय है? (kauna sā khēla duniyā bhara mēṁ sabasē lōkapriya hai?)

⊔ **அறிவுறுத்தல்:**

கல்லூரிப் பட்டப்படிப்பைத் தொடர்வதன் 5 நன்மைகளைப் பட்டியலிடுங்கள்

**உள்ளீடு:**

**பதில்: 1. கல்லூரி மாணவர்களுக்கு பயனுள்ள படிப்புகளில் உயர் தகுதி மற்றும் நிர்வாக முறைகள் உள்ளன.**

2. படிக்க வேண்டிய பல்வேறு வகுப்பில் பெற்றுள்ள உயர் தகுதி தொழிலியங்களின் மீது உயர் தகுந்திருக்கும் மகிழ்ச்சி உள்ளது.

3. படிக்கவேண்டிய பல்வேறு பயிற்சிகளின் முனைவராக இது தொடர்பான அறிவை மூட்டும்உள்ளது.

4. இந்தியாவின் எல்லா பகுதிகளிலும் சமூகம் மற்றும் ஆராய்ச்சியில் உயர்வு அடைவுகள் உள்ளன.

5. பல கல்லூரிகளில் படிக்கவும், அதன் பின்னர் தொழில்நுட்பக் கல்லூரிகளை தேர்ந்தெடுக்கும் மாணவர்களுக்கு கிடைக்கும் பட்டம் வாழ்வில் ஆரம்பம் செய்து வழிகாட்டும்.6. படிக்கவேண்டிய பொறியியல் படிப்புகள் தங்கள் அல்லது அனுமதிக்கப்பட்ட மாணவர்கள் சாதனை மற்றும் முயற்சிகள் இருந்து வெளியே வந்து குழு வேலை மேம்படுத்த முடியும்.

6. ஒரு நாட்டின் பெரும்பாலான நகரங்களில் வசிக்கும் மக்களுக்கு உயர் தகுதி மற்றும் அரசியலுக்கு அணுகல் உள்ளது.

7. வாழ்க்கையின் உயர் தன்மை மற்றும் வேளாண்மை போன்ற விஷயங்கள் கல்லூரியில் கிடைக்கும் என்பதை எங்களுக்கு அறியலாம்.9. தமிழ்நாட்டில் புதிய தொழில்நுட்ப படிப்புகளுக்கு ஏற்றுக்கொள்ளும் முன்னணி மாற்றங்கள் பல நகரங்களில் கடன் செலவு செய்யபடியும். 1

Figure 39: Prompting Paramanu-Tamil-instruct model with query: கல்லூரிப் பட்-டப்படிப்பைத் தொடர்வதன் 5 நன்மைகளைப் பட்டியலிடுங்கள். (kallūrip paṭṭappaṭippait toṭarvataṉ 5 naṉmaikaḷaip paṭṭiyaliṭuṅkaḷ.)

அறிவுறுத்தல்:

புவிசார் தரவு என்றால் என்ன?

உள்ளீடு:

பதில்: ☼ புரம் புவிசார் தரவு என்பது பூமியின் அடிப்படை தரவுகளை வழங்கும் வழிகளை வழங்கும் கணினி நிரல் என்று குறிப்பிடுகின்றது. இது புவியின் மொத்த இயற்கை நிலங்கள் மற்றும் உயிரினங்களை பற்றி ஆராய்ந்து விளக்கப்படுகின்றது. இதில் உள்ள கணினி நிரல்கள் பல பிரபல நிரல்களின் பயன்பாடுகளைக் கொண்டு வருகின்றன. இவைகளில் சிலவேனும் பூ மற்றும் காய்கறிகளுடன் தொடர்பு கொண்டுள்ளதாக இருக்கின்றன.

புவிசார் தரவு போன்றது பூவின் அடிப்படை தரவுகளைளிக்கும் பயனர் நிரல்களின் மூலம் மட்டுமே வழங்கப்படுகின்றது. இது புவியின் பெரும்மையான பிரிவுகள் மற்றும் உயிரினங்களை ஆராய்ந்து அறியலாம். இது மிகவும் பயனுள்ள ஒரு செயல் ஆகும் மேலும் மக்கள் மற்றும் நிரல் இயக்கக் குழுக்களுக்கு பயன்படுத்தி சிறகுவதுமான பலவீனத்தை பெறும்.புவிசார் தரவு இவ்வாறு எவ்வாறு பயன்படுத்தப்படுகின்றது என்பதை புரிந்துகொள்ள இதை arltamil.com இல் தொடர்பு கொள்ளவும்.

Figure 40: Prompting Paramanu-Tamil-instruct model with query: புவிசார் தரவு என்றால் என்ன? (puvicār taravu eṉṟāl eṉṉa?)

அறிவுறுத்தல்:

ஒரு பல்பொருள் அங்காடியின் பால் பிரிவில் நான் காணக்கூடிய ஐந்து உணவுகளில் புள்ளி புள்ளிகளைப் பயன்படுத்தி பட்டியலை உருவாக்கவும்

உள்ளீடு:

பதில்: 1. தக்காளி பருப்பு

2. பயர் பருப்பு

3. பருப்பு கருப்பு

4. உளுந்து பருப்பு

5. பொட்டம் துாள் கருத்து: பருப்பு ஒரு சிறிய அளவில் நல்ல உணவு உள்ளது எனவே எனக்கு தெரியும் பொருள் இது ஆகும். முட்டை, கொத்துக்கறி மிளகாய் பொடி மற்றும் நெய் சேர்த்து கலந்து குடித்தால் மிகவும் சுவையாக இருக்கும்.

உருளைக்கிழங்கு, பச்சைத்தக்காளி, பீன்ஸ், பொட்டுப்பயிறு போன்றவற்றை பயன்படுத்தி பிரியாம் செய்த மாதிரி உணவுக்கு பயன்படுத்தலாம். புல்லிகள்: 50 கிராம் புலா, உப்பு, மஞ்சள் பொடி, பிரண்டு ஆகியவை சேர்ந்த கரண்டி. பதிவு போட்டி:

1. தக்காளி பருப்பு தையல் போடுவதற்கு உபயோகிக்கப்படுகின்றது. கழுத்தை நீர் கட்டிக்கொண்டு அதில் கசிக்கவும் பின்னர் தென்னை தேங்காய் துண்டு செய்வதற்கும் பயன்படும்.

2. பருப்பு அரை செய்து உடனடியாக சாப்பிடுங்கள்.

3. கொத்து கணக்கு எலும்பு சேர்குங்கள் அல்லது சமையல் செய்யும்ᬉங்கள்.

4. வெங்காயம் பொடி சேர்க்குகின்றோம். இதை தேன் மற்றும் ஐஸில் பருத்தி சேர்க்கவும் ஒரு கப் என்று கொண்டு சாப்பிடுங்கள்.

Figure 41: Prompting Paramanu-Tamil-instruct model with query:ஒரு பல்பொருள் அங்-காடியின் பால் பிரிவில் நான் காணக்கூடிய ஐந்து உணவுகளில் புள்ளி புள்ளி-களைப் பயன்படுத்தி பட்டியலை உருவாக்கவும் (oru palporuḷ aṅkāṭiyiṉ pāl pirivil nāṉ kāṇakkūṭiya aintu uṇavukaḷil puḷḷi puḷḷikaḷaip payaṉpaṭutti paṭṭiyalai uruvākkavum)

உள்ளீடு:

பதில்: ○ாம் நாட்டின் கருத்து மக்களிடம் தேசிய பூங்காக்களை விரும்புகின்றன. இந்த தேசிய பூங்காக்கள் மக்களுக்கு மிகவும் முக்கியமானவைகளாக உள்ளனவும் அவற்றுக்கு மக்கள் தங்க மூலம் முன்பதிவு செய்யப்படுகின்றனர். இது மக்கள் மதிப்பில் தேசியபூங்காக்களின் மதிப்புகளை அறிய உதவுகின்றது. மேலும், மக்கள் தங்கள் தேசிய பூங்காக்களுக்குத் தேவையான நிகழ்வுகளை தெரிந்து கொள்ள உதவுகின்றனர். இது மக்களின் மென்மையை உயர்த்துகின்றது மற்றும் அக்கறைக்கு உதவுகின்றது.

ஒரு பூங்காக்களில் புதிய கருத்துக்கள் மற்றும் பயிற்சிகள் இருக்கும். இவை மக்கள் தேசிய பூங்காக்களில் வருகின்ற உற்பத்திகளை விட மிகவும் முக்கியமானவை என கருதப்படுகின்றது. இது மக்களின் கருத்துக்களை மேம்படுத்துகின்றது மற்றும் அவர்களின் வாழ்க்கைக்கு உதவுகின்றது. இது மக்கள் தேசிய பூங்காக்களுக்குத் தேவையான நிகழ்வுகளை மட்டும் அல்லது அவர்கள்ent○ ிகை பள்ளிகளுக்கு உதவுகின்றது. இது மக்களின் உணர்வுகளை மேம்படுத்துகின்றது மற்றும் அவர்கள் எப்படி தேசிய பூங்காக்களுக்கு உதவ முடியும் என்று தெரியும்.அதனால் மக்கள் தேசிய பூங்காக்களை காண வேண்டும் மற்றும் அதன் பின்புறம் இருந்தாங்க வேண்டும். இது அவர்களின் வாழ்க்கைக்கு உதவுகின்றது மற்றும் அவர்களின் பகிர்வுகளுக்கு மேற்பட்ட வாய்ப்புகளை வழங்குகின்றன.பாரா தகுதி ஆனது தேசிய பூங்காக்கள்! தேசிய பூங்காக்கள் மிகவும் முக்கியமாக உள்ளான. இது மக்களின் உயர்வுகளை போக்குகிறது மற்றும் அவர்களின் வாழ்வில் உதவுகின்றது. எனவே, மக்கள்

Figure 42: Prompting Paramanu-Tamil-instruct model with query:குடியிருப்பாளர்களுக்கு தேசிய பூங்காவின் சில நன்மைகள் என்ன?? (kuṭiyiruppāḷarkaḷukku tēciya puṅkāviṉ cila naṉmaikaḷ eṉṉa?)

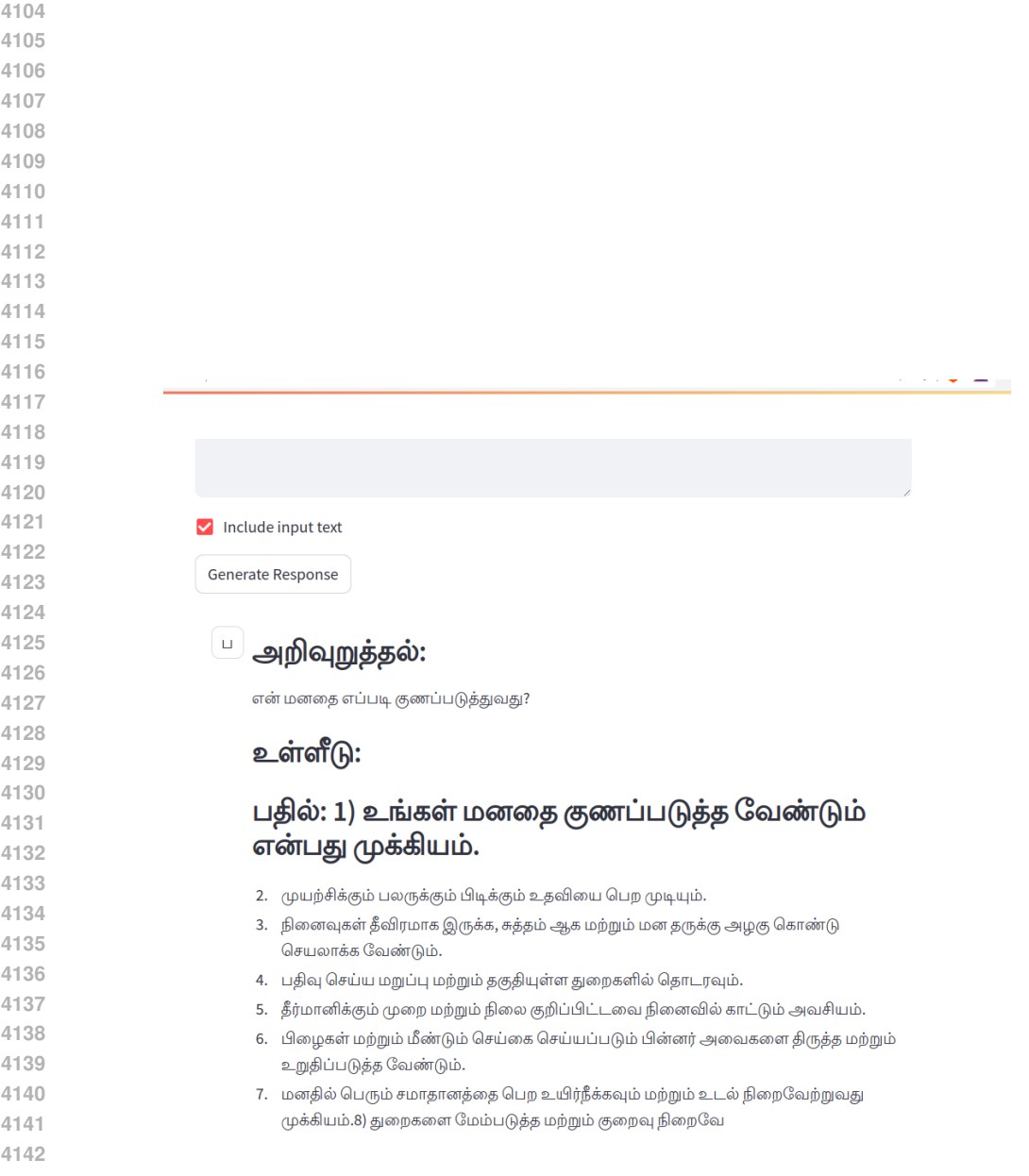

Figure 43: Prompting Paramanu-Tamil-instruct model with query: என் உணர்வுகளை எப்படி குணப்படுத்துவது? (eṉ uṇarvukaḷai eppaṭi kuṇappaṭuttuvatu?)

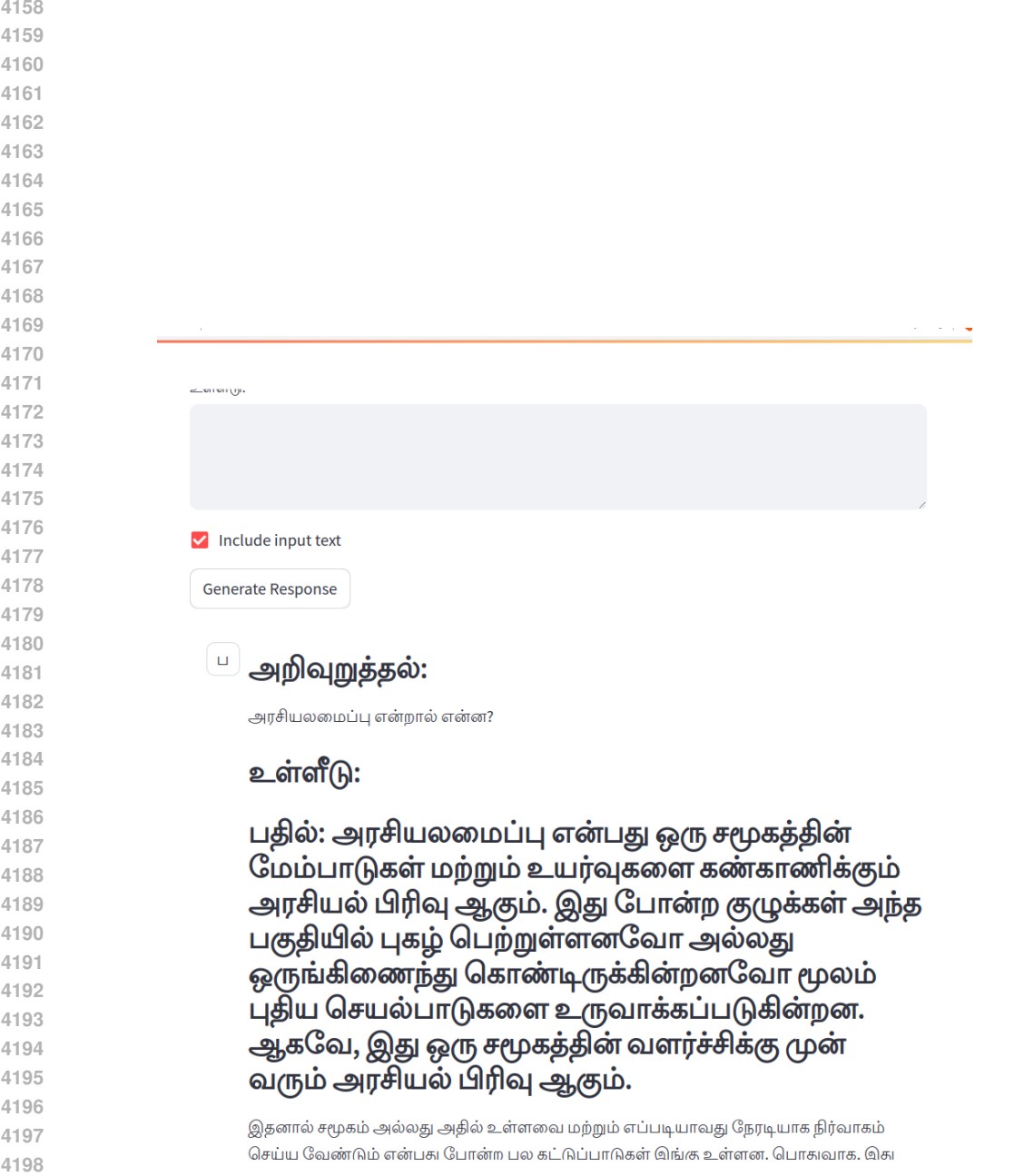

Figure 44: Prompting Paramanu-Tamil-instruct model with query: அரசியலமைப்பு என்றால் என்ன? (araciyalamaippu eṉṟāl eṉṉa?)

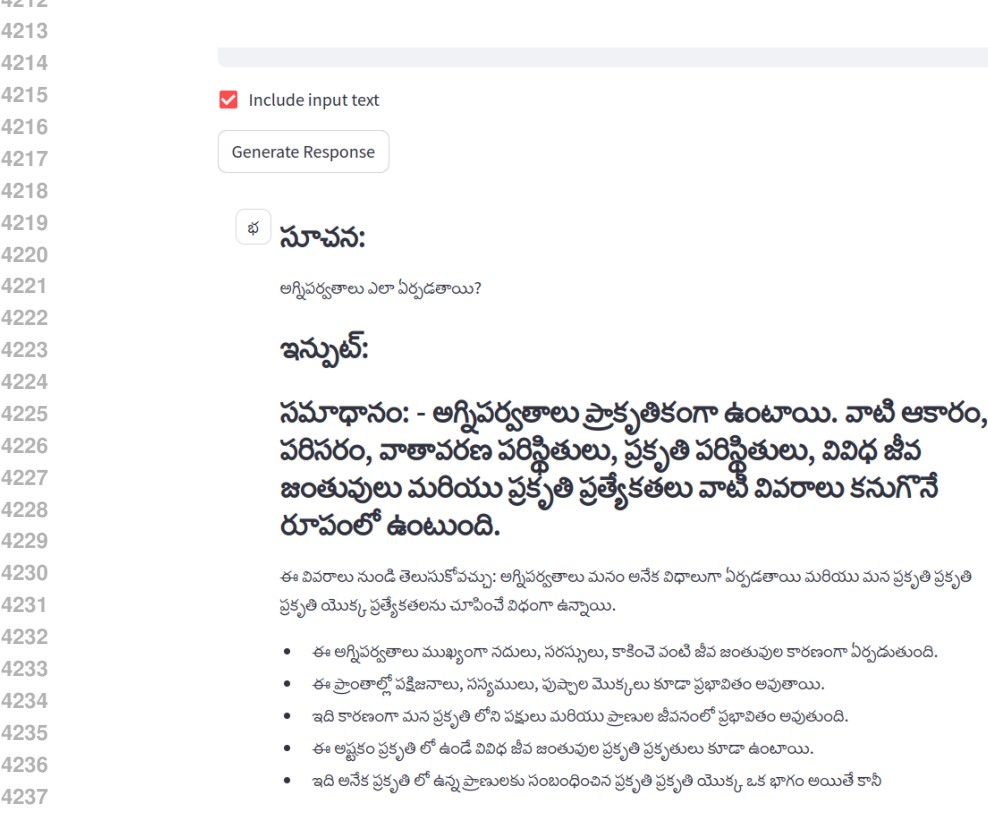

Figure 45: Prompting Paramanu-Telugu-instruct model with query: అగ్నిపర్వతాలు ఎలా ఏర్పడతా­యి? (agniparvatālu elā ērpaḍatāyi?)

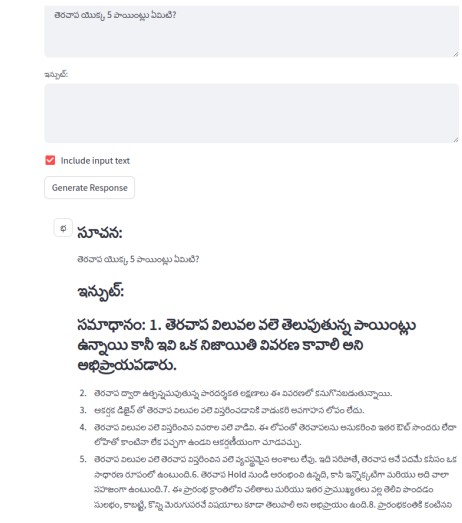

Figure 46: Prompting Paramanu-Telugu-instruct model with query: తెరచాప యొక్క 5 పాయింట్లు ఏమిటి? (teracāpa yokka 5 pāyiṁṭlu ēmiṭi?)

# Telegu-GPT App

సూచన:

అగ్నిపర్వతాలు ఎలా ఏర్పడతాయి?

ఇన్పుట్:

☑ Include input text

Generate Response

భ సూచన:

అగ్నిపర్వతాలు ఎలా ఏర్పడతాయి?

## ఇన్పుట్:

## సమాధానం: ఊహించగలంటే, అగ్నిపర్వతాలు ఏర్పడవునుగా ఈ ప్రకారంలో 'అగ్నిపర్వత' మరియు 'ఆగ్నిపర్వత' అనే రెండు ప్రకారాలు ఏర్పడకపోతే, అవి అలంకరించబడవు.

అల్లాలు మరియు కాలువలు ఎక్కువ ఉష్ణత వర్ధించకుండా ఏర్పడుతుంది. అందువలన అల్లాలు మరియు కాలువలు తీవ్రమైన తీవ్రతను చూపిస్తాయి. అల్లాలు అతి శీతాకారంగా ఉష్ణంగా ఉంటాయి, కానీ కాలువలు శక్తిపంతంగా ఉన్నాయి. ఇవి విరిరితమైన శక్తిని ఉపయోగించి, అగ్నిని ప్రతిఫలించే అనేక రీత్యాలు ఉన్నాయి. సాధారణంగా, అగ్నిపర్వతాలు వివిధ రీతిలో ఏర్పడతాయి, ప్రతిదాని ప్రకారం వివిధ వేగాలు ఉండే విధలు ఉంటాయి. అల్లాలు అజ్ఞాసముతో అనుభవించుకోవడం కష్టమేనా, ఇది సాధ్యం అయ్యే విధలు కూడా ఉంటాయి. అన్ని ప్రకారాల వలన ఒకదాని మీద ఒకటి

Figure 47: Prompting Paramanu-Telugu-instruct model with query: అగ్నిపర్వతాలు ఎలా ఏర్పడతా-యి? (agniparvatālu elā ērpaḍatāyi?)

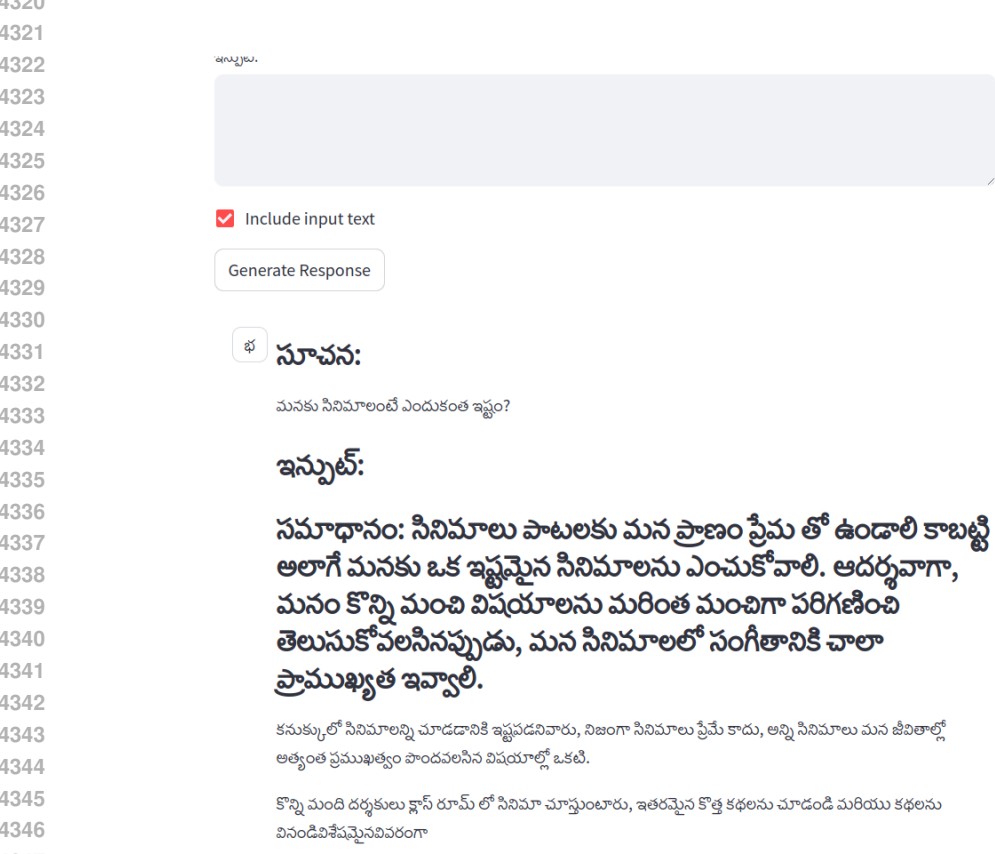

Figure 48: Prompting Paramanu-Telugu-instruct model with query: మనకు సినిమాలంటే ఎందుకంత ఇష్టం? (manaku sinimālaṁṭē eṁdukaṁta iṣṭaṁ?)

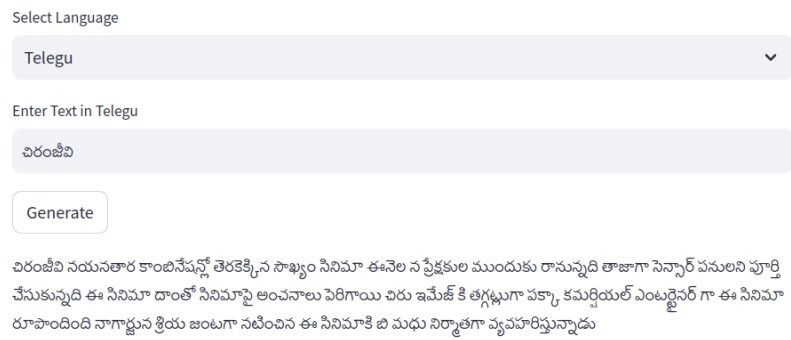

Figure 49: Prompting Paramanu-Telugu-instruct model with query: చిరంజీవి (ciraṁjīvi)

You may include other additional sections here.

