# OpenReview forum: "Paramanu: A Family of Novel Efficient Generative Foundation Language Models for Indian Languages"
_ICLR.cc/2025/Conference — Submitted to ICLR 2025_

### Official Review · Reviewer_cKJh · 2024-11-01

**Soundness:** 2
**Presentation:** 1
**Contribution:** 2
**Rating:** 3
**Confidence:** 5

**Summary:**

In this paper, the authors present a collection of small-scale auto-regressive, monolingual, bilingual and multilingual pretrained decoder-only LMs for 10 Indic languages covering 5 distinct scripts, supporting a context window of 1024. The authors also present a tokenization scheme combining both unigram tokenization and byte-pair encoding. Additionally, they propose an engineering method to scale down position IDs to allow longer context pre-training than maximum permissible context length on the physical memory.

**Strengths:**

- The extensive experiments showcase the capabilities of language-specific Paramanu-family LMs on multiple benchmarks (Belebele, XCOPA, MMLU, ARC, XNLI, HellaSwag, etc.) against Large-scale multilingual LMs which are $\geq$ 20 times the parameter count of Paramanu models.

**Weaknesses:**

- The paper is difficult to follow. At many points in the paper, the authors provide little to no background on the proposed approaches and evaluation metrics. For example, no background is provided for the Fertility metric, which is used to show the effectiveness of the mBharat tokenizer (Figure 2). Similarly, the perplexity metric is vaguely mentioned only in Appendix C.0.1 and is not referenced in the main text. This might make the paper difficult to read for people who are less familiar with this area of work.

- Instruction tuning details are underspecified/inconsistent. Abstract, Section 1 and Section 3.2 mention that 23K instances are used for instruction tuning Paramanu Models. However, Section 3.9 mentions 27K + 52K instances being used for Paramanu-Hindi 356M and 27K instances being used for instruction tuning Paramanu-Bangla 108M.

- In section 3.6, The heuristic used to set overall vocab size to 1750, 1K vocab size for konkani and 750 vocab size for Maithili is not clear.  A vocab size of 1750 may be too small to draw conclusions from. The two tokenization comparisons should be studied on a range of vocab sizes to ensure generalizability. Moreover, it might be interesting to look at language-specific perplexity scores, to validate how well the model trained using the merged-tokenizer performs for both the languages on their own. Overall, the comparison lacks experimental rigor.

- The proposed tokenization approach simply combines two pre-existing works: byte-pair encoding and unigram tokenization via a set operation.

- Quantitative figures on language and domain-wise data distribution in the pre-training corpora are missed out from the main text of the paper. In my opinion, pre-training corpora is one of the most important aspects of LM pre-training and should be included in the main text. `[This point is not considered while assigning scores]`

**Questions:**

- Can the authors provide more information on the Human evaluation conducted in this study. Mainly, how many annotators were involved in the evaluation? Does this work quantitatively measure inter-annotator agreement?

- Can the authors clarify the exact number of instruction tuning instances used for each model. If different models used different numbers of instances, explicitly state and explain this in the paper.

---

> ### Author Response · Authors · 2024-11-20
> **Rebuttal**
>
> 1. Little background
>
> R: We acknowledge that we consider the readers to be aware of the background information for fertility score but we have mentioned background information for language modelling (C0.1), perplexity metric (C0.1), RoPE (C0.2), RMSNorm (C0.3), Model FLOPs Utilisation (C.1), Maximal Update Parametrization (C.2), and also on carbon footprint calculation (C.3) in the appendix. In our revised paper, we will provide background information regarding fertility score.
>
> 2. Instruction tuning details
>
> R: Thank you for pointing out the inconsistencies. We used 23,000 instances of instructions dataset for instruction-tuning of pretrained Paramanu models for all languages except Hindi. For Hindi, we used 23k + 52k instruction sets (machine translation of Alpaca). We will update this in the revised paper.
>
> 3. Vocabulary sizes
>
> R: Through internal tests and optimization across a range of vocabulary size in conjunction with the pretraining corpora size, we found the optimal vocabulary size for Konkani to be 1K and for Maithii to be 750 for experimentation for the 15 million parameters model as mentioned in L298-304. We performed several internal tests and hyperparameter tuning to find the optimal vocabulary size for these languages based on the pretraining corpora (L251-254). In L267-269, we mentioned using the merged tokenizer to train another 27M bilingual Maithili-Konkani GPT and found the validation perplexity of the bilingual modeling to drop to 8.54 from 12.43. This validates our proposed approach as how a merged language specific tokenizer improves the performance of bilingual language modeling. Similarly, we extended our proposed approach to multilingual language modeling. Our proposed multilingual tokenizer, mBharat, has the least fertility score among the BPE tokenizers of Llama-3, Sarvam 2B, Gemma. This shows the superiority of our novel tokenization technique.
>
> 4. Tokenization
>
> R: We are the first to propose the combination of two existing tokenizers into one and on top, we proposed and performed domain specific tokenization for monolingual models and language specific tokenization for multilingual models (L93-94). Doing that, we achieved the lowest fertility score among tokenizers of Llama-3, GPT-4o, Gemma, Sarvam 2B LLMs for Indian languages across 5 scripts.
>
> 5. Data distribution
>
> R: We apologise that we did not provide quantitative figures of domain wise data distribution, as we focused primarily on other contributions. We will include this information.
>
> 6. Prompts for human evaluation
>
> R: The chosen prompts reflect the local, cultural, and literary contexts of Assamese, Bangla, Hindi, Odia, and Sanskrit (L462-463). Lines 1226-1228 reflect the Bengali prompts and their significance. All these prompts are related either to common expressions in Bangla or covers diverse Bangla literature and cultural context in the Bangla speaking community in West Bengal, India. For Hindi prompts, these prompts are related to popular celebrities across cricket, films, politics and music respectively in India (L1421-1422). For human evaluation, we asked 10 annotators to evaluate top-3 responses from models for each prompt on a scale of 0 (worst) to 5 (best). We report the average score of all ratings. We also have reported normalised scores of ratings in Table 19 in appendix (L1371) to handle inconsistencies among annotators. In our revised paper, we will specify the number of human annotators.
>
> 7. Instruction tuning instances
>
> R: Thank you for pointing out the inconsistencies. We used 23,000 instances of instructions dataset for instruction-tuning of pretrained Paramanu models for all languages except Hindi. For Hindi, we used an additional 52,000 instructions, making a total of 75,000 dataset. We will update the inconsistencies in the revised paper.

---

> > ### Comment · Reviewer_cKJh · 2024-11-23
> > **Rebuttal Response**
> >
> > Authors: Through internal tests and optimization across a range of vocabulary size in conjunction with the pretraining corpora size, we found the optimal vocabulary size for Konkani to be 1K and for Maithii to be 750 for experimentation for the 15 million parameters model as mentioned in L298-304. We performed several internal tests and hyperparameter tuning to find the optimal vocabulary size for these languages based on the pretraining corpora (L251-254). In L267-269, we mentioned using the merged tokenizer to train another 27M bilingual Maithili-Konkani GPT and found the validation perplexity of the bilingual modeling to drop to 8.54 from 12.43. This validates our proposed approach as how a merged language specific tokenizer improves the performance of bilingual language modeling. Similarly, we extended our proposed approach to multilingual language modeling. Our proposed multilingual tokenizer, mBharat, has the least fertility score among the BPE tokenizers of Llama-3, Sarvam 2B, Gemma. This shows the superiority of our novel tokenization technique.
> >
> > **What criteria is used to identify that a particular vocab size is optimal? Please tabulate and present atleast a subset of the experiments performed to select the given vocab size for Konkani and Maithili. What insights do you draw from this exercise of finding language-specific optimal vocab size against the pre-training corpora size?**
> >
> > ---
> >
> > Authors: We are the first to propose the combination of two existing tokenizers into one and on top, we proposed and performed domain specific tokenization for monolingual models and language specific tokenization for multilingual models (L93-94). Doing that, we achieved the lowest fertility score among tokenizers of Llama-3, GPT-4o, Gemma, Sarvam 2B LLMs for Indian languages across 5 scripts.
> >
> > **Given vocabulary $V$, during inference (post-training), how do you tokenize any input to the LM. Can you give an example and explain the step-by-step tokenization of the input.**
> >
> > ---
> >
> > Authors: The chosen prompts reflect the local, cultural, and literary contexts of Assamese, Bangla, Hindi, Odia, and Sanskrit (L462-463). Lines 1226-1228 reflect the Bengali prompts and their significance. All these prompts are related either to common expressions in Bangla or covers diverse Bangla literature and cultural context in the Bangla speaking community in West Bengal, India. For Hindi prompts, these prompts are related to popular celebrities across cricket, films, politics and music respectively in India (L1421-1422). For human evaluation, we asked 10 annotators to evaluate top-3 responses from models for each prompt on a scale of 0 (worst) to 5 (best). We report the average score of all ratings. We also have reported normalised scores of ratings in Table 19 in appendix (L1371) to handle inconsistencies among annotators. In our revised paper, we will specify the number of human annotators.
> >
> > **Quantitative figures on inter-annotator agreement is still required even with normalized and averaged scores.**

---

### Official Review · Reviewer_gqHC · 2024-11-04

**Soundness:** 1
**Presentation:** 2
**Contribution:** 2
**Rating:** 1
**Confidence:** 4

**Summary:**

The authors have developed a set of language models focusing on 10 Indian languages. They have studied various tokenization approaches for these languages. Additionally, they have created instruction datasets for these languages. The authors claim superior performance compared to larger models that are not focused on these languages. They have also performed human evaluation focusing on grammar, coherence, creativity, and factuality.

**Strengths:**

1. The authors curate a large pre-training dataset for Indian languages which are usually underrepresented in the available pre-training corpus.
2. The authors curate a human written instruction dataset for Bengali language.

**Weaknesses:**

1. The instruction tuning dataset is created in one language (Bengali) and then translated using Google-Translate to the other languages. This is a significant limitation as the quality of the instruction data is not on-par with human written instruction data for most of the languages.
2. The authors do not clearly explain how they merge Unigram and BPE tokenizers. It is not clear how the authors tokenizing a given text. Are they using BPE decoding algorithm or the Unigram decoding algorithm?
3. The choice to exclude English from pre-training is puzzling as there is a lot of available training data available for English. This is a significant limitation as the trained model cannot deal with source code, scientific journals/articles, medical and other technical domain data. The authors have not explained the motivation behind this decision.
4. Human evaluation is performed on just 4 prompts. This is not enough to make any reliable conclusions on the quality of the models. The authors have not reported inter-annotator consistency for the human evaluations or even how many independent human evaluations were taken per sample. Thus the authors claim that their models are better than existing LLMs is not supported.

**Questions:**

1. What criteria has been used to chose prompts for human evaluation? How many human annotators were there? How was the inconsistency among annotators resolved?
2. How do you combine a BPE and a Unigram based tokenizer?

---

> ### Author Response · Authors · 2024-11-20
> **Rebuttal**
>
> 1. Translation
>
> R: While it is true that the instruction-tuning data has been translated from only Bengali, to assure quality, we also performed a manual check on the translations generated for Hindi and Marathi.  We found out around 8% translation errors on average for Hindi and Marathi. Unfortunately, we could not do it for the rest of the languages due to resource limitations at our end.
>
> 2. Tokenizer
>
> R: In L194-L200, we explained the process of our tokenization process. We trained individual BPE tokenizer and Unigram tokenizer of the same vocabulary size and then we merged the two tokenizers via merge by set operation to form the final tokenizer. Figure 1 in the paper shows our novel tokenization method. This results in removing overlapping tokens of languages belonging to the same script and typology, resulting in an optimised tokenizer. Based on the empirical results and from Figure 2, we also found that our novel tokenization technique resulted in developing the most powerful tokenizer for Indian languages with the least fertility score among tokenizers of GPT-4o, Llama-3, Gemma, Sarvam 2B.
>
> 3. Exclusion of English
>
> R: Our main objective is to show that powerful generative language models for low resource Indian languages can be made without including English corpus. We are the first to propose and execute end-to-end empirical experiments and evaluation. To the best of our knowledge, the existing LLMs and multilingual LLMs struggle to generate grammatically correct and coherent sentences in Indian languages (and reasoning ability is too far) as shown in Appendix tables 15, 21-22, 24-35. Moreover, we excluded English as it is not linguistically similar to any Indian languages in neither typology/script nor grammatical and morphology nuances. Also, the existing multilingual models have heavy English centric bias due to large data imbalance towards English. Consequently, the Bloom series and other LLMs do not generate grammatically coherent sentences in Indian languages (Appendix tables 15, 21-22, 24-35). Many LLMs even generate text in Arabic or Japanese when we prompted with prompts in Bengali, Hindi, and Sanskrit. We also found that LLMs including ChatGPT (Oct '23) were not able to distinguish languages of the same script such as Bengali versus Assamese (L1385-1389). Human evaluation by annotators confirm this (Section A.5 in Appendix L1184-1437). We are sorry that we did not mention these reasons explicitly in the paper. We will include them.
>
> 4. Human evaluation
>
> R: We humbly beg to differ about the conclusion. Our claims are based on empirical results of both automatic LLM metrics as well as human evaluation on the generation ability of our pretrained models in various Indian languages. We performed extensive zero-shot quantitative evaluation across several LLM benchmarks (MMLU, HellaSwag, ARC, XCOPA, XNLI, Belebele) for both pretrained models and instruction-tuned models, and then human evaluation of our pretrained models covering cultural and linguistics nuances in Bengali, Hindi, and Sanskrit using 4 prompts each. The human evaluation was performed by 10 independent native speakers annotators who evaluated the models’ generation capabilities with respect to prompts on grammar, coherence, creativity, and factuality metrics. L368-374 and L409-426 clearly show the superior performance of our models with respect to LLMs. Tables 4-8 list the performance comparison of our models with LLMs which are bigger in size by multiple orders of magnitude on various automatic LLM benchmarks.
>
> 5. Prompts for human evaluation
>
> R: The chosen prompts reflect the local, cultural, and literary contexts of Assamese, Bangla, Hindi, Odia, and Sanskrit (L462-463). Lines 1226-1228 reflect the Bengali prompts and their significance. All these prompts are related either to common expressions in Bangla or covers diverse Bangla literature and cultural context in the Bangla speaking community in West Bengal, India. For Hindi prompts, these prompts are related to popular celebrities across cricket, films, politics and music respectively in India (L1421-1422). For human evaluation, we asked 10 annotators to evaluate top-3 responses from models for each prompt on a scale of 0 (worst) to 5 (best). We report the average score of all ratings. We also have reported normalised scores of ratings in Table 19 in appendix (L1371) to handle inconsistencies among annotators. In our revised paper, we will specify the number of human annotators.

---

> > ### Comment · Reviewer_gqHC · 2024-11-22
> > **About the tokenizer**
> >
> > Can you explain how you merge a BPE and a Sentencepiece tokenizer? What do you mean by "merge by set operation"? When you are tokenizing a sentence do you use the BPE algorithm or the Sentencepiece algorithm to tokenize the text?

---

> > ### Comment · Reviewer_gqHC · 2024-11-22
> > **About Human Evaluation and Zero-shot Evaluation**
> >
> > 1. Please report standard Inter-annotator agreement metrics such as Kohen's Kappa, Fleiss's Kappa.
> > 2. My point about just using 4 prompts still stands. With such a small sample, I do not think it is possible to get statistically significant results.
> > 3. For table 4-8 you should show in bold which model performs best. I have gone through these tables again. The models proposed by the authors do not have the best performance. If the argument is that the models are performing well given their parameter count, the comparison should be with models with similar parameter count.
> > 4. I understand that human evaluation is very expensive. However there are still some automatic evaluation approaches targeting specific aspects of quality such as Self-BLEU (tests for diversity).

---

> ### Author Response · Authors · 2024-11-22
> **Rebuttal: About the tokenizer**
>
> We train individual BPE and Unigram tokenizers using SentencePiece module and then we merged both the tokenizers by intersection to remove overlapping tokens from two tokenizers to form the final merged tokenizer. We used this merged tokenizer for our pretraining, and instruction fine-tuning experiments and also during inference.
>
> Please acknowledge our multiple novel contributions:
>
> 1. RoPE embedding scaling method for long context modeling capabilities without increased GPU memory.
> 2. Novel tokenization using a combination of BPE and Unigram
> 3. Language specific tokenization for multilingual language modeling and domain specific tokenization for monolingual modeling.
> 4. Pretraining data contribution for 10 Indian languages
> 5. Instructions tuning dataset for Bengali and its translation to 4 different Indian languages.
> 6. Typological grouping and pretraining on comparable corpora to handle bias and data imbalance.
> 7. We show for the first time, that we can develop strong tiny/small generative language models from scratch with limited compute (single GPU) with long context modeling capabilities and limited tokens budget (w/o including English tokens). Our models are very sustainable, on-prem deployable in smartphones and highly cost efficient in both training and inference as shown in Table 1 we compare the training hours of our models with models of same size and much bigger LLMs.

---

> > ### Comment · Reviewer_gqHC · 2024-11-22
> > **Response**
> >
> > 1. Please cite related work for your embedding scaling method and how it is novel from existing literature.
> > 2. I do not understand your tokenization approach. I have repeatedly asked you to provide the algorithm for how you merge BPE and Unigram tokenizers. These two tokenizers use very different algorithms to tokenize a string. I am not understanding how you combine the two.
> > 3. Language specific tokenization is not a novel contribution.
> > 4. I have acknowledge this in my original review.
> > 5. I have acknowledged your Bengali instruction dataset contribution. However, machine translating the Bengali dataset to 4 other languages is not a novel contribution.
> > 6. Please explain this more. This point is related to how you merge tokenizers.
> > 7. Where do you test your model's long context capabilities?

---

> > > ### Comment · Reviewer_cKJh · 2024-11-23
> > > **Reiterating Reviewer gqHC's 2nd point**
> > >
> > > Dear Authors,
> > > You mention that during training, you train both BPE and Unigram, and then merge the respective vocabularies $V_1$ and $V_2$ without conflict to obtain $V$. Given vocabulary $V$, **during inference (post-training)**, how do you tokenize any input to the LM. Can you give an example and explain the step-by-step tokenization of the input.

---

> > > > ### Author Response · Authors · 2024-11-24
> > > > **Response to Rebuttal: Tokenizer**
> > > >
> > > > Thank you for reading our rebuttal.
> > > >
> > > > 1. Tokenization
> > > >
> > > > Consider two independent tokenizers (BPE and Unigram tokenizers) of size V as two arrays with respective tokens as their elements. Now, use array merge operation to form a new array (tokenizer) of size V'. This new tokenizer with V' is used to tokenize the pretraining data and used during both pretraining, fine-tuning, and inference. No other tokenizers are involved during end-to-end pretraining, instruction fine-tuning, and during inference. The tokenizer with V' is the tokenizer which is used to tokenize the dataset, execute end-to-end training and inference.
> > > >
> > > > Step 1: Train independent BPE and Unigram tokenizers of same size.
> > > >
> > > > Step 2: Merge the independent tokenizers as array merging results in size V'.
> > > >
> > > > Step 3: Use the merged tokenizer of size V' to tokenize the dataset, RoPE embedding, and use it for end-to-end pretraining, fine-tuning, and inference. No other tokenizers are involved except the merged tokenizer.

---

> > > ### Author Response · Authors · 2024-11-24
> > > **Response to Rebuttal**
> > >
> > > Thank you for reading our rebuttal.
> > >
> > > 1. Tokenization
> > >
> > > Consider two independent tokenizers (BPE and Unigram tokenizers) of size V as two arrays with respective tokens as their elements. Now, use array merge operation to form a new array (tokenizer) of size V'. This new tokenizer with V' is used to tokenize the pretraining data and used during both pretraining, fine-tuning, and inference. No other tokenizers are involved during end-to-end pretraining, instruction fine-tuning, and during inference.
> > >
> > > 2. Language specific and domain specific tokenization
> > >
> > > To the best of our knowledge, we are the first to train generative language models from scratch using the concept of language specific tokenization for multilingual model where we train individual language tokenizers and then we merge all the independent language tokenizers to form the final mulitlingual tokenizer. We also compared with language agnostic approach by training a single tokenizer on multilingual dataset (L269-293) to show the advantage of our language specific tokenization for multilingual language modeling from scratch. We are also the first to train individual domain tokenizers for monolingual modeling and then performing merge operation to form a final tokenizer for monolingual models and use it to tokenize monolingual dataset, followed by pretraining, fine-tuning, and inference.
> > >
> > > 3. Long context capabilities
> > >
> > > We have tested our RoPE embedding scaling method by training generative language models from scratch at context size of 4096 and 8192 on single A100 40G GPU in our another conference submission. Our method does not require additional GPU memory for long context modeling, enabling pretraining for long context modeling on single GPU. The validation perplexity of the model trained with 4096 context size is 1.6561  whereas the perplexity of 8192 model is 1.4582. Since, in this paper, we have multiple novel contributions, to our humble opinion, long context modeling with our embedding scaling method is out of the scope of this paper.
> > >
> > > We urge you to kindly acknowledge our scientific contributions, dataset contribution, model contributions, and findings to improve your ratings. We have also open sourced some of our models to the research community.

---

> > > > ### Comment · Reviewer_gqHC · 2024-11-25
> > > > **Long-context capabilities**
> > > >
> > > > If you studied long context capabilities of your proposed approach in another paper (conference submission), you cannot claim that as contribution on this paper.

---

> > > > ### Comment · Reviewer_gqHC · 2024-11-25
> > > > **The two main issues with this paper**
> > > >
> > > > There are two aspects of this paper that need to be addressed before I will consider increasing the score.
> > > >
> > > > First, the tokenization approach. It is unclear what the authors are actually doing when they say "merging two tokenizers". How can a BPE and a Unigram tokenizer be merged? The two follow different algorithms for tokenization. The rebuttal responses do not address this issue.
> > > >
> > > > Second, evaluation of the models on just four prompts. This is unacceptable. With a sample size four no significant conclusion can be made either way.
> > > >
> > > > These two aspects are most important as they go to the heart of "soundness" of this paper.

---

> > > > > ### Author Response · Authors · 2024-11-25
> > > > > **Response to "The two main issues with this paper"**
> > > > >
> > > > > 1. Tokenization
> > > > >
> > > > > It is quite trivial, you are getting over-complicated as in the process as how two different BPE and Unigram tokenizers of size V are constructed following two different algorithms but in the end these are just data structures, does not matter how the tokenizers are constructed from scratch. Ultimately, they are only data structures with tokens as their elements. Now you can merge the data structures by intersection (tokenizers) using SentencePiece module to form the final merged tokenizer of size V'.
> > > > >
> > > > > Use this merged tokenizer V' to tokenize the dataset, RoPE, training, and inference, and no other tokenizers are involved. This is how we pretrain all our models from scratch, performed instruction fine-tuning and using the same merged tokenizer for inference and evaluation.
> > > > >
> > > > > 2. Evaluation
> > > > >
> > > > > We would like to state that apart from human evaluation performed by 10 annotators across three language, you are completely overlooking our quantitative automatic LLM benchmarks (MMLU, HellaSwag, ARC, XCOPA, XStroryCloze, Belebele) and our tiny models (less than 370M parameters) outperformed all LLMs under comparison of size 2 billion parameters across Hindi, Bengali, Marathi, Tamil, Telugu.
> > > > >
> > > > > 3. Dataset contribution
> > > > >
> > > > > Pretraining dataset for 10 Indian languages
> > > > >
> > > > > 4. Instruction-tuning dataset for Bengali and then machine translation to 4 major languages with correction of translations errors.
> > > > >
> > > > > 5. New models of 3 types (monolingual, bilingual, multilingual) pretrained from scratch on single GPU 40GB at context size of 1024 with our proposed RoPE scaling method whereas Llama was pretrained from scratch on 2000 GPUs at context size of 2048.
> > > > >
> > > > > 6. The most powerful tokenizer ever developed for Indian languages, least fertility score among Llama-3, GPT-4o, Sarvam 2B, Llama-2, etc.
> > > > >
> > > > > 7. Topological grouping of languages of same script and pretraining on comparable corpora to handle data and language bias in multilingual models.
> > > > >
> > > > > 8. We show for the first time that we can make powerful generative language models pretrained from scratch of varying size and various types for low resource Indian languages using least compute (single GPU) , generating least carbon footprint (as shown in Table 1) and limited token budget without including English tokens which have been typically done before by other researchers. We show that yes without huge compute and trillion tokens or even hundreds of billion tokens, tiny generative models pretrained from scratch are equally competitive or even better than most LLMs of trillion tokens and billion parameters with hundred thousands of training time unlike ours approach.

---

> > > > > > ### Author Response · Authors · 2024-11-29
> > > > > > **Rebuttal Revision Update to Reviewer gqHC**
> > > > > >
> > > > > > Thank you for reading our rebuttal and your feedback. We have incorporated your feedback and revised our paper. Our changes are in blue and the removed lines are crossed out.
> > > > > >
> > > > > > We have the following revisions:
> > > > > > 1. We fixed the inconsistencies of number of instructions instances for Hindi. (L39, L121, L365)
> > > > > > 2. We added a pie chart to show the data distribution of our training data  (L160) (Fig. 3 in the Appendix)
> > > > > > 3. We added the number of human annotators and also provided a figure (Fig. 4 in the Appendix) to represent inter-annotator agreement Kappa metric for our human evaluation for Bengali, Hindi, and Sanskrit. (L451-455)
> > > > > > 4. We split our quantitative benchmark performance comparison into two tables, in one table, we compare our models with multilingual LLMs of size <=2B (Tables 4-8) and the other table (Tables 12-15 in the Appendix) , we compared our models with LLMs > 2 B (L316-317). We bold the max scores for each benchmark. (L386, L397, L439, L448, L493)
> > > > > > 5. We provided supporting statements as why we planned to exclude English from our models pretraining. (L75-86)
> > > > > > 6. We explained our tokenizer process more lucidly. (L194-196)
> > > > > > 7. We reported the percentage of machine translation errors and how we addressed the machine translation errors while translating Bengali instructions to Hindi and Marathi. (L183-185)
> > > > > > 8. We described as how we found optimal vocabulary size for our models. (L270-271)
> > > > > > 9. We also cited some related work which we mentioned before but previously forgot to cite (L135)
> > > > > > 10. We added number of speakers, language family, and script for Indian languages in Table 1 (L172)

---

### Official Review · Reviewer_XvpY · 2024-11-04

**Soundness:** 2
**Presentation:** 2
**Contribution:** 2
**Rating:** 3
**Confidence:** 3

**Summary:**

This paper presents monolingual, bilingual, and multilingual Indian language models trained from scratch for 10 Indian languages across 5 scripts. The models are small (ranging 13M to 368M parameters, all trained on a single A100 GPU), but shown to outperform recent open-source LLMs for the target languages. In addition to automatic metrics human evaluation is performed.

To do so, they present both new data and modelling experiments:
- an instruction-tuning dataset with 23k instructions, automatically translated with Google Translate
- automatically translate existing English benchmarks such as HellaSwag to evaluate Indic languages
- an adapted RoPE positional embedding
- compare unigram and BPE tokenization (at a specific vocabulary size), and combine language-specific and domain-specific tokenizers to create a tokenizer with improved fertility

**Strengths:**

Trains a variety of small GPT models for 10 Indic languages which outperform significantly larger public models for the target languages.
Model artifacts and datasets may be of use to future researchers.

**Weaknesses:**

Many choices are not experimentally validated.

The main text directly lists results from tables, instead of providing additional insights or analyses.

References very little past work on Indic languages. See for example the work from AI4Bharat on creating models and corpora for the languages described here such as [IndicLLMSuite](https://arxiv.org/abs/2403.06350), which presents instruction-tuning datasets and public models. Tokenization and vocabulary choice for Indic languages has also been the subject of a fair amount of prior work in for example the recent WAT evaluations which is not referenced. It has typically been found significantly beneficial to transliterate or romanize Indic scripts to create a shared vocabulary - see [RomanSetu](https://arxiv.org/abs/2401.14280)'s related work

**Questions:**

How were the vocabulary sizes (750 and 1k) per language and domain chosen? Such small vocabulary sizes will not allow sufficient merges to result in specialized vocabulary - were these numbers chosen experimentally, or how were they chosen?

Creating multilingual datasets via automatic translation can introduce errors; was there any evaluation or spot checking of the data quality?

---

> ### Author Response · Authors · 2024-11-20
> **Rebuttal**
>
> 1. Experimental choices
>
> R: We have performed many internal tests and hyperparameter tuning to find optimal tokenizer size, and other hyperparameters (L251-254).
>
> 2. Additional insights
>
> R: We are sorry if we have missed out on providing additional insights due to the page limit.
>
> In L359-365 (Table 5 and Table 6), we discuss the cross-lingual language transfer capabilities of our model among the languages of the same script and typology (Marathi and Hindi belonging to Devanagari script). Table 5 and Table 6 assess various pretrained monolingual and multilingual models for Devanagari script across several benchmarks, highlighting cross-lingual language transfer among Devanagari languages (Hindi, Marathi). Interestingly, neither Paramanu-Sanskrit nor mParamanu were pretrained on Hindi or Marathi but still performed well on their benchmarks. This is possibly due to the same script. The stronger performance of mParamanu in Hindi (mParamanu was not pretrained on Hindi but on other languages that use the Devanagari script) illustrates effective language transfer within the same script and typology. This finding confirms the generalisation ability across Indian languages with unique typological features.
>
> In addition, in L405-416, we mention that from Table 3, we observe that the performance of our models drop from zero-shot setting to 25-shot setting on XNLI-Hindi, XStoryCloze for Hindi and Telugu, and XCOPA for Tamil. This type of phenomenon has also been observed in PlanningBench (Valmeekam et al., 2024) where performances of GPT-3.5-Turbo, GPT-4, and GPT4-o on Blocksworld dropped from zero-shot to one-shot significantly. Perhaps, n-shot examples become additional soft constraints on the generation which might be the reason for degradation of performance from the original training dataset.
>
> In L417-431, we report the analyses from Table 7 and Table 8 on comparing pretrained multilingual LLMs and instruction-tuned models on Tamil and Telugu benchmarks. The improvements in metric scores for our Tamil and Telugu instruction-tuned models were modest, likely due to lower-quality machine translations from Bangla compared to Hindi. Nonetheless, these results show strong performance of our models on various NLP tasks despite their smaller size and fewer tokens, challenging the notion that larger models are always better. Our findings suggest that smaller pretrained models can excel when trained on high-quality, preprocessed data over multiple epochs, outperforming larger models trained on lower-quality data for a single epoch.
>
> 3. Related work and English
>
> R: We are sorry if we have missed out on important past work. Some of them appeared after we submitted our work. We will review and include them.
>
> We have excluded English datasets and only focused on keeping language purity and typology grouping for our models and not clubbing with languages of Roman script with Indian languages. Thus, we did not find it relevant to romanize Indian languages to Roman script. The main purpose of our paper was to show that strong generative language models are possible to train from scratch for low resource languages with limited time budget, and without including English tokens. The AI4Bharat models have performed the common approach of continual pretraining of English Llama to Hindi by vocabulary expansion. Their approach is not sustainable and cost-efficient as they spent many more hours just to do continual training of Llama compared to us pretraining from scratch. Moreover, these models still suffer from heavy English centric bias as 7B parameters were trained on trillion English tokens and only few LoRA/QLoRA parameters were trained for Indian languages for Llama adaptation for non-English languages.
>
> 4. Vocabulary sizes
>
> R: We have not set vocabulary size of 750 and 1K for every language, it is only for the Konkani and Maithili languages for experimentation for the 15 million parameters model as mentioned in the Section 3.7 (L298-304). We performed several internal tests and hyperparameter tuning to find the optimal vocabulary size for these languages based on the pretraining corpora (L251-254).
>
> 5. Automatic translation
>
> R: We agree that creating multilingual datasets using automatic machine translation from Bengali to other Indian languages introduce translation errors but we also performed human checks and corrected them. We found around 8% word errors on average for Hindi and Marathi using Google Translate. In our revised paper, we will add the information. Due to resource constraints, we could not do the rest in the current draft.

---

### Official Review · Reviewer_Jmhb · 2024-11-05

**Soundness:** 3
**Presentation:** 2
**Contribution:** 2
**Rating:** 5
**Confidence:** 4

**Summary:**

The paper introduces Paramanu models, designed to improve NLP capabilities for ten Indian languages across five scripts. The authors developed monolingual, bilingual, and multilingual models using novel tokenization techniques, optimized for low-resource settings, and a scalable RoPE embedding method that enables efficient pretraining on a single GPU. Evaluations across benchmarks showed that Paramanu models outperform many larger language models, especially on tasks involving grammar, coherence, creativity, and factuality in Indian languages.

**Strengths:**

- Paramanu models effectively address the low-resource problem for Indian languages with high accuracy across language-specific NLP tasks.
- The paper introduces an efficient RoPE embedding scaling technique that enables larger context sizes without requiring increased GPU memory.
- The novel tokenization approach combining BPE and Unigram tokenizers improves performance, especially for Indian languages with complex morphology.

**Weaknesses:**

- The model's performance was primarily evaluated on a limited set of benchmarks, potentially limiting insights into other diverse language tasks.
- Some models may be undertrained, as indicated by perplexity scores, suggesting room for improvement with extended training.
- The approach may require further testing to generalize across Indian languages with unique typological features, beyond the ten languages used.

**Questions:**

-

---

> ### Author Response · Authors · 2024-11-20
> **Rebuttal**
>
> 1. Limited benchmarks
>
> R: We humbly beg to differ. We have evaluated our models across various automatic LLM benchmarks of multiple choice questions answering across common sense reasoning, logical reasoning, machine reading comprehension, natural language inference (ARC, MMLU, HellaSwag, Belebele, XStoryCloze, XCOPA, XNLI) in major Indian languages across 5 scripts. We have also done human evaluation on the generation capabilities of our models for low and mid-resource Indian languages on grammar, coherence, creativity, factuality metrics. Similarly, for a fairer comparison we have compared mostly with much larger multilingual 7B/2B LLMs that have been pretrained on Indian languages such as Bloom series, Sarvam 2B, OpenHathi, etc.
>
> 2. Perplexity scores
>
> R: As we mentioned in L98, that we performed pretraining of more than 1 epoch for most of our of models but we also acknowledge in our paper (L375) that some models have scope of improvement in downstream performance and also in the perplexity scores such as our Hindi 367M model. It may happen that with more training steps, the downstream performance of our models become better; however, we do not see it as a weakness but rather a strength, since the results can only improve from here.
>
> 3. Generalize across Indian languages
>
> R: Due to resource constraints, we could not test beyond the current set of languages and scripts. However, our experiments on cross-lingual transfer (for example, on Sanskrit, Konkani, Maithili using mParamanu model on Hindi) illustrates effective knowledge transfer among languages that use the same script and typology. This finding confirms the generalisation ability across Indian languages.

---

> > ### Author Response · Authors · 2024-11-29
> > **Rebuttal Revision Update to Reviewer Jmhb**
> >
> > Thank you for reading our rebuttal and your feedback. We have incorporated your feedback and revised our paper. Our changes are in blue and the removed lines are crossed out.
> >
> > We have the following revisions:
> > 1. We fixed the inconsistencies of number of instructions instances for Hindi. (L39, L121, L365)
> > 2. We added a pie chart to show the data distribution of our training data  (L160) (Fig. 3 in the Appendix)
> > 3. We added the number of human annotators and also provided a figure (Fig. 4 in the Appendix) to represent inter-annotator agreement Kappa metric for our human evaluation for Bengali, Hindi, and Sanskrit. (L451-455)
> > 4. We split our quantitative benchmark performance comparison into two tables, in one table, we compare our models with multilingual LLMs of size <=2B (Tables 4-8) and the other table (Tables 12-15 in the Appendix) , we compared our models with LLMs > 2 B (L316-317). We bold the max scores for each benchmark. (L386, L397, L439, L448, L493)
> > 5. We provided supporting statements as why we planned to exclude English from our models pretraining. (L75-86)
> > 6. We explained our tokenizer process more lucidly. (L194-196)
> > 7. We reported the percentage of machine translation errors and how we addressed the machine translation errors while translating Bengali instructions to Hindi and Marathi. (L183-185)
> > 8. We described as how we found optimal vocabulary size for our models. (L270-271)
> > 9. We also cited some related work which we mentioned before but previously forgot to cite (L135)
> > 10. We added number of speakers, language family, and script for Indian languages in Table 1 (L172)

---

### Meta-Review · Area_Chair_9tLn · 2024-12-19

**Metareview:**

The paper introduces Paramanu, a family of small-scale generative language models specifically designed for Indian languages across five scripts. The work includes monolingual, bilingual, and multilingual models, a RoPE embedding scaling method for larger context sizes, and a tokenization approach combining BPE and Unigram methods. The paper emphasizes efficiency in low-resource settings, claiming improved performance over larger existing LLMs on tasks involving grammar, coherence, creativity, and factuality.

However, the submission has several notable weaknesses. Despite claims of novelty, the tokenization method remains unclear, as repeated clarification attempts on the merging of BPE and Unigram tokenizers were unconvincing. This fundamentally undermines the paper's technical soundness. Furthermore, while the authors highlight human evaluation as a strength, it was conducted on only four prompts per language with limited methodological rigor. Therefore claims of model superiority are not sufficiently supported. Additionally, critical experimental choices, such as vocabulary size selection and the exclusion of English, are insufficiently justified and not evaluated for broader generalizability. The relation to prior work is also incomplete, with notable omissions of relevant efforts like IndicLLMSuite. Given these weaknesses, the paper does not meet the standard for ICLR and is therefore recommended for rejection.

**Additional Comments On Reviewer Discussion:**

During the rebuttal period, the discussion focused on three key issues: (1) the clarity and validity of the proposed tokenization method, (2) the insufficient human evaluation, and (3) the lack of justification and rigor in certain experimental choices.

Tokenization Method: Reviewers, particularly gqHC and cKJh, repeatedly questioned the authors' description of the tokenization approach, which claimed to merge BPE and Unigram tokenizers. Multiple attempts by the authors to clarify only addressed how the vocabularies are merged, not how the combined tokenization algorithm works. This lack of a clear algorithmic explanation undermined the technical soundness of this core contribution.

Human Evaluation: Reviewer gqHC criticized the limited human evaluation, which was conducted on only four prompts per language. Although the authors reported using ten annotators and provided inter-annotator agreement scores, reviewers argued that the small sample size was insufficient for drawing meaningful conclusions. The authors defended this with qualitative insights and pointed to additional quantitative evaluations on LLM benchmarks, but concerns about statistical significance remained unresolved.

Experimental Choices and Justification: Reviewers raised concerns about the small vocabulary sizes, inconsistent reporting of instruction-tuning datasets, and lack of justification for excluding English during training. The authors provided responses citing internal tests and optimizations for vocabulary size, clarified inconsistencies in instruction dataset counts, and reiterated their focus on Indian languages without English. However, reviewers argued that these justifications lacked sufficient experimental validation and broader generalizability. Additionally, the paper did not sufficiently position itself against related works, such as AI4Bharat’s IndicLLMSuite.

In my final decision, I weighed the unresolved issues. The unclear and unsubstantiated tokenization method raises significant concerns about the paper's technical rigor and reproducibility. The limited scope and methodology of the human evaluation significantly weaken claims of model superiority, despite additional quantitative results. Lastly, while the authors attempted to address experimental concerns, their justifications lacked the rigor expected for ICLR, and the paper remains poorly contextualized within existing literature. These issues persisted throughout the discussion, with reviewers maintaining their reservations. Given these fundamental weaknesses, I decided to recommend rejection, as the paper does not meet the standard for acceptance.

---

### Decision · Program_Chairs · 2025-01-22

Reject